# CHEMDFM-R: A CHEMICAL REASONING LLM ENHANCED WITH ATOMIZED CHEMICAL KNOWLEDGE

## ABSTRACT

While large language models (LLMs) have achieved impressive progress, their application in scientific domains such as chemistry remains hindered by shallow domain understanding and limited reasoning capabilities. In this work, we focus on the specific field of chemistry and develop a Chemical Reasoning LLM, **ChemDFM-R**[1]. We first construct a comprehensive dataset of atomized chemical knowledge, *ChemFG*, annotating the presence of functional groups in molecules and the changes of functional groups during chemical reactions, to enhance the model's understanding of the fundamental principles and internal logic of chemistry. Then, we propose a *mixed-source distillation* method that integrates expertise in atomized knowledge with general reasoning skills, followed by domain-specific reinforcement learning to enhance chemical reasoning. Experiments on diverse chemical benchmarks demonstrate that ChemDFM-R achieves cutting-edge performance while providing interpretable, rationale-driven outputs. Further case studies illustrate how explicit reasoning chains significantly improve the model's reliability, transparency, and practicality in real-world human-AI collaboration scenarios.

## 1 INTRODUCTION

With the remarkable capabilities and performance demonstrated by large language models (LLMs) (Brown et al., 2020; Achiam et al., 2023; Team et al., 2023), the development of domain-specialized LLMs has emerged as a popular approach to addressing complex problems (Hendrycks et al., 2020; Chen et al., 2021; Wang et al., 2024). Specifically, many efforts focus on specializing LLMs into scientific domains to build general-purpose scientific assistants (Zhao et al., 2025b; Zhang et al., 2024a; Zhao et al., 2024; Zhang et al., 2025c; Tan et al., 2025) which could assist scientists during their research through conversational human-AI interactions. However, given the inherent complexity and high demand for reliability in the scientific domain, current models often struggle with inadequate performance and limited interpretability, which significantly hinders their practicality.

Recently, great success has been achieved in constructing reasoning LLMs in the general domain (Jaech et al., 2024; Guo et al., 2025a; Team et al., 2025; Comanici et al., 2025). Beyond enhancing overall model performance, the reasoning-before-answering pattern directly demonstrates how and why the LLM arrives at the answer, thereby markedly improving the reliability and interpretability of the LLM's response. Through the generated rationale, people can confirm the correctness of the answer or identify why the model makes mistakes. Therefore, reasoning-augmented LLMs offer a new promising approach to addressing the aforementioned challenges in scientific-domain LLMs, potentially enhancing their practical utility.

Currently, the research on reasoning LLMs has predominantly focused on general domains, such as mathematics (Yang et al., 2024b; Shao et al., 2024) and programming (Zhu et al., 2024; Hui et al., 2024). In contrast, the reasoning capability of existing LLMs remains highly limited in scientific domains. There are two reasons that hinder current LLMs from excelling in scientific-domain reasoning. 1) The understanding of domain knowledge remains superficial owing to the shortage of in-depth enough training data. The advanced domain knowledge is typically insufficient in general-purpose corpora, while even in current domain-specific corpora, the domain knowledge is still shallow. Current domain-specific corpora (Taylor et al., 2022; Xie et al., 2023; Zhao et al.,

---

[1]The inference code, training datasets, and the model parameters will be open-sourced.

2025b; Zhang et al., 2024a; Yu et al., 2024) often focus on literature text and high-level phenomena and fail to conduct a proper breakdown into internal mechanisms and atomized knowledge points. For example, in the field of chemistry, the types and positions of *functional groups*[2] within molecules fundamentally determine molecules' properties and reactivities. However, instead of training on functional-group-level knowledge, current chemical LLMs usually directly learn molecule-level knowledge about properties and reactivities. The lack of atomized knowledge significantly constrains the capacity of these models for providing high-quality rationales. 2) The intrinsic reasoning logic in these domains differs significantly from that in mathematics and programming, making it difficult for models to generalize relevant reasoning skills through training in general domains. This not only prevents general-domain LLMs from performing high-quality reasoning in scientific domains, but also introduces additional challenges for building domain-specific reasoning models, which usually involve distillation before reinforcement learning. The common distillation method involves gathering rationales from advanced reasoning LLMs, such as DeepSeek-R1 (Guo et al., 2025a) and o3-mini (OpenAI, 2025b), and training student models using supervised finetuning. This process assumes that the teacher LLM is capable of generating sufficiently reasonable rationales. However, this assumption often fails to hold in the scientific fields, such as chemistry. Owing to the limited understanding of atomized knowledge and chemical logic, even powerful general-domain reasoning models will highly probably fail to generate accurate and in-depth reasoning for chemical problems.

In this work, we focus on addressing the two aforementioned challenges in the chemistry domain and develop a chemical reasoning LLM, ChemDFM-R. Specifically, we consider the presence of functional groups in molecules and the changes of functional groups during reactions to be atomized knowledge points. We develop a toolkit to identify these features from molecules and reactions and incorporate them into the domain pretraining corpus. The resulting corpus contains over 101 billion tokens from 12 million literature, 30 million molecules, and 7 million reactions. To equip the model with the capability of chemistry-specific reasoning, we develop a mixed-source distillation process that can take advantage of both the expertise in the carefully curated knowledge points and the advanced reasoning capabilities of general LLMs. Domain-specific reinforcement learning is applied after the distillation process to integrate different capabilities the model learned from corresponding sources and further enhance the reasoning capability. The resulting model, ChemDFM-R, shows promising reasoning capabilities as well as advanced performance in multiple chemical benchmarks and can provide high-quality rationale, helping researchers deeply understand and verify its answer. In short, the contributions of this work are threefold:

- We developed a toolkit to identify the functional groups of molecules and the changes of functional groups during reactions. Using this toolkit, we built a 101-billion-token domain pretraining corpus, ChemFG, which encodes atomized **f**unctional-**g**roup–level chemical knowledge.

- We proposed a mixed-source distillation method to effectively initialize the model's reasoning ability under limited resources by incorporating both expertise in functional-group knowledge and distilled rationales. With subsequent reinforcement learning, we achieve a chemical reasoning LLM, ChemDFM-R.

- Extensive experiments demonstrate the promising chemical reasoning capability of ChemDFM-R. The created model achieves outstanding performance and manages to generate clear and rational rationales, which significantly boost the reliability and interpretability of the final answer.

## 2 CHEMFG

In the field of chemistry, functional groups serve as the bridge between molecular structures, properties, and reactivities, making them one of the most critical intermediate reasoning steps in chemical reasoning. As demonstrated in Figure 1, functional groups directly influence the properties of molecules and determine the types of reactions that can take place. However, existing training corpora of LLMs often lack detailed information on molecular functional groups, preventing models from directly and precisely learning this atomized chemical knowledge. Therefore, we collect a

---

[2]A group of atoms in a molecule with distinctive chemical properties. Please refer to https://en.wikipedia.org/wiki/Functional_group for more details.

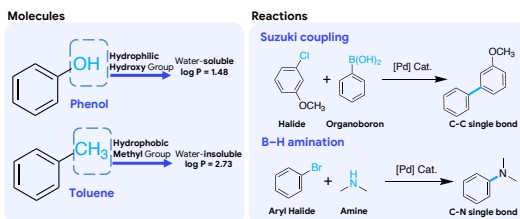

Figure 1: The influence of functional groups

Table 1: Data composition of ChemFG.

| Type | Sources | #Entries | #Tokens |
|---|---|---|---|
| Literature | Internet | 12M | 79B |
| Molecules | PubChem & PubChemQC | 30M | 22B |
| Reactions | USPTO-FULL | 7M | 0.6B |

functional-group-centered domain pretraining corpus, ChemFG, which consists of data from three sources: literature, molecules, and reactions. The basic statistics of ChemFG are shown in Table 1 with details provided in Appendix A.1.

## 2.1 FUNCTIONAL GROUP IDENTIFICATION

Despite the Internet-scale publicly available molecule and reaction corpora, there are no existing databases that describe the correspondence between functional groups and molecules or reactions. To tackle this issue, we develop a functional group identification toolkit based on `thermo` library[3] by extending its embedded SMARTS[4] list from 83 types of functional groups to 241 and improving its algorithms. With the help of the developed toolkit, we annotate the functional groups of all our domain-pretraining *molecule* data. Further details are provided in Appendix A.2.

As for *reactions*, we annotate the changes of functional groups during reactions with the following process. First, with the help of atom mapping annotations provided by the USPTO-FULL dataset, we identify the reaction centers as the atoms that are involved in bond changes during reactions. Based on these reaction centers and our functional group identification toolkit, we identify the functional groups of the reactants that directly participate in the reaction and those of the product that directly result from the reaction. Finally, the reaction can be described as a functional group transformation, where reacting functional groups are converted into product functional groups. Besides functional groups, there are other structural changes during reactions that are equally important, including ring breaking, ring forming, and bond changes outside functional groups. Therefore, we also construct tools to identify these changes in a similar manner.

## 2.2 QUALITY CONTROL

To ensure the annotation quality of functional groups, we hire three graduate-level chemical experts to conduct manual inspections. Firstly, all the experts agree that the extended SMARTS list has already covered the most common functional groups. For molecules, our tool's annotation accuracy of 100 random samples reaches 98%, with errors primarily due to corner cases such as rare functional groups or complex interactions between functional groups and aromatic rings. For the annotation of reactions, our tool achieves 89% accuracy when tested with 100 random samples. The errors mainly arise from invalid reactions or wrong atom mapping annotations. Examples and analyses of the error are demonstrated in Appendix A.3.

## 3 CHEMDFM-R

As outlined in Figure 2, the training pipeline of ChemDFM-R can be divided into two parts: 1) Domain Pretraining and Instruction Tuning (§ 3.1), where the basic general LLM is trained with atomized chemical knowledge; 2) Distillation and Reinforcement Learning (§ 3.2), where the model's chemical reasoning capability is enhanced.

---

[3] https://thermo.readthedocs.io/
[4] https://www.daylight.com/dayhtml/doc/theory/theory.smarts.html, SMiles ARbitrary Target Specification, a normalized form of SMILES.

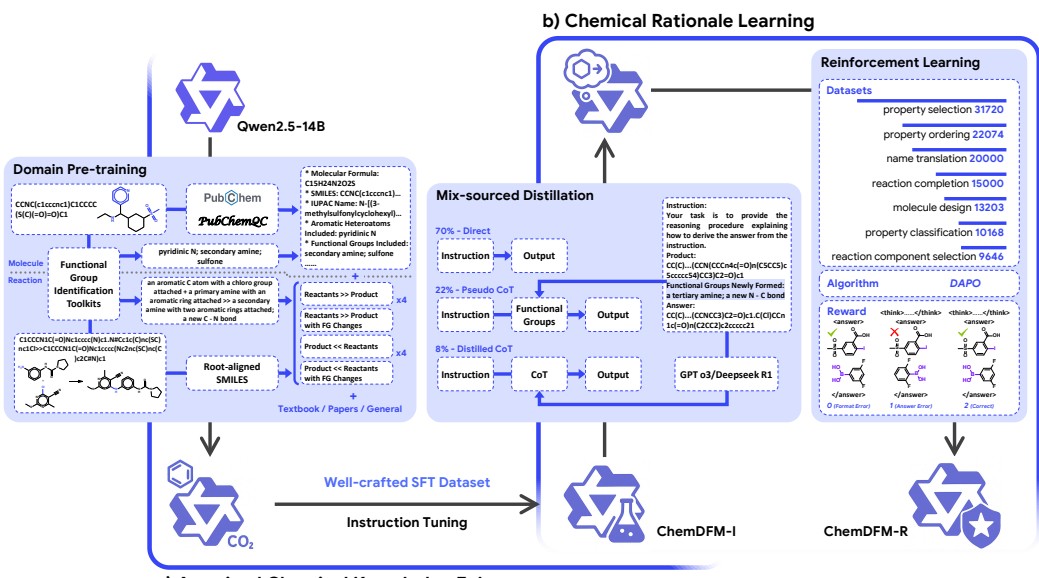

Figure 2: The overview of the training pipeline of ChemDFM-R.

## 3.1 ATOMIZED CHEMICAL KNOWLEDGE ENHANCEMENT

In this stage, our model mainly learns the atomized chemical knowledge to prepare itself with "ingredients" to "cook" the chemical rationales. Specifically, we achieve that through domain pretraining and instruction tuning.

**Domain Pretraining.** In domain pretraining, we leverage the 101-billion-token ChemFG corpus introduced in Section 2 to familiarize our model with the knowledge related to functional groups. We train our model from one of the most advanced general LLMs, Qwen2.5-14B (Yang et al., 2024a). Considering that general knowledge is also vital for Chemical LLMs, we also incorporate a substantial amount of general-domain pretraining data into our domain pretraining corpus to ensure that the model retains its general capabilities as much as possible.

**Instruction Tuning.** The primary goal of instruction tuning is to teach the model how to analyze the purpose and requirements of a given task and make proper use of the knowledge learned in the pretraining phase. However, existing instruction tuning datasets in the field of chemistry are typically derived from well-studied chemistry tasks and suffer from a severe lack of diversity of both task varieties and instruction expressions. Therefore, we construct a new instruction tuning dataset for ChemDFM-R based on the instruction tuning dataset of ChemDFM (Zhao et al., 2025b). To improve the overall task and instruction diversity, we introduce numerous new chemistry-related tasks, such as scientific paper QA, chemical property ordering, and reaction step prediction, and perform instruction-rewriting to achieve an average instruction-entry ratio of 1:50. For detailed information on the construction and composition of the instruction tuning dataset, please refer to Appendix B. To maintain the general capabilities of our model, we mixed our chemical instruction tuning data with general instruction tuning data in a 1:2 ratio. The Qwen2.5-14B model is finetuned for 2 epochs on this mixed dataset after domain pretraining, resulting in the ChemDFM-I model.

## 3.2 CHEMICAL RATIONALE LEARNING

The primary goal of this stage is to teach the model how to reason with the atomized knowledge it has acquired. Chemical reasoning requires a deep understanding of chemical principles and logic, as well as the capability to apply them for analysis. These capabilities can not be learned or induced from general-domain reasoning training. Therefore, we propose a chemical rationale learning pipeline to specifically enhance the chemical reasoning capabilities of LLMs based on distillation and reinforcement learning.

**Mixed-Source Distillation.** We leverage distillation to prevent the early unstable cold start phase of reinforcement learning. It could illustrate the reasoning patterns to the model and build up its basic reasoning capabilities.

Specifically, the entries in the distillation dataset come from three sources, each of which corresponds to part of the abilities required for chemical reasoning: 1) the instruction tuning dataset of ChemDFM-R (∼70%) to maintain basic chemical knowledge and prevent catastrophic forgetting; 2) pseudo-reasoning data describing the functional groups of involved molecules or reactions (∼22%) and highlighting vital intermediate reasoning steps and functional group analyses; 3) teachers' rationales from DeepSeek-R1 and o3-mini (∼8%)

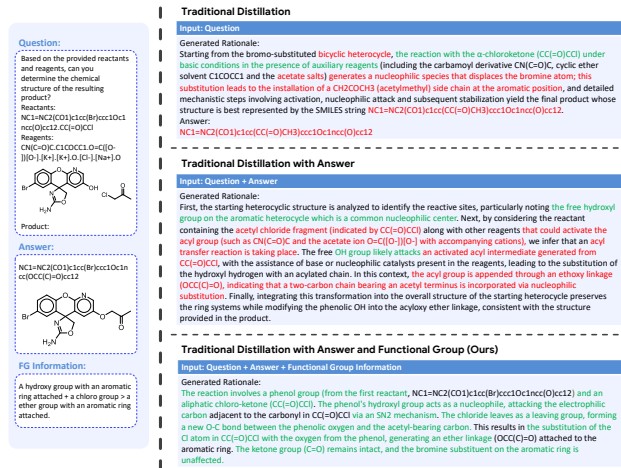

Figure 3: Comparison of rationales generated by o3-mini with and without additional input information. We mark the correct analyses in the rationale as green, while the wrong ones as red. For more examples and detailed analyses, please refer to Appendix C.1.

which introduce general reasoning patterns to the model and initiate its reasoning capabilities.

To improve the quality and efficiency of the teacher's rationales in the chemical domain, instead of asking teacher models to generate rationales from scratch, we provide them with rich additional information. Specifically, the teacher models are provided with the question instruction, the ground truth answer, and the functional group information of the molecules and reactions in the question. Comparison of the rationales generated by DeepSeek-R1 with and without the additional information is illustrated in Figure 3. The rationales generated with full additional information are significantly more valid and in-depth than the other two. More examples with detailed analysis of the rationale generation are given in Appendix C.1. Moreover, to comprehensively assess the quality of our distilled rationales, we sampled a small subset of them and conducted a quantitative human assessment. The results are illustrated in Appendix C.2.

Similar to instruction tuning, we mix our mixed-source distillation dataset with general data in a 1:2 ratio. The general data are also sampled from multiple sources, where ∼92% of the entries are sampled from the general data for instruction tuning of ChemDFM-R and ∼8% are from AM-Deepseek-R1-Distill-1.4M (Zhao et al., 2025a). The ChemDFM-I model is finetuned for one epoch on this mixed dataset.

**Reinforcement Learning.** After distillation, reinforcement learning (RL) is leveraged to further enhance the reasoning capabilities of our model. The composition of the RL dataset is illustrated in Figure 2, while more details of our reinforcement learning process are illustrated in Appendix D and E.

## 4 EXPERIMENTAL RESULTS

We first evaluate ChemDFM-R with multiple baselines on benchmarks specifically designed for assessing the chemical capabilities of LLMs (§ 4.1). Then, we validate the effectiveness of our methods by ablation studies (§ 4.2). Finally, to highlight the importance and effectiveness of chemical reasoning, we demonstrate its capability in facilitating reliable human-AI collaboration (§ 4.3).

### 4.1 BENCHMARK EVALUATION

We evaluate ChemDFM-R and the baseline models on two of the most popular and comprehensive benchmarks specifically designed for assessing the chemical capabilities of LLMs: SciKnowEval (Feng et al., 2024) and ChemEval (Huang et al., 2024). Given the large number of tasks included

Table 2: Benchmark results on SciKnowEval and ChemEval. "mol." stands for "molecule" and "react." stands for "reaction". Qwen-2.5-14B represent Qwen2.5-14B-Instruct. The numbers in the table represent the average performance over five inference runs, with the values in parentheses indicating the standard deviation. The best performance for each task is indicated using **boldface**. * We use RPS (Peng et al., 2025) to balance the different scales of the scores on different tasks in the ChemEval benchmark.

| Model | SciKnowEval | | | | ChemEval[*] | | | |
|---|---|---|---|---|---|---|---|---|
| | text | mol. | react. | all | text | mol. | react. | all |
| Qwen2.5-14B | $77.0_{(0.11)}$ | $35.7_{(0.23)}$ | $72.1_{(0.24)}$ | $61.3_{(0.12)}$ | $79.6_{(1.43)}$ | $24.8_{(0.63)}$ | $46.8_{(1.81)}$ | $57.7_{(0.72)}$ |
| ChemDFM-I | $\mathbf{78.8}_{(0.38)}$ | $50.6_{(0.40)}$ | $91.1_{(0.70)}$ | $\mathbf{69.7}_{(0.23)}$ | $\mathbf{81.2}_{(2.48)}$ | $68.4_{(3.56)}$ | $54.6_{(5.19)}$ | $70.7_{(1.10)}$ |
| ChemDFM-R | $76.7_{(0.18)}$ | $\mathbf{51.1}_{(0.56)}$ | $\mathbf{93.8}_{(0.37)}$ | $69.1_{(0.27)}$ | $78.3_{(2.38)}$ | $\mathbf{83.5}_{(1.89)}$ | $\mathbf{58.5}_{(2.61)}$ | $\mathbf{73.8}_{(1.83)}$ |

in SciKnowEval (19 tasks) and ChemEval (36 tasks), to facilitate fair and clear comparison, we categorized the tasks into three groups: text-centric, molecule-centric, and reaction-centric tasks. Details of the task categorization are provided in the Appendix G.

### 4.1.1 PERFORMANCE COMPARISON WITH BASELINES

First, we show the effectiveness of our training pipeline by comparing the performances of ChemDFM-R with those of 1) Qwen2.5-14B-Instruct (Yang et al., 2024a), which is the general-domain instruction tuning model of Qwen2.5-14B, and 2) ChemDFM-I, which incorporates atomized chemical knowledge enhancement but precedes the stage of chemical rationale learning. The quantitative results are illustrated in Table 2, while examples of the ChemDFM-R's rationales are analyzed in Appendix F.

As showcased in Table 2, ChemDFM-R consistently outperforms Qwen2.5-14B-Instruct on both SciKnowEval and ChemEval, demonstrating that our specialization pipeline has successfully improved the model's chemical capabilities. Specifically, the performances on text-centric tasks remain almost intact while those on molecule-centric and reaction-centric tasks increase significantly. This proves that our method manages to improve the chemical capabilities of LLM while largely maintaining its abilities in understanding natural language.

Moreover, we also evaluated ChemDFM-I to illustrate the contributions of the different stages in our training pipeline. Results show that **the atomized chemical knowledge enhancement stage consistently improves performance across all task categories**, while **the chemical rationale learning stage further strengthens performance on molecule- and reaction-centric tasks**. We attribute this phenomenon to two factors. *First*, the molecule- and reaction-centric tasks typically demand more domain-specific chemical reasoning, such as molecular property prediction or retrosynthesis analysis. In contrast, text-centric tasks rely more on natural language understanding, such as chemical named entity recognition and literature question-answering. As a result, learning chemical reasoning over molecules and reactions

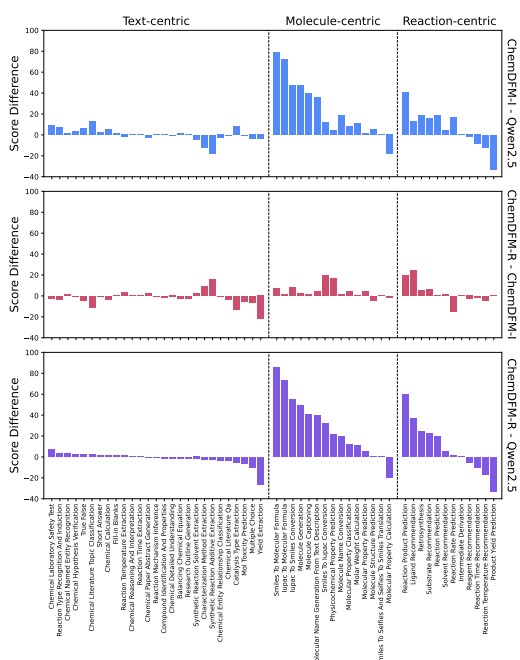

Figure 4: The performance change for all the individual tasks in SciKnowEval and ChemEval between Qwen-2.5-14B-Instruct, ChemDFM-I, and ChemDFM-R.

Table 3: Benchmark results on SciKnowEval and ChemEval. "mol." stands for "molecule" and "react." stands for "reaction". The best performance for each task is indicated using **boldface**, while the second-best is indicated using underline. * We use RPS (Peng et al., 2025) to balance the different scales of the scores on different tasks in the ChemEval benchmark.

| Model | SciKnowEval | | | | ChemEval* | | | |
|---|---|---|---|---|---|---|---|---|
| | text | mol. | react. | all | text | mol. | react. | all |
| MolInst | 69.4 | 38.4 | 41.7 | 55.0 | 47.3 | 15.6 | 8.9 | 29.1 |
| ChemLLM-20B-DPO | 73.6 | 38.4 | 88.1 | 62.1 | 64.5 | 25.4 | 17.5 | 42.2 |
| ChemDFM-13B-v1.0 | 70.2 | 42.6 | 85.0 | 61.6 | 61.5 | 42.3 | 34.0 | 49.2 |
| ChemDFM-8B-v1.5 | 72.2 | 45.6 | 86.9 | 63.9 | 26.7 | 12.6 | 18.3 | 21.1 |
| ether0 | 38.7 | 33.7 | 66.0 | 39.8 | 9.4 | 14.3 | 11.2 | 11.0 |
| GPT-4o | 76.1 | 44.6 | 44.6 | 61.2 | **82.7** | 35.0 | 52.9 | 63.3 |
| Qwen3-14B (no think) | 76.5 | 34.5 | 80.2 | 61.4 | 81.5 | 25.4 | 26.9 | 53.1 |
| DeepSeek-R1 | 80.1 | 30.8 | 83.0 | 62.2 | 81.2 | 19.7 | 47.9 | 57.6 |
| Qwen3-14B (think) | **86.6** | 40.8 | 88.5 | 69.9 | 79.0 | 24.7 | 40.7 | 55.7 |
| o4-mini | 81.3 | **57.0** | **97.3** | **74.0** | 78.4 | 60.7 | 53.6 | 67.3 |
| ChemDFM-R | 76.7 | 51.1 | 93.8 | 69.1 | 78.3 | **83.5** | **58.5** | **73.8** |

provides limited benefit to these text-focused tasks. *Second*, for the sake of answer verifiability, the reinforcement learning tasks do not include purely text-based tasks, which may adversely affect the model's text reasoning ability. Incorporating text-related tasks into the RL stage through joint training might help preserve performance on text-based tasks.

Furthermore, Figure 4 illustrates the performance changes across individual tasks. The results clearly show that most tasks benefit from our training pipeline, especially the molecule-centric tasks and reaction-centric tasks. Moreover, the two training stages provide complementary gains across different tasks, enabling the final model to achieve superior results on a broader range of tasks. Notably, among the tasks where ChemDFM-R does not surpass Qwen2.5-14B-Instruct, a substantial proportion involves numerical prediction, such as Yield Extraction, Molecular Property Calculation, and Product Yield Prediction. In fact, almost all the molecule-centric and reaction-centric tasks where ChemDFM-R falls short of Qwen2.5-14B-Instruct are those involving numerical reasoning and prediction. This pattern suggests that the numerical calculation and prediction abilities of ChemDFM-R are relatively weak, highlighting a potential direction for further improvements.

### 4.1.2 PERFORMANCE COMPARISON WITH ADVANCED LLMS

To further demonstrate the prowess of ChemDFM-R, we compare it with three sets of models: 1) existing chemical LLMs, including MolInst (Fang et al., 2024), ChemLLM (Zhang et al., 2024a), ChemDFM (Zhao et al., 2025b), and ether0 (Narayanan et al., 2025); 2) advanced non reasoning LLMs in the general domain, including GPT-4o (Hurst et al., 2024) and Qwen3-14B (no think) (Yang et al., 2025); 3) advanced reasoning LLMs in the general domain, including DeepSeek-R1 (Guo et al., 2025a), Qwen3-14B (think) (Yang et al., 2025), and o4-mini (OpenAI, 2025a). The experimental results are illustrated in Table 3. For detailed performances of individual tasks and more evaluations on FGBench (Liu et al., 2025b) and ChemCOTBench (Li et al., 2025b), please refer to Appendix G.

As shown in Table 3, ChemDFM-R significantly outperforms both the general-domain LLMs and domain-specific LLMs of similar size, especially in the molecule-centric and reaction-centric tasks. Specifically, on ChemEval, it even considerably surpasses Qwen3-14B, the next-generation model in the same series as our base model Qwen2.5-14B. When compared to cutting-edge LLMs, ChemDFM-R achieves better performance than GPT-4o and DeepSeek-R1, while demonstrating competitive results relative to o4-mini on SciKnowEval. Considering the tiny size of our model, this result strongly demonstrates the prowess of ChemDFM-R and the effectiveness of our specialization process.

Table 4: Ablation study results on SciKnowEval and ChemEval. "No Thinking" denotes the data source of the instruction-tuning dataset, "Pseudo" denotes the pseudo-reasoning data source, and "Distilled" denotes the teacher's rationale data source. The best performance for each task is indicated using **boldface**. * We use RPS (Peng et al., 2025) to balance the different scales of the scores on different tasks in the ChemEval benchmark.

| Data Source for Distillation | | | SciKnowEval | | | | ChemEval* | | | |
|---|---|---|---|---|---|---|---|---|---|---|
| No Thinking | Pseudo | Distilled | text | mol. | react. | all | text | mol. | react. | all |
| ✗ | ✗ | ✗ | **77.7** | 49.9 | 94.2 | 69.2 | 76.4 | 70.9 | 59.9 | 70.4 |
| ✗ | ✗ | ✔ | 75.5 | 49.6 | 87.9 | 67.2 | 72.0 | 68.8 | **63.6** | 68.9 |
| ✔ | ✗ | ✔ | 77.4 | 50.6 | 92.4 | 69.1 | 78.0 | 83.1 | 61.8 | 74.5 |
| ✔ | ✔ | ✔ | 76.8 | **52.0** | **94.5** | **69.5** | 80.3 | **84.5** | 61.3 | **75.8** |

## 4.2 ABLATION STUDY

We conducted two sets of ablation studies, which verify the two key components of ChemDFM-R's training pipeline, respectively: 1) atomized chemical knowledge enhanced domain pretraining (demonstrated in Appendix H); 2) mixed-source distillation (demonstrated below).

To validate the effectiveness of our newly designed mixed-source distillation method, we conduct an ablation study by gradually simplifying the composition of the distillation dataset. The results are shown in Table 4. The results prove that the traditional distillation method (Row 2) struggles to achieve positive impacts on performance in the chemical domain. It even underperforms the "zero" method (Row 1) proposed by Deepseek-R1, where there is no distillation stage before reinforcement learning. With the help of data sampled from the instruction-tuning dataset to maintain chemical capabilities and knowledge (Row 3), the model's performance gets boosted significantly. Moreover, the pseudo-reasoning data further help the model to achieve higher performance (Row 4), which corresponds to the final setting of our proposed mixed-source distillation method.

## 4.3 RELIABLE HUMAN-AI COLLABORATION

Another important advantage of ChemDFM-R's reasoning capability is that it allows humans to verify the correctness of answers, identify and correct errors, and discover new insights or perspectives. This enables more practical, reliable, and flexible human-AI collaboration. In this section, we first present human evaluation scores of the rationales generated by different reasoning models, which highlight the advantages of ChemDFM-R in facilitating more practical, reliable, and user-friendly human–AI interaction. We then demonstrate the prowess of ChemDFM-R in enabling reliable human–AI collaboration through illustrative examples. Specifically, an example is showcased in Figure 5, while more examples are illustrated in Appendix J.

### 4.3.1 HUMAN ASSESSMENT OF RATIONALE QUALITY

To assess the quality of rationales under practical situations, we constructed ten graduate-level questions based on recent publications from several influential chemistry journals. The questions cover different major subfields of chemistry, including organic chemistry, inorganic chemistry, materials chemistry, analytical chemistry, and polymer chemistry. Then, different reasoning LLMs are leveraged to solve these questions through human-AI interactions. Five graduate-level chemistry experts were hired to evaluate these interactions across five dimensions. The evaluation results are demonstrated in Table 5, while the questions and metrics are introduced in Appendix I. From these results, we draw three main conclusions:

First, our model outperforms all baselines, including DeepSeek-R1, in both chemical correctness and answer accuracy, indicating that it possesses a stronger grasp of chemical knowledge. In particular, our model shows a clearer advantage on SMILES-related tasks, demonstrating a more precise understanding of molecular structures.

Second, our model achieves significantly higher scores in effective information density. Unlike models such as Qwen3-14B and DeepSeek-R1, which often generate extremely lengthy reasoning chains

Table 5: Human assessment of the quality of rationales from different reasoning LLMs. Please refer to Appendix I for the introduction of the five metrics and the detailed questions used in this assessment. As the rationales of o4-mini are not accessible, we only use the answer provided by o4-mini. All metrics are higher-is-better.

| Models | Chemical Correctness | Answer Accuracy | Analytical Coverage | Reasoning Coherence | Effective Information Density |
|---|---|---|---|---|---|
| ether0 | 1.56 | 1.60 | 1.40 | 1.76 | 1.56 |
| DeepSeek-R1 | 4.10 | 3.88 | **4.34** | 4.42 | 3.62 |
| Qwen3-14B (think) | 3.56 | 3.92 | 4.24 | 3.98 | 3.00 |
| o4-mini | 3.94 | 3.96 | 3.84 | 4.16 | 4.12 |
| ChemDFM-R | **4.52** | **4.28** | 3.94 | **4.62** | **4.70** |

exceeding 1,000 tokens, our model produces concise yet informative reasoning. This considerably reduces the burden on human users when verifying correctness or identifying errors.

Finally, our model performs slightly lower than Qwen3-14B and DeepSeek-R1 on analytical coverage. This is partially because their extremely lengthy reasoning chains allow them to enumerate many possible considerations. It is also true that our model commonly focuses on one or a few major possibilities rather than adequately enumerating all potential factors. This highlights a productive direction for future enhancement of our model's analytical breadth.

### 4.3.2 PRACTICAL CASES OF HUMAN-AI COLLABORATION

To make our example more realistic, we design our conversation topic according to a newly published chemical research paper, Li et al. (2025a). It is worth noticing that this paper was published in 2025, so it is impossible for ChemDFM-R to encounter this paper during either the pretraining of the base model, Qwen2.5-14B, or our domain-pretraining process. To demonstrate the potential of ChemDFM-R, we assume the role of a researcher seeking a new research inspiration and engage the model in a dialogue about the subfield to which this paper belongs.

First, as a researcher seeking inspiration, we ask ChemDFM-R to introduce the oxygen evolution reaction (OER) briefly, including the definition and mechanism of it. Although the answer itself is a relatively rigid response to the question, the model's rationale reveals interesting insights. In its rationale, ChemDFM-R mentions that OER requires a transition metal oxide catalyst, which often presents valuable opportunities for further research. Therefore, we further ask it to give more detailed recommendations for the catalysts. As a response, ChemDFM-R proposes multiple different oxides, which is a wide range

Figure 5: An example of reliable human-AI collaboration using ChemDFM-R. This conversation is inspired by Li et al. (2025a). This example illustrates a process of research inspiration seeking with the help of ChemDFM-R.

for us to dig into. However, in its rationale, ChemDFM-R itself says that "iridium oxide (IrO2) is

*particularly notable* for its ...", which is very inspiring. Since it is well known that iridium-based compounds are often very expensive, a natural follow-up question arises: how can we optimize this catalyst to reduce its cost while maintaining its catalytic performance? Surprisingly, ChemDFM-R manages to propose the initial ideas that align closely with the ideas presented in Li et al. (2025a), "forming mixed metal oxides or heterostructures" and "optimizing the morphology and surface area of the catalysts". At this point, a broad research direction has taken shape.

It is worth noticing that nearly all the inspirations are drawn from the rationale of ChemDFM-R, which demonstrates the significance and value of ChemDFM-R's ability to generate reasoning. This example shows that, with the enhanced chemical knowledge and strong chemical reasoning capabilities, ChemDFM-R has the potential to facilitate reliable human-AI collaboration, thereby advancing AI-driven research and applications. More examples involving error correction and answer improvement with the help of rationales are demonstrated in Appendix J.

## 5 RELATED WORK

**General Domain Reasoning LLMs.** Shortly after the emergence of LLMs, their remarkable reasoning capabilities were discovered by Kojima et al. (2022) and explored by works such as ToT (Yao et al., 2023) and PAL (Gao et al., 2023). Recently, OpenAI-o1 (Jaech et al., 2024) followed by DeepSeek-R1 (Guo et al., 2025a) and Kimi K1.5 (Team et al., 2025) demonstrated the prowess of reasoning models and the method to enhance LLMs' reasoning capabilities using reinforcement learning-based pipelines. Subsequently, many studies have focused on improving the reasoning capabilities of models in various general domains, primarily in mathematics and coding. For example, Shao et al. (2024) and Zhang et al. (2024b) have further proven and discussed the effectiveness of reinforcement learning in terms of enhancing models' reasoning capabilities, while Dou et al. (2024) and Zhang et al. (2025d) have explored better reward functions in mathematics and coding.

**Chemical LLMs.** The specialization of LLMs has become one of the most popular research areas after the emergence of general-use LLMs, including the development of chemical LLMs. LlaSMol (Yu et al., 2024) and Mol-Instruct (Fang et al., 2024) construct a chemical instruction tuning dataset and develop models that could excel in multiple chemical tasks, while extensive training with only chemical tasks has led to a substantive loss of natural language capabilities and task generalization ability in these models. Shortly after, Zhang et al. (2024a) leveraged high-quality instruction tuning and developed ChemLLM, which has acquired advanced chemical capabilities while retaining a considerable level of general language abilities. Furthermore, ChemDFM (Zhao et al., 2025b) achieved stronger chemical and generalization capabilities through domain pretraining and instruction tuning with both chemical data and general-domain data. It is worth noticing that the data used in previous work overlooks the intrinsic chemical essence, which is crucial for LLMs to master reasoning with chemical intuition and principles. To tackle this issue, we construct a function-group-centric domain pretraining corpus to introduce such atomized chemical knowledge to LLMs. Recently, there has been pioneering work building chemical reasoning models for specific tasks, such as RetroDFM (Zhang et al., 2025b) for retrosynthesis and multi-task specialist ether0 (Narayanan et al., 2025). Although their models exhibit strong reasoning capabilities and favorable performance on the training tasks, they often lack generalization ability. Consequently, they cannot accommodate the complex scenarios and demands that arise in real human–AI interactions, making it difficult for them to meet the requirements of a practical chemistry research assistant.

## 6 CONCLUSION

In this work, we have developed a chemical reasoning LLM, ChemDFM-R, by tackling both the limitations in understanding atomized chemical knowledge and the domain-specific reasoning logic. By incorporating atomized knowledge about molecular functional groups and their changes during reactions into the pretraining corpus, and applying a mixed-source distillation approach before reinforcement learning, we have enhanced the model's ability to reason efficiently and effectively in chemistry. Our extensive experiments demonstrate that ChemDFM-R significantly improves chemical problem-solving and reasoning capabilities, making it a valuable tool for facilitating reliable human-AI collaboration in chemistry and advancing AI-driven research and applications.

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

## DISCLOSING OF LLM USAGE

LLMs are leveraged exclusively as an assist tool for paper writing. We only use them to help us polish the language and improve readability. They do not contribute to any of the creative work involved in this paper, such as research ideation, experiment design, data analysis, and result interpretation. Therefore, they should not be regarded as authorship or substantive contribution.

# A    DETAILS ABOUT CHEMFG

## A.1    RAW DATA COLLECTION

**Literature.**    Literature, including papers and textbooks, contains not only the widely accepted chemical knowledge and principles, but also the cutting-edge research in the field of chemistry. Therefore, to take full advantage of chemical literature, we collected over 12 million literature from the open Internet dated prior to January 2022. After further cleaning and deduplication, 79B tokens are obtained from it.

**Molecules.**    Molecules are the fundamental participants in various chemical processes. Therefore, it is crucial for Chemical LLMs to understand molecular structures and properties. We manage to acquire large-scale molecule datasets from PubChem[5], one of the biggest open accessible chemical databases with more than 100M compounds. We include 30 million molecules along with their notations, descriptions (if applicable), and properties. Besides PubChem, we also leverage the PubChemQC (Nakata & Shimazaki, 2017) dataset, which contains the quantum chemical calculation results of 86M molecules from PubChem, to supplement the quantum chemical properties of these molecules, such as dipole moment and orbital energy. To diversify the final data entry, we randomize the order of the properties and use three different formats: markdown list, markdown table, and JSON dictionary to formulate the molecule data.

**Reactions.**    Reactions are the major process in the chemical world. In ChemFG, we use the reactions from USPTO-FULL (Dai et al., 2019), one of the most comprehensive open-sourced chemical reaction databases. To avoid data leakage, we exclude the test set of USPTO-FULL, USPTO-MIT (Jin et al., 2017), and USPTO-50K (Schneider et al., 2016) according to the products of reactions. Moreover, to further enhance the data diversity, we leverage the SMILES (Simplified Molecular Input Line Entry System) augmentation method introduced in R-SMILES (Zhong et al., 2022) and achieve a total of 10 times augmentation of data. Finally, a corpus of 7 million reactions is obtained.

## A.2    FUNCTIONAL GROUPS COVERAGE

The functional groups that can be recognized by our toolkit are categorized based on the heteroatoms and listed as follows:

- **Hydrocarbon Groups (7)**: alkene, alkyne, allene, cumulene, carbocation, carbanion, carbene.

- **Boron Groups (6)**: borane, boronic acid, boronic ester, borinic acid, borinic ester, borate ester.

- **Oxygen Groups (36)**: alcohol, alkoxide, ether, phenol, phenolate, enol, enolate, enol ether, alkynol, alkynolate, alkynol ether, ketone, ketene, aldehyde, hemiketal, hemiacetal, ketal, acetal, carboxylic acid, carboxylate, ester, organic acid anhydride, carboxylic anhydride, organic carbonate, organic hydroperoxide, organic peroxide, peroxyacid, ortho ester, ortho-carbonate ester, methylenedioxy, ethylenedioxy, oxonium ion, oxocarbenium ion, carbonyl ylide, oxonium ylide, epoxy.

- **Nitrogen Groups (62)**: primary amine, secondary amine, tertiary amine, ammonium cation, quat, amine oxide, enamine, hydroxylamine, hemiaminal, hemiaminal ether, thioaminal, thioaminal ether, aminal, primary ketimine, secondary ketimine, primary aldimine, secondary aldimine, amidine, guanidine, ketoxime, aldoxime, hydrazone, organic amide, amidate anion, imide, carbamic acid, carbamate ester, carbamate anion, azide, azo, hydrazine, acylhydrazine, amidrazone, cyanate, isocyanate, nitrile, isonitrile, cyanamide, carbodiimide, nitrate ester, nitrite ester, nitro, nitroso, nitrosamine, iminium cation, nitrone, nitronic acid, imidic acid, imidate anion, imidate, imidocarbonate, imidocarbamate, urea, azoxy, N-oxoammonium, hydroxamic acid, hydroxamate, azanide, azomethine ylide, nitrile ylide, isodiazene, nitronate.

---

[5] https://pubchem.ncbi.nlm.nih.gov/

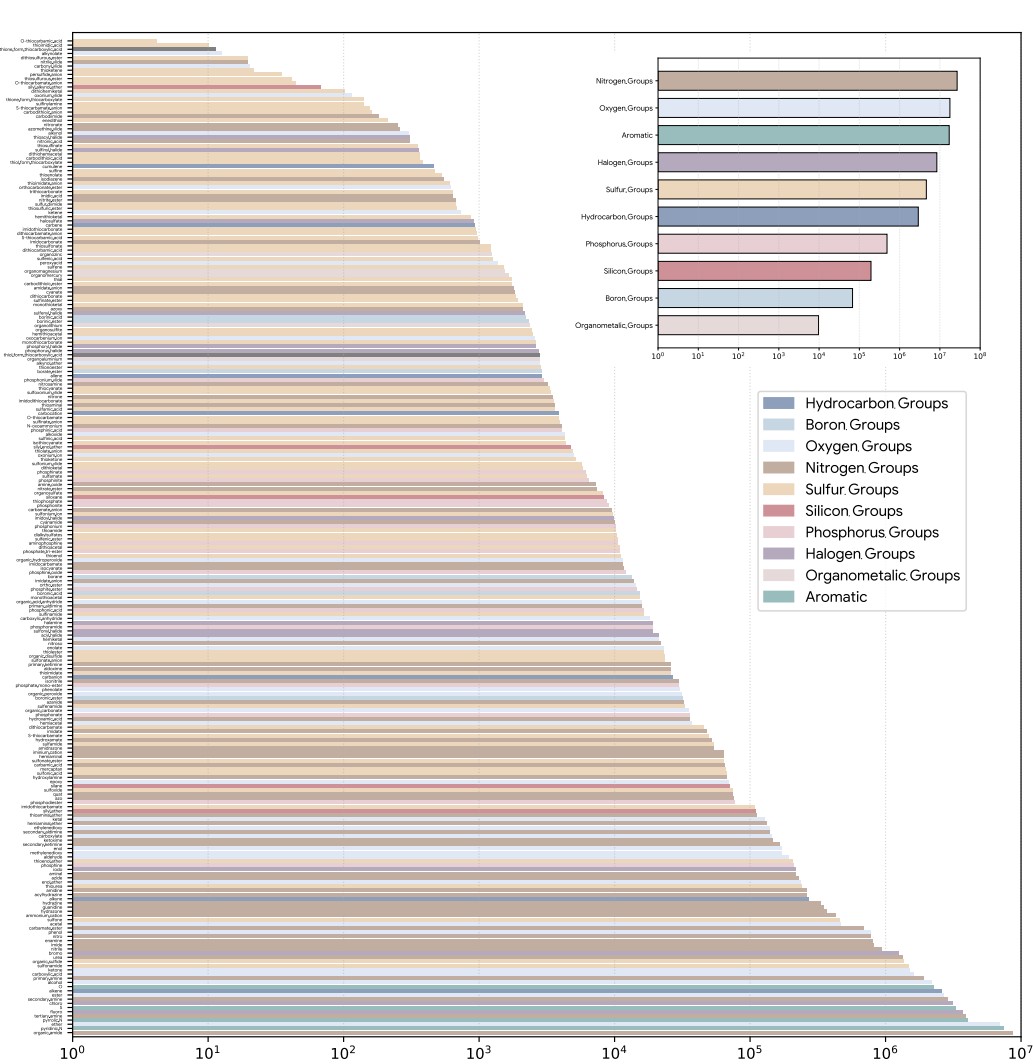

Figure 6: The distribution of the functional groups in the domain-pretraining corpus.

- **Sulfur Groups (85)**: mercaptan, thiolate anion, organic sulfide, thioenol, enedithiol, thioenolate, thioenol ether, persulfide anion, organic disulfide, sulfenic acid, sulfenic ester, sulfenamide, sulfoxide, sulfone, sulfine, sulfene, sulfinylamine, sulfur diimide, sulfinic acid, sulfonic acid, sulfinate ester, sulfonate ester, sulfinate anion, sulfonate anion, thiosulfinate, thiosulfonate, thiosulfurous ester, dithiosulfurous ester, thiosulfuric ester, organosulfite, organosulfate, dialkylsulfates, sulfinamide, sulfonamide, sulfamic acid, sulfamate, sulfamide, thiocyanate, isothiocyanate, thioketone, thioketene, thial, thioamide, thiourea, hemithioketal, hemithioacetal, dithiohemiketal, dithiohemiacetal, monothioketal, monothioacetal, dithioketal, dithioacetal, carbothioic S-acid, carbothioic O-acid, thiol form thiocarboxylate, thione form thiocarboxylate, thiolester, thionoester, carbodithioic acid, carbodithioic anion, carbodithioic ester, monothiocarbonate, xanthic acid, xanthate, xanthate anion, dithiocarbonate, trithiocarbonate, O-thiocarbamic acid, S-thiocarbamic acid, O-thiocarbamate, S-thiocarbamate, O-thiocarbamate anion, S-thiocarbamate anion, thioimidic acid, thioimidate anion, thioimidate, dithiocarbamic acid, dithiocarbamate, dithiocarbamate anion, imidothiocarbonate, imidodithiocarbonate, imidothiocarbamate, sulfonium ion, sulfonium ylide, sulfoxonium ylide.

- **Silicon Groups (5)**: silane, siloxane, silyl ether, silyl enol ether, silyl alkynol ether.

- **Phosphorus Groups (17)**: phosphine, phosphonium, aminophosphine, phosphine oxide, phosphinic acid, phosphinate, phosphonic acid, phosphonate, phosphite ester, phosphinite, phosphonite, phosphodiester, phosphate mono-ester, phosphate tri-ester, phosphoramide, thiophosphate, phosphonium ylide.

- **Halogen Groups (14)**: fluoro, chloro, bromo, iodo, halamine, sulfenyl halide, sulfinyl halide, sulfonyl halide, halosulfate, phosphoryl halide, phosphorus halide, acyl halide, imidoyl halide, thioacyl halide.

- **Organometalic Groups (5)**: organolithium, organomagnesium, organoaluminium, organozinc, organomercury.

- **Aromatic (4)**: pyrrolic N, pyridinic N, aromatic O, aromatic S.

The occurrence of these functional groups in the domain-pretraining corpus is shown in Figure 6.

## A.3   QUALITY CONTROL

To validate the correctness of our functional group identification toolkit, we hired three graduate-level chemical experts to conduct manual inspections. Specifically, we sampled 100 annotated molecules and reactions, respectively, and asked the experts to determine whether the annotations were correct. Results show that our functional group identification toolkit achieves 98% accuracy rate on molecules and 89% on reactions. Examples of the errors are demonstrated in Figure 7.

## B   INSTRUCTION TUNING DATASET

### B.1   RAW DATA COLLECTION

Our instruction tuning dataset is constructed of three parts corresponding to the three main information carriers in chemistry: molecule-centric tasks, reaction-centric tasks, and knowledge-centric tasks. The distribution of instruction tuning data is shown in Figure 8.

### B.1.1   MOLECULE-CENTRIC TASKS

- **Name Translation**: The name translation between SMILES, IUPAC name, and molecular formula. The data is constructed from PubChem[6].

- **Description Generation**: The molecule description task is to describe the molecule given its SMILES. The data is constructed from PubChem. We only use the high-quality descriptions that contain more than two sentences.

---

[6]https://pubchem.ncbi.nlm.nih.gov/

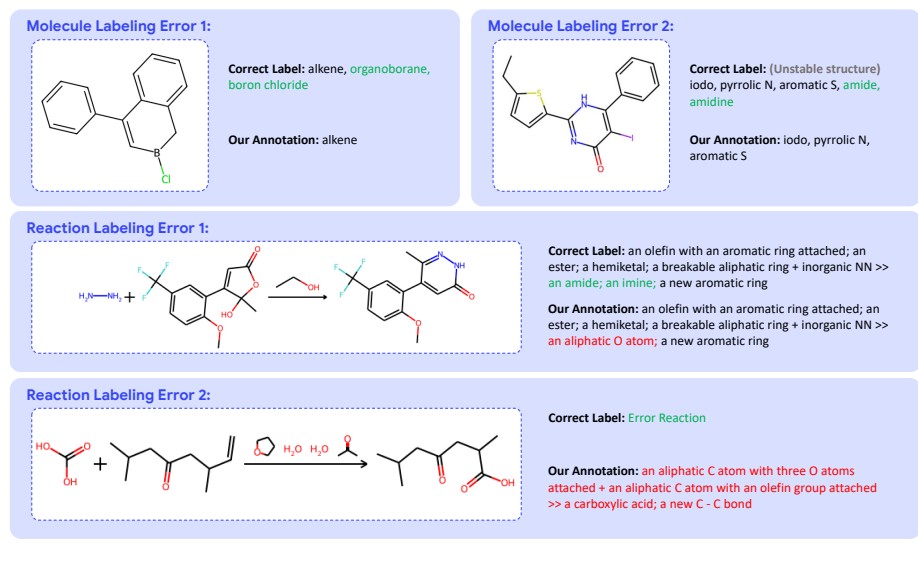

Figure 7: Examples of the error annotations of our functional group identification toolkits.

- **Molecule Design**: The molecule design task is the reverse task of molecule description. It requires the model to predict the SMILES given the molecule description. We use the same high-quality description data from PubChem to construct this task.

- **Property Classification**: These tasks require models to predict the value of molecular properties from a list of candidates (usually yes and no). The data is constructed from 5 of the most popular property classification datasets in MoleculeNet (Wu et al., 2018), namely BACE, BBBP, ClinTox, HIV, and Tox21.

- **Property Regression**: These tasks require the models to predict the value of molecular properties, which is a real number. Data are also from MoleculeNet, namely FreeSolv, Lipo, and QM9.

- **Property Ordering**: Provided a list of molecules, models are asked to rank them in ascending or descending order of some specific property. Raw data comes from the same source as property regression.

- **Property Selection**: Provided a list of molecules, models are asked to select the one with the highest or lowest value of some specific property. Raw data comes from the same source as property regression.

### B.1.2 REACTION-CENTRIC TASKS

- **Reaction Completion**: Given an incomplete reaction, models need to complete the missing reactants, reagents, or products. Raw data comes from USPTO-Full (Dai et al., 2019), USPTO-MIT (Jin et al., 2017), and USPTO-50K (Schneider et al., 2016).

- **Step Prediction**: Given a reaction, models are required to predict the experimental procedure to conduct it in the laboratory. Raw data comes from USPTO (Dai et al., 2019).

- **Yield Prediction**: In this task, models are required to predict the yield of the given reactions. The data is constructed from the USPTO dataset.

- **Temperature Prediction**: In this task, models are required to predict the temperature that is suitable for the given reactions to conduct. The data is constructed from the USPTO dataset.

- **Reaction Component Selection**: In this task, a series of reactants and reagents is given with a list of candidate molecules. Models need to pick from the candidates the molecules that could participate in the reaction and lead to the highest yield. The data is constructed from the USPTO dataset.

### B.1.3 KNOWLEDGE-CENTRIC TASKS

- **Exam Questions**: This task is composed of questions from the exams in middle school and high school. Raw data comes from the Open Internet.

- **Literature QA**: In this task, models are required to answer questions based on the given paragraph. The data is extracted from the long paragraph following the method in SciQAG (Wan et al., 2024). The raw data comes from the articles in the domain-pertaining. The articles are split into sections and then truncated into paragraphs within 2k to 3k tokens. We ask GPT-4o-mini to extract 15 keywords from each paragraph, then generate 10 question-answer pairs according to them. We adopt another LLM, Qwen2.5-14B-Instruct, to evaluate the quality of the QA pair in 4 dimensions: completeness, accuracy, reasonableness, and agnosticism. The LLM will score the QA pair from 1 to 5 using the designed prompts. QA pairs with any scores below 5 are discarded. If there are more than 1 QA pair left, the questions are asked in conversation turns.

- **Literature Summarization**: In this task, models are required to give a summarization of the paragraph. The summarization is generated from GPT-4o-mini from the paragraph sample.

- **Literature Translation**: In this task, models are required to translate the English paragraph into Chinese. The translation is generated from GPT-4o-mini from the paragraph sample. Since the source data consists of OCR text extracted from English articles, which is inherently noisy, we decided to discard the reverse task of translating Chinese paragraphs into English.

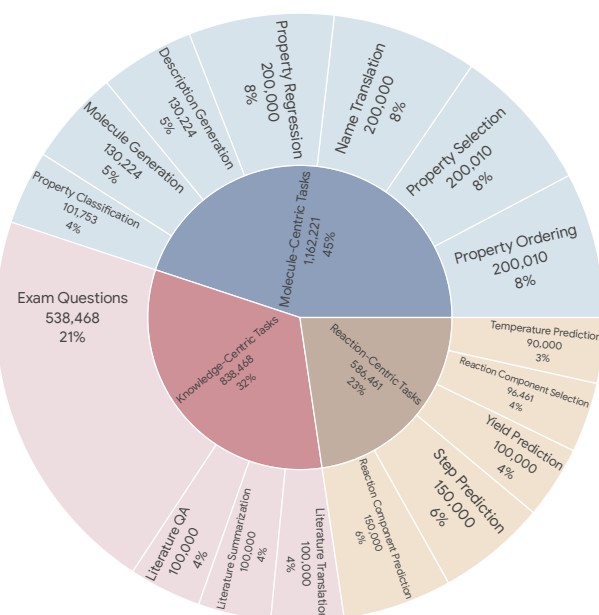

Figure 8: The distribution of instruction tuning data.

### B.2 INSTRUCTION GENERATION

To acquire a higher generalization capability, we adopt a two-stage process to obtain as diverse a set of instructions as possible for each task. Specifically, based on the number of data entries for each task, we first manually write 5-20 seed task descriptions accordingly. Then, we ask three different models, Qwen2.5-72B-Instruct, Llama-3.1-70B-Instruct, and GPT-4o-mini, to diversify these task descriptions. Specifically, during each request, we sample 5 descriptions from all the generated descriptions and ask the model to generate 10 new descriptions using 5 different prompts one by one. Following this, we append to each of the descriptions the instructions that introduce the input of each data entry by a formatted string of Python and ask GPT-4o-mini to further diversify them. As

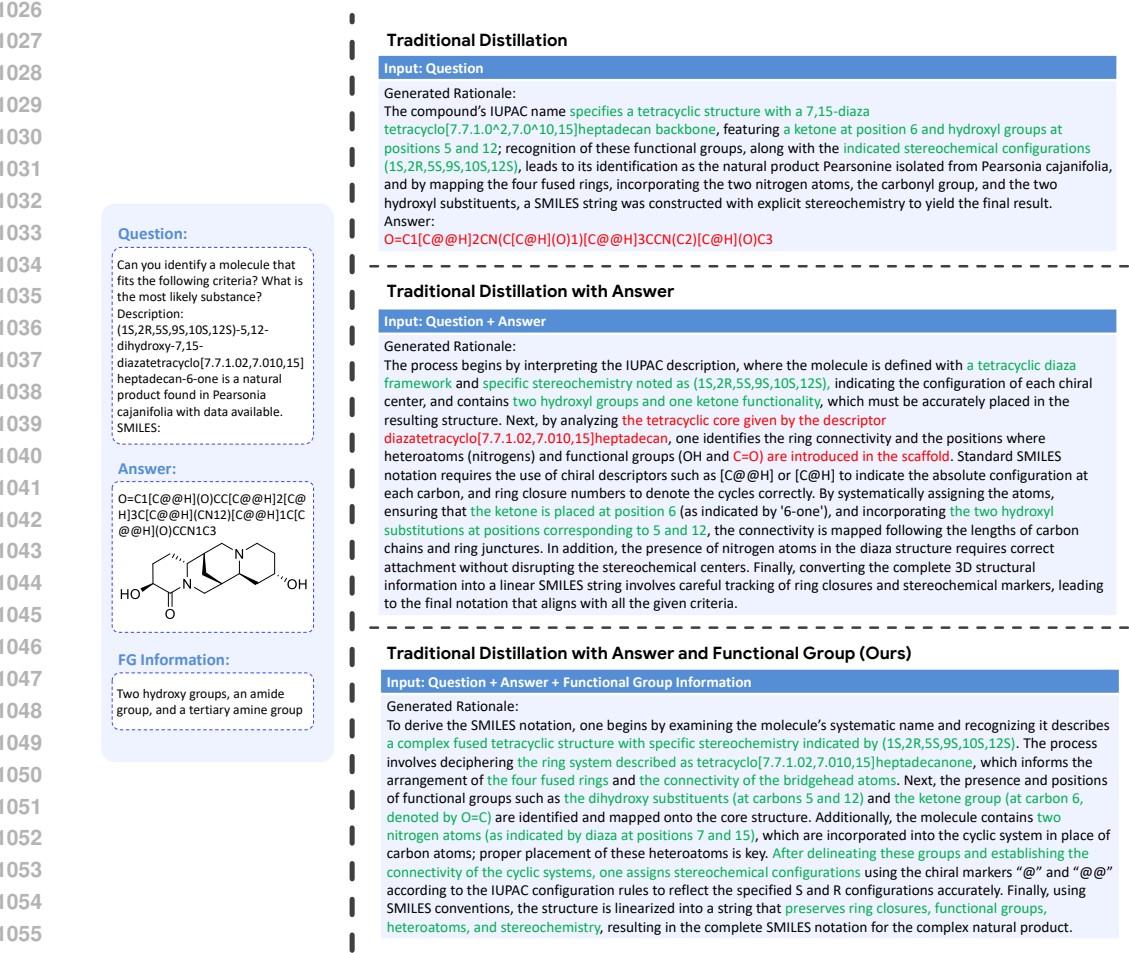

Figure 9: Comparison of rationales generated by o3-mini with and without additional input information. We mark the correct analyses in the rationale as green, while the wrong ones as red.

a result of the above diversification process, we constructed a number of instructions equivalent to approximately 2% of its total data volume for each task. In other words, each instruction appears around 50 times in the instruction-tuning dataset.

# C    QUALITY CONTROL OF THE DISTILLATION RATIONALE GENERATION

## C.1    COMPARISON OF RATIONALES GENERATED BY TEACHER MODELS

As shown in Figure 3, 9, and 10, the rationales generated using our method exhibit distinct advantages in terms of accuracy, completeness, and logicality.

**The advantage in accuracy is mainly reflected in Figure 3.**    In Figure 3, o3-mini completely misunderstands the chemical reaction that occurs between the given molecules. This might result from its incorrect identification of the reagents and the structure of the complex reactants (these errors are also reflected in the rationales). When given the correct answer, o3-mini still mistakenly identified chloro-ketone as acyl chloride and consistently adhered to this error throughout the reasoning process, resulting in a series of related structural inaccuracies. On the contrary, when the functional group information is provided, o3-mini manages to generate a near-perfect rationale with zero error.

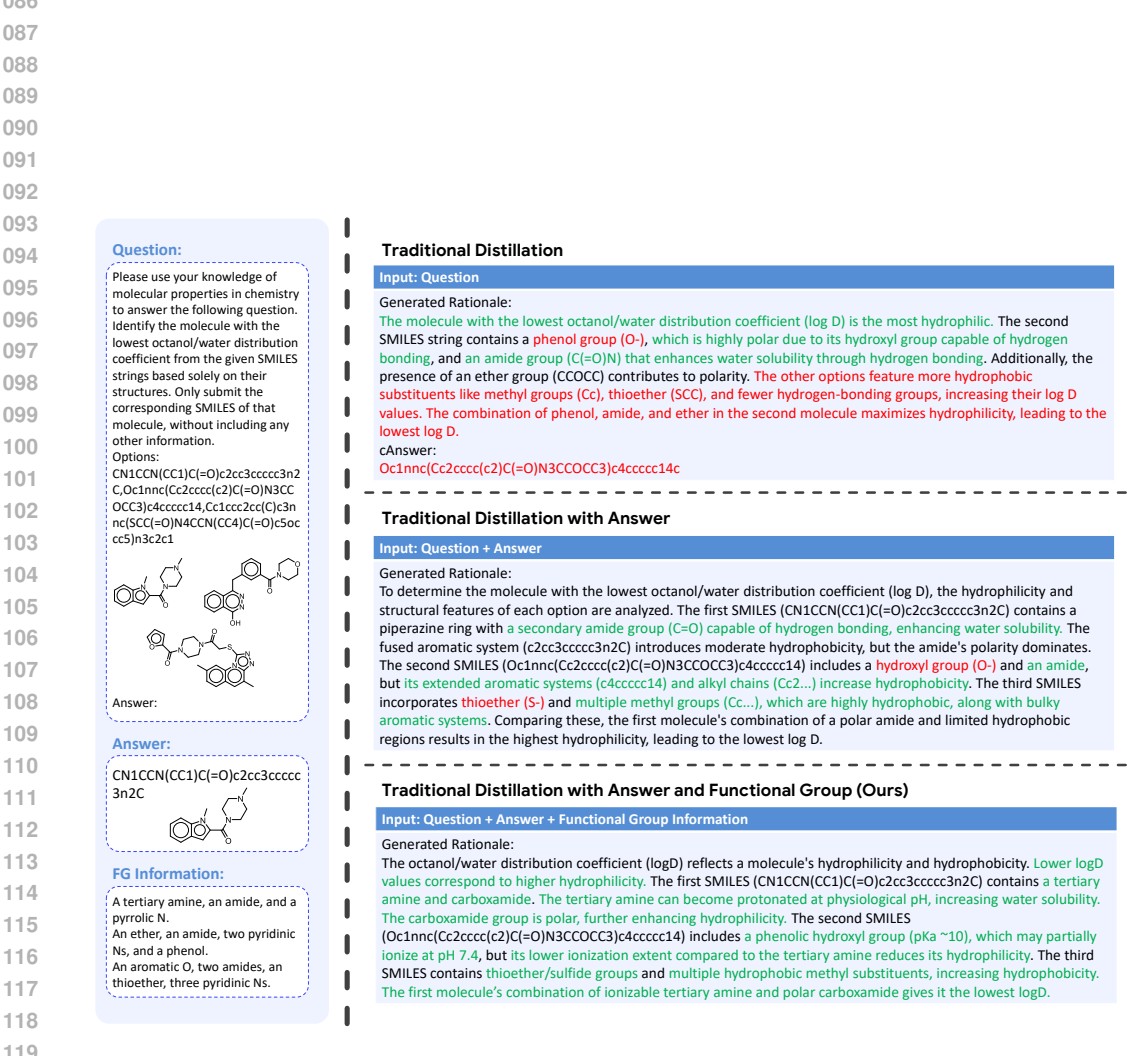

Figure 10: Comparison of rationales generated by DeepSeek-R1 with and without additional input information. We mark the correct analyses in the rationale as green, while the wrong ones as red.

**The advantage in completeness is mainly reflected in Figure 9.** In Figure 9, given only the question, DeepSeek-R1 only generates a brief analysis on the second option while dismissing the other three options with a single sentence in total. This overly simple analysis leads to a wrong prediction. With the help of the ground truth answer, the generated rationale analyzes all options individually. However, due to its lack of chemical knowledge, the analysis still exhibits errors in functional group recognition or overlooks key influencing factors. After enhancing chemical knowledge with the functional group information, DeepSeek-R1 finally manages to generate a more comprehensive analysis with few errors.

**The advantage in logicality is mainly reflected in Figure 10.** In Figure 10, with only the question, o3-mini can hardly generate any useful rational. The rational merely repeats the IUPAC components mentioned in the question before rushing to a highly inaccurate conclusion without substantive analysis. When given the ground truth answer, o3-mini can construct a reasonably good rationale with minimal factual error. However, the rationale still contains non-negligible issues in terms of logical coherence. A sound reasoning process should follow the approach exemplified by the reasoning chain generated by o3-mini using our method: analyzing in the order of molecular skeleton, functional groups, heteroatoms, and chiral centers. This sequence reflects a step-by-step refinement from the fundamental molecular structure to more intricate structural details. However, with only the question and answer, the generated rationale mixes these analytical steps and lacks critical details, such as the precise position of the nitrogen atom, resulting in a disorganized and incomplete reasoning process.

## C.2 QUALITY CONTROL

To quantitatively validate the quality of the teachers' rationales generated by our method, we hired three graduate-level chemical experts to perform manual assessments. Results in Figure 11 show that among the sampled 264 rationales, 177 of them (67%) exhibit sufficiently high quality, 60 of them (23%) have minor, acceptable flaws such as reasoning step skipping or missing possibilities, and 27 of them (10%) contain substantive issues such as logic errors or nonsense reasoning. Two examples of distilled rationales with minor flaws are shown in Figure 12, and two examples of distilled rationales with substantive issues are shown in Figure 13. Considering that reinforcement learning will be used after distillation to correct errors and improve performance, and that it is difficult to systematically distinguish these problems in the rationales, we used all the data during the distillation process.

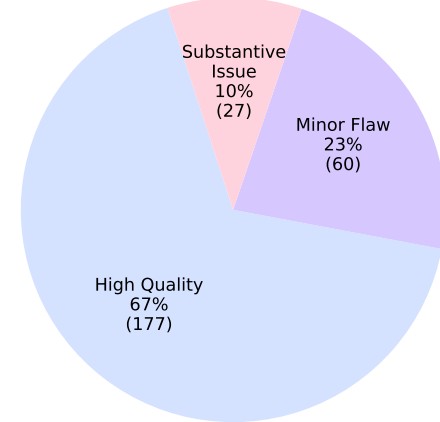

Figure 11: Human validation result of teachers' rationales. "Minor Flaw" represents acceptable flaws, such as skipping reasoning steps or missing possibilities. "Substantive Issue" means severe logic errors or nonsense reasoning.

## D DETAILS OF REINFORCEMENT LEARNING

To construct the reinforcement learning dataset, a subset of tasks that have verifiable answers is selected from our instruction-tuning dataset. We sampled data from the same sources used to build the instruction-tuning dataset, while minimizing overlap between the specific instances contained in the two datasets. These tasks are mixed together and uniformly sampled during training. We use the DAPO (Yu et al., 2025) algorithm for reinforcement learning. As for the reward system, we follow the method recommended by DeepSeek-R1 (Guo et al., 2025a), where the reward system consists of format rewards and accuracy rewards. The format rewards evaluate whether a response strictly follows the reasoning format, and the accuracy rewards evaluate whether a response is correct. Specifically, considering the redundancy of the casual SMILES notations (one molecule could correspond to multiple SMILES), we first canonicalize all the SMILES in the response before calculating the accuracy rewards.

**Example 1 of Distilled Rationales with Minor Flaws:** Overly simplify description of the reaction process

**Instruction:** Predict the product of the reaction using the given reactants and reagents, and express it in SMILES notation.
Given:
 Reactants: Cc1cc2c3ccccc3ccc2c2ccccc12.O=C1CCC(=O)N1Br
 Reagents: ClCl1C=CC=CC=1.CO.C(OOC(=O)C1C=CC=CC=1)(=O)C1C=CC=CC=1
Provide the resulting product and its SMILES notation.

**Rationale:**
The reactants are a methyl-substituted polycyclic aromatic compound (SMILES: Cc1cc2c3ccccc3ccc2c2ccccc12) and a brominated imide halamine (SMILES: O=C1CCC(=O)N1Br). The halamine contains an N-Br bond, which is reactive. The aliphatic methyl group (CH3) attached to the aromatic system undergoes bromination via radical substitution. The N-Br bond in the halamine cleaves homolytically, generating a bromine radical (Br·). This abstracts a hydrogen atom from the methyl group, forming a methyl radical (CH2·). The bromine radical then combines with the methyl radical, resulting in a bromomethyl group (CH2Br) and a new Br-C bond. The aromatic system remains intact, and the imide group (O=C1CCC(=O)N1) from the halamine loses Br to form a secondary amine (NH) as a byproduct.

- - - - - - - - - - - - - - - - - - - - - - - - - - - - - - - - - - - - - - - - - - - -

**Example 2 of Distilled Rationales with Minor Flaws:** Inappropriate naming

**Instruction:** To maximize the reaction yield, choose a candidate from the list Nc1ccc(-c2ccccc2)cn1, Cc1ccc(N)nc1, COc1ccc2nc(N)sc2c1, Nc1ncc(-c2ccccc2)cc1-c1ccccc1
Answer: that best completes the given reaction O=C(CBr)c1ccc(Br)cc1.O=C([O-])O.[Na+]>CCO>Unknown product, ensuring each substance is represented in SMILES notation and adhering to the standard format for chemical reaction equations.

**Rationale:**
Starting with the reaction details, we first note that the mechanism likely involves nucleophilic attack by an amine on an electrophilic center generated from an acyl bromide species. Since all options supply a primary amine functionality, attention turns to the aromatic framework that supports it. By examining the aromatic heteroatoms and substituent patterns, one assesses how each molecule's electronic properties might stabilize the intermediate and facilitate the reaction. In this case, the extended aromatic system and the presence of pyridinic nitrogen—with its electron-withdrawing nature—helpfully modulate reactivity and lower steric hindrance. Additionally, comparing specific substituents such as ether or additional alkyl groups, one finds that the option offering a conjugated biphenyl structure with optimized resonance stabilization is predicted to provide the highest yield by enhancing nucleophilicity while accommodating the transition state effectively. Given these considerations, the reasoning supports selecting the molecule with the most extended conjugated system featuring a primary amine attached to a pyridinic framework, as it best meets the criteria for high-yield synthesis under the reaction conditions.

Figure 12: Examples of the distilled rationales with minor flaws.

**Example 1 of Distilled Rationales with Substantive Issue:** Incorrect identification of the key components

**Instruction:** Can you please rank these molecules, given in SMILES format, according to their internal energy at 0 K from highest to lowest?
SMILES List:
[H]OC([H])([H])C([H])([H])[C@@]1([H])O[C@@]2([H])C([H])([H])[C@@]2([H])O1,
[H]C#CC([H])([H])C1[NH2+][C@@]2([H])C([H])([H])[C@@]2([H])O1,
[H]OC([H])([H])C1[NH2+][C@@]2([H])C([H])([H])C([H])([H])[C@@]2([H])O1,
[H]OC([H])([H])C([H])([H])C1[NH2+][C@@]2([H])C([H])([H])[C@@]2([H])O1
Order of internal energy at 0 K:

**Rationale:**
At 0 K, internal energy is determined by molecular potential energy, which depends on bond strength, ring strain, and functional group stability. The first SMILES contains a six-membered acetal ring (low ring strain) and hydroxy groups. The second SMILES has an alkyne (high bond energy due to sp-hybridized carbons) and a five-membered hemiaminal ether ring (moderate ring strain). The third SMILES includes a six-membered hemiaminal ether ring (lower strain than five-membered) with an alcohol group and a charged NH2+ group (higher electrostatic potential energy). The fourth SMILES contains a five-membered hemiaminal ether ring (higher strain than six-membered) with an alcohol group and NH2+ charge. Alkyne > charged groups > five-membered ring strain > six-membered ring stability dictates the ranking: third (six-membered charged),second (alkyne + five-membered),fourth (five-membered charged),first (six-membered uncharged).

- - - - - - - - - - - - - - - - - - - - - - - - - - - - - - - - - - - - - - - - - - - -

**Example 2 of Distilled Rationales with Substantive Issue:** Multiple logic errors

**Instruction:** For chemical analysis, select the molecule with the highest dipole moment by comparing the SMILES of the given options: [H]/N=C/NC(=O)NC(=O)C,[H]/N=C/NC(=O)C(C)(C)C,[H]/N=C(/C(=O)/C(=N/[H])N)N,[H]/N=C\C(=O)N/C=N/[H])N. Provide the SMILES of the chosen molecule.

**Rationale:**
To determine which molecule has the highest dipole moment, analyze the polarity and symmetry of the functional groups. Dipole moments increase with greater electronegativity differences and additive polar bond vectors. The first molecule contains urea (two amide groups) and imide (two carbonyl groups),which are polar, but symmetry may reduce the net dipole. The second molecule has an amidine and a single amide, but bulky tert-butyl substituents (C(C)(C)C) introduce steric hindrance, potentially reducing polarity. The third molecule features a ketone (C=O) and amidine, but conjugated resonance between the ketone and amidine groups (C(=O)/C(=N/[H])) may delocalize electron density, opposing dipole contributions. The fourth molecule includes a carboxamide (C(=O)N) and two amidine groups (N=C-N), arranged asymmetrically. The carboxamide's carbonyl (C=O) and N-H bonds create strong dipoles, while the amidine groups (with protonated N atoms and adjacent C=N) contribute additional polarity. The lack of symmetry in the fourth molecule allows these dipoles to align constructively, resulting in the highest net dipole moment.

Figure 13: Examples of the distilled rationales with substantive issues.

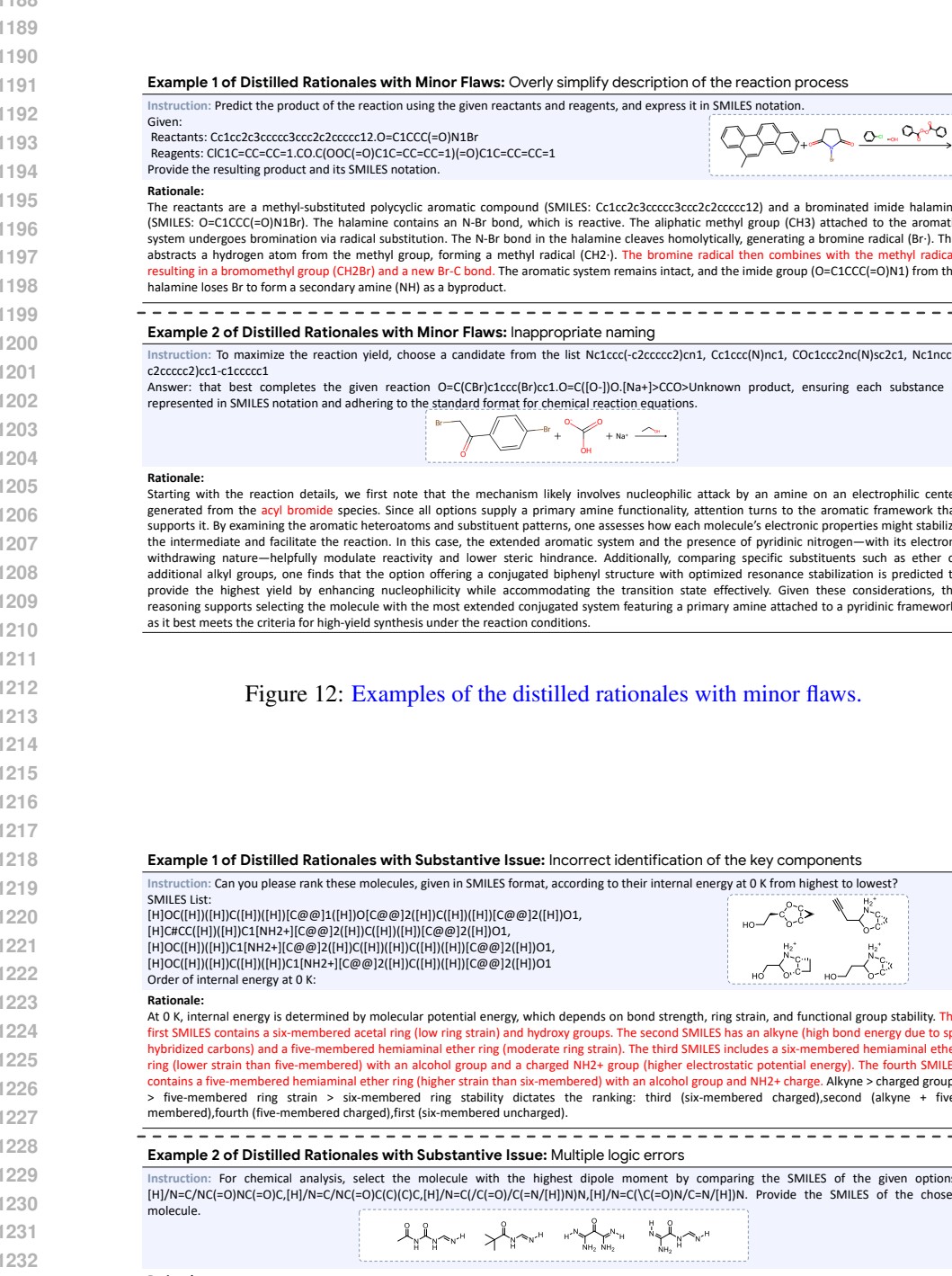

# E  TRAINING DETAILS OF CHEMDFM-R

## E.1  MITIGATING DATA LEAKAGE

Data leakage is a crucial problem for the evaluation of LLMs. During the training of ChemDFM-R, measures have been taken to reduce the risk of data leakage. For the domain pretraining corpus, we avoid using the same molecules and reactions as those presented in SciKnowEval (Feng et al., 2024) and ChemEval (Huang et al., 2024) benchmarks. For instruction tuning and mixed-source distillation, the same molecules and reactions are deliberately excluded from a task if they appear in relative tasks in SciKnowEval and ChemEval. We use canonicalized SMILES to determine whether two molecules are the same. Reactions are considered the same if they share the same product.

## E.2  STATISTICS ABOUT COMPUTATION AND DATA

We trained ChemDFM-R on NVIDIA A800 Tensor Core GPUs for a total of 30840 GPU hours. Specifically:

1). for domain pretraining, the model was trained on the 101-billion ChemFG corpus A for 24728 GPU hours;

2). for instruction tuning, the model was trained on over 7.5 million instructions B, which consist of 2.5 million chemical instructions and 5 million general instructions, for 3785 GPU hours;

3). for mixed-sourced distillation, the model was trained on the mixed-sourced distillation dataset, which is of the same scale as the instruction-tuning dataset, for 2059 GPU hours;

4). for reinforcement learning, the model was trained on 121,811 samples for 268 GPU hours.

## E.3  HYPERPARAMETERS AND SYSTEM PROMPTS

**Training Hyperparameters.**  The hyperparameters used during the training of ChemDFM-R are listed in Table 6.

**Inference Setting.**  During the inference of ChemDFM-R, we set the temperature to 0.6, topK to 1, and topP to 1 with no other penalties.

**System Prompts.**  During the training of ChemDFM-R, three different system prompts are used. Specifically:

- For all samples in the instruction-tuning dataset and the non-reasoning samples in the mixed-source distillation dataset, we use: "You are a helpful assistant."
- For the pseudo-reasoning data in the mixed-source distillation dataset, we use: "You are a helpful assistant that is good at answer chemical questions. You will analyze the presence of functional groups in molecules and the changes in functional groups during reactions before giving response. These analyses will help you solve the problem better. The analyses and answer are enclosed within <think> </think> and <answer> </answer> tags, respectively.\ni.e.,\n<think>\nanalyses here\n</think>\n<answer>\nanswer here\n</answer>"
- For the distilled data and the training of reinforcement learning, we use "You are a helpful assistant that is good at reasoning. You always reason thoroughly before giving a response. The reasoning process and answer are enclosed within <think> </think> and <answer> </answer> tags, respectively.\ni.e.,\n<think>\nreasoning process here\n</think>\n<answer>\nanswer here\n</answer>"

# F  THE ANALYSIS OF CHEMDFM-R'S RATIONALE

In Figure 14, ChemDFM-R is asked to pick a correct product for the given reaction. Instead of wasting time (tokens) on analyzing the SMILES in great detail, which DeepSeek-R1 always does,

Table 6: The training hyperparameters used during the training of ChemDFM-R.

|  | Domain Pretraining | Instruction Tuning | Mixed-source Distillation | Reinforcement Learning |
|---|---|---|---|---|
| Initial Learning Rate | 1e-5 | 1e-5 | 1e-5 | 5e-7 |
| Minimal Learning Rate | 1e-6 | 0 | 0 | 0 |
| Optimizer | Adam(0.9, 0.95) | | | |
| Scheduler | Cosine | | | |
| Max Sequence Length | 8192 | 8192 | 8192 | - |
| Max Generation Length | - | - | - | 1024 |
| Train Batch Size | 624 | 512 | 512 | 128 |
| Rollout Batch Size | - | - | - | 512 |
| Epochs | 1 | 2 | 1 | 1 |
| DAPO Group Size | - | - | - | 8 |
| DAPO Epsilon | - | - | - | (0.2, 0.3) |
| Initial KL Coefficient | - | - | - | 1e-3 |

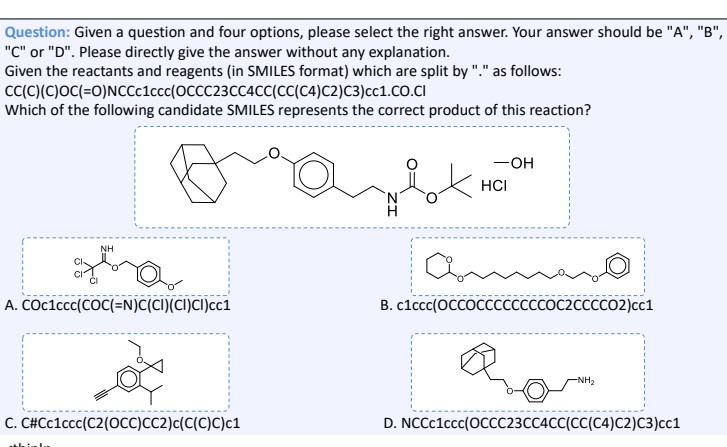

Figure 14: An example of the responses generated by ChemDFM-R on the reaction prediction task.

**Question:** You are an expert chemist. Given the molecule SMILES, your task is to provide the detailed description of the molecule using your experienced chemical knowledge.
Please strictly follow the format, no other information can be provided.

The Molecule SMILES is as follows:
CCCCCC/C=C/C(=O)SCCNC(=O)CCNC(=O)[C@H](O)C(C)(C)COP(=O)(O)OP(=O)(O)OC[C@H]1O[C@@H](N2C=NC3=C(N)N=CN=C32)[C@H](O)[C@@H]1OP(=O)(O)O

<think>
To derive the molecular description, one first examines the SMILES notation to identify the structural components such as the **alkene, hydroxy, ether, anhydride, primary amine, carboxamide, thiolester, and phosphate mono-ester groups**, as well as the aromatic heteroatoms like **pyrrolic N and pyridinic N**. Recognizing these functional groups and their connectivity, one then **correlates the structure with known biochemical molecules, particularly those involved in metabolic pathways**. By noting **the presence of a coenzyme A moiety and a deprotonated thiolester linkage, which are characteristic of acyl-CoA derivatives**, one can infer that the molecule is an acyl-CoA compound. This structural analysis, combined with knowledge of metabolic roles in organisms such as Escherichia coli and mouse, leads to the conclusion that **the molecule is a metabolite with a specific acyl group attached to coenzyme A**.
</think>
<answer>
The molecule is a medium-chain fatty acyl-CoA that results from the formal condensation of the thiol group of coenzyme A with the carboxy group of trans-2-nonenioic acid. It is a trans-2-enoyl-CoA, a medium-chain fatty acyl-CoA and a monounsaturated fatty acyl-CoA. It is a conjugate acid of a trans-2-nonenoyl-CoA(4-).
</answer>

Figure 15: The overview of the training pipeline of ChemDFM-R on the molecule captioning task.

**Question:** Please give the IUPAC name of the molecule given by SMILES: Cc1ccc(C(C)NC(=O)CCCN(c2cccc(Cl)c2)S(C)(=O)=O)c(C)c1.

<think>
To derive the IUPAC name, one first examines the SMILES notation to identify the molecular structure, noting the presence of **an aromatic ring substituted with methyl groups, a secondary amine linkage, and a sulfonamide group attached to a benzene ring bearing a chloro substituent**. The next step involves **determining the connectivity and substituent positions on the aromatic system, ensuring that the substituents are numbered according to IUPAC rules to give the lowest possible locants**. The molecule is then divided into **its principal structural components, with one part being the substituted benzene ring and the other part being the acyl chain linked to the sulfonamide group. The acyl chain is further analyzed to determine its substituent pattern, including the N-substitution with the sulfonamide group**. By applying systematic nomenclature rules, including the priority of functional groups and the use of prefixes and suffixes to denote substituents and connectivity, the complete IUPAC name is constructed. This process integrates the identification of functional groups, the determination of substituent positions, and the systematic assembly of the name according to IUPAC guidelines.
</think>
<answer>
4-(3-chloro-N-methylsulfonylanilino)-N-[1-(2,4-dimethylphenyl)ethyl]butanamide
</answer>

Figure 16: The overview of the training pipeline of ChemDFM-R on the SMILES to IUPAC task.

ChemDFM-R1 goes directly into the key point of this question: the functional groups present in the reactants and the potential reactions between them. Specifically, ChemDFM-R successfully identifies the key functional group, the carbamate ester group. By recalling that the carbamate ester group is typically used to protect amino groups, ChemDFM-R infers that the reaction taking place is likely a deprotection reaction. Then, ChemDFM-R confirms its assumption by examining the provided reagents. Finally, ChemDFM-R predicts the feature of the possible product and picks the option that matches it. This example demonstrates the precision of ChemDFM-R in finding the key point of chemical questions, and the effectiveness and efficiency of ChemDFM-R's rationales. Moreover, it is also worth noticing that, instead of using the "elimination-shortcut" which is commonly adopted by other cutting-edge reasoning LLMs, ChemDFM-R directly reasoned out the reaction mechanism and the features of the correct answer, thereby selecting the correct option.

As illustrated in Figure 15, when asked to describe a molecule given by SMILES, ChemDFM-R first analyzes the functional groups present in the molecule, such as the alkene group, the phosphate mono-ester group, and the pyrrolic N group. Then, ChemDFM-R successfully correlates the composition and connectivity of these functional groups with metabolic pathways and further manages to identify the molecule as a coenzyme A derivative. After that, it recognizes the deprotonated thiolester linkage in the molecule and further narrows down the molecule to an acyl-CoA derivative. Finally, ChemDFM-R gives a relatively comprehensive description of the molecule. ChemDFM-R even provides the potential role of this molecule in metabolic processes in its rationale, further demonstrating its strong reasoning ability as well as the value of the rationale as a complement to the final answer.

Figure 16 showcases an example response of ChemDFM-R when asked to generate the IUPAC name of the given molecule. The IUPAC name is the standard name for a molecule, assigned according to the rules established by the International Union of Pure and Applied Chemistry (IUPAC). It can effectively reflect the functional groups present in the molecule and their connectivity. Therefore, ChemDFM-R starts its reasoning with a comprehensive analysis of the functional groups of the given molecule. Then, it emphasizes the importance of correctly labeling the atoms, which is precisely an area where large language models are particularly prone to errors. After this, ChemDFM-R follows the rule of IUPAC naming and divides the molecule into its principal structural components. It also specifically points out the N-substitution with the sulfonamide group. Finally, a complicated and correct IUPAC name is predicted by ChemDFM-R.

## G   DETAIL RESULTS OF BENCHMARK EVALUATION

### G.1   CHEMEVAL

We consider the L1 and L2 level tasks in ChemEval (Huang et al., 2024) as text-centric tasks, the L3 level tasks as molecule-centric tasks, and the L4 level tasks as reaction-centric tasks. Moreover, there are tasks that we can not achieve a feasible grading in ChemEval. We temporarily skip these tasks. The raw results are demonstrated in Table 7.

As illustrated in Table 7, ChemDFM-R manages to achieve competitive performance in the text-centric tasks compared with the cutting-edge LLMs, while achieving SOTA performance across all the molecule-centric tasks and a large portion of the reaction-centric tasks. A detailed analysis of the task characteristics reveals that ChemDFM-R tends to perform less effectively on tasks involving numerical prediction, which will be a key focus of our future optimization efforts.

### G.2   SCIKNOWEVAL

We group the tasks in SciKnowEval (Feng et al., 2024) based on their input and output. Specifically, the task is classified as a text-centric task if there is no SMILES appear in its input or output, as a reaction-centric task if there are reaction SMILES appear in its input or output, and as a molecule-centric task otherwise. Due to budget limit, we currently skip the tasks that require GPT-4o for grading. The raw results are demonstrated in Table 8.

As illustrated in Table 8, ChemDFM-R achieves competitive performance on SciKnowEval compared to cutting-edge LLMs. It is worth noting that ChemDFM-R's performance advantage is less pronounced on SciKnowEval than on ChemEval. This is primarily because most tasks in SciKnowEval are formulated as multiple-choice questions, which substantially reduce the burden on the model's

| Tasks | Metric | MolInst | ChemLLM-20B-DPO | ChemDFM-13B-v1.0 | ChemDFM-8B-v1.5 | ether0 | GPT-4o | Qwen3-14B (no think) | DeekSeek-R1 | Qwen3-14B (think) | o4-mini | ChemDFM-R |
|---|---|---|---|---|---|---|---|---|---|---|---|---|
| *Text-Centric Tasks* | | | | | | | | | | | | |
| MCTask | Acc | 40.0 | 35.0 | 50.0 | 0.0 | 10.0 | 75.0 | 75.0 | **90.0** | 80.0 | 65.0 | 64.0 |
| FBTask | Acc | 0.0 | 0.0 | **5.0** | 0.0 | 0.0 | 0.0 | 0.0 | 0.0 | 0.0 | 0.0 | 1.0 |
| TFTask | Acc | 55.0 | 75.0 | 70.0 | 5.0 | 0.0 | 80.0 | 85.0 | **95.0** | 80.0 | 85.0 | 81.0 |
| SATask | BLEU | 0.6 | **17.3** | 12.2 | 10.6 | 0.4 | 15.0 | 9.3 | 4.3 | 9.2 | 10.3 | 11.2 |
| CNER | F1 | 0.0 | 44.1 | 59.0 | 22.8 | 19.2 | 66.7 | 60.9 | **69.6** | 67.1 | 57.1 | 63.1 |
| CERC | F1 | 4.5 | 0.0 | 12.3 | 0.0 | 0.0 | 20.3 | 24.1 | **37.7** | 20.2 | 25.5 | 24.9 |
| AddE | F1 | 34.2 | 58.7 | 46.7 | 5.0 | 0.0 | 70.0 | 65.0 | 70.8 | 65.0 | 68.3 | **73.3** |
| SolvE | F1 | 58.3 | 70.0 | **82.5** | 25.0 | 5.0 | 75.0 | 73.3 | 79.0 | 54.0 | 72.3 | 76.3 |
| TempE | F1 | 70.0 | 75.0 | 20.2 | 25.0 | 10.0 | 75.0 | **95.0** | **95.0** | 90.0 | **95.0** | **95.0** |
| TimeE | F1 | **95.0** | **95.0** | 40.0 | **95.0** | 5.0 | **95.0** | **95.0** | **95.0** | **95.0** | **95.0** | **95.0** |
| CharME | F1 | 20.7 | 67.0 | 26.2 | 49.1 | 16.8 | **79.0** | 73.8 | 71.0 | 60.7 | 76.0 | 61.6 |
| CatTE | F1 | 80.0 | 90.0 | 55.0 | 30.0 | 0.0 | 95.0 | 85.0 | **100.0** | 85.0 | 90.0 | 67.0 |
| YieldE | F1 | 0.0 | 55.0 | 20.0 | 30.0 | 20.0 | 80.0 | 75.0 | 75.0 | **95.0** | 75.0 | 57.0 |
| AbsGen | BLEU | 56.3 | 45.7 | 54.0 | 4.4 | 2.1 | 66.8 | **70.4** | 62.2 | 64.7 | 67.4 | 65.1 |
| OLGen | BLEU | 18.9 | 12.5 | 22.4 | 2.7 | 8.7 | 14.3 | **23.6** | 10.0 | 18.0 | 13.1 | 15.4 |
| TopC | Acc | 40.0 | 50.0 | 40.0 | 20.0 | 0.0 | 50.0 | 40.0 | **55.0** | 50.0 | 50.0 | 45.0 |
| ReactTR | F1 | 20.0 | 13.3 | 15.0 | 5.0 | 5.0 | **35.0** | 30.0 | 25.0 | 30.0 | 20.0 | 28.5 |
| *Molecule-Centric Tasks* | | | | | | | | | | | | |
| MolNG | BLEU | 3.6 | 45.5 | 75.9 | 34.6 | 46.7 | 59.9 | 43.1 | 9.9 | 41.7 | 68.6 | **83.3** |
| IUPAC2MF | EM | 0.0 | 0.0 | 25.0 | 0.0 | 0.0 | 40.0 | 0.0 | 5.0 | 0.0 | 60.0 | **73.0** |
| SMILES2MF | EM | 0.0 | 0.0 | 45.0 | 0.0 | 0.0 | 10.0 | 5.0 | 0.0 | 0.0 | 55.0 | **86.0** |
| IUPAC2SMILES | EM | 0.0 | 0.0 | 10.0 | 0.0 | 30.0 | 0.0 | 0.0 | 5.0 | 0.0 | 15.0 | **55.0** |
| SMILES2IUPAC | EM | 0.0 | 0.0 | 0.0 | 5.0 | 0.0 | 0.0 | 0.0 | 0.0 | 0.0 | 0.0 | **32.0** |
| S2S | EM | 0.0 | 0.0 | 0.0 | 0.0 | 0.0 | 0.0 | 0.0 | 0.0 | 0.0 | **5.0** | 0.0 |
| MolPC | Acc | 49.9 | 52.8 | 61.0 | 9.4 | 0.0 | 64.1 | 54.5 | 57.1 | 58.9 | 65.7 | **72.5** |
| Mol2PC | BLEU | 29.8 | 36.4 | 38.9 | 18.5 | 2.3 | 37.2 | 40.8 | 30.4 | 38.5 | 33.7 | **57.8** |
| *Reaction-Centric Tasks* | | | | | | | | | | | | |
| SubRec | Acc | 0.0 | 3.3 | 8.2 | 2.0 | 0.0 | 7.5 | 0.0 | 5.0 | 0.0 | 18.3 | **23.5** |
| LRec | Acc | 0.0 | 0.0 | 20.1 | 0.0 | 0.0 | 10.0 | 5.0 | 10.0 | 35.0 | 0.0 | **42.0** |
| RRec | Acc | 0.0 | 0.0 | 19.0 | 0.0 | 5.0 | 10.0 | 15.0 | 5.0 | 25.0 | 30.0 | **34.0** |
| SolvRec | Acc | 0.0 | 0.0 | 5.0 | 0.0 | 0.0 | **35.0** | 2.0 | 25.0 | **35.0** | 30.0 | 31.0 |
| TempRec | RMSE↓ | 76.5 | 27.1 | 71.5 | 22.2 | $\infty$ | 21.7 | 23.4 | **17.9** | 34.6 | 22.2 | 31.8 |
| TimeRec | RMSE↓ | 20.5 | 18.7 | 17.8 | 9.4 | $\infty$ | **6.6** | 8.0 | 9.5 | 10.7 | 9.6 | 18.8 |
| PPre | Acc | 0.0 | 0.0 | 3.3 | 5.0 | **73.3** | 10.0 | 0.0 | 18.2 | 5.8 | 35.0 | 60.0 |
| YPred | Acc | 20.0 | 20.0 | 20.0 | 0.0 | 0.0 | **55.0** | 14.9 | 25.0 | 25.0 | 45.0 | 23.0 |
| RatePred | Overlap | 3.0 | 13.8 | 4.7 | 9.0 | 0.0 | 18.2 | 14.9 | **31.0** | 9.9 | 7.2 | 14.1 |
| IMDer | Acc | 0.0 | 0.0 | **10.0** | 0.0 | 0.0 | 5.0 | 0.0 | 5.0 | 0.0 | 5.0 | 0.0 |

Table 7: The detailed benchmark results of different models on ChemEval (Huang et al., 2024).

| Tasks | MolInst | ChemLLM-20B-DPO | ChemDFM-13B-v1.0 | ChemDFM-8B-v1.5 | ether0 | GPT-4o | Qwen3-14B (no think) | DeekSeek-R1 | Qwen3-14B (think) | o4-mini | ChemDFM-R |
|---|---|---|---|---|---|---|---|---|---|---|---|
| *Text-Centric Tasks* | | | | | | | | | | | |
| Chem LiterQA | 75.4 | 78.7 | 77.6 | 79.3 | 57.3 | 85.8 | 84.2 | 86.3 | 84.0 | **88.9** | 80.2 |
| React Mech Infer. | 96.3 | 98.5 | 98.1 | 97.8 | 77.7 | **100.0** | 98.5 | 96.7 | 98.1 | 97.8 | 98.4 |
| Comp Iden. and Prop. | 93.6 | 97.4 | 95.2 | 96.8 | 72.8 | 98.6 | 98.2 | 96.4 | 98.6 | **99.0** | 97.3 |
| Chem DU | 95.2 | 98.2 | 97.4 | 96.5 | 74.4 | **99.5** | 98.6 | 98.6 | 98.9 | 99.4 | 97.9 |
| Chem HV | 82.8 | 88.8 | 85.9 | 88.3 | 0.9 | **94.9** | 93.6 | 93.2 | 92.1 | 93.4 | 92.1 |
| Chem RI | 92.8 | 95.9 | 94.6 | 95.5 | 60.0 | 96.9 | 97.1 | 95.3 | 96.3 | 96.3 | **97.2** |
| Balancing Eq. | 0.6 | 16.3 | 12.3 | 21.7 | 0.0 | 10.5 | 7.7 | 11.2 | 7.25 | 10.5 | **27.4** |
| Chem Cal. | 34.9 | 32.7 | 36.8 | 35.3 | 22.7 | 52.4 | 53.9 | 78.8 | 91.4 | **92.2** | 52.3 |
| Mol Tox. Pred | 53.7 | 54.3 | 44.3 | 45.2 | 3.9 | 37.4 | 51.8 | **62.4** | 51.3 | 55.1 | 43.0 |
| Chem Safe Test | 68.7 | 74.8 | 59.5 | 65.3 | 17.5 | **85.2** | 81.2 | 81.7 | 83.1 | 80.4 | 81.5 |
| *Molecule-Centric Tasks* | | | | | | | | | | | |
| Mol Name Conv. | 60.3 | 70.8 | 69.7 | 81.0 | 35.1 | 86.7 | 71.5 | 41.8 | 83.7 | **95.7** | 89.9 |
| Mol Prop. Pred | 31.8 | 33.9 | 38.1 | 48.0 | **48.9** | 44.0 | 37.9 | 41.2 | 41.6 | 48.1 | 44.6 |
| Mol Cap. | 14.4 | 21.0 | 22.7 | 22.5 | 0.9 | 13.9 | 8.6 | 12.6 | 10.0 | 14.0 | **50.9** |
| Mol Weight Cal. | 25.7 | 25.9 | 16.9 | 22.9 | 22.4 | 25.9 | 17.0 | 4.4 | 31.1 | **62.8** | 35.4 |
| Mol Prop. Cal. | 38.9 | 33.2 | 31.1 | 29.3 | 27.2 | 34.7 | 37.2 | 39.1 | 38.1 | **45.4** | 18.7 |
| Mol Stru. Pred | 33.1 | 30.9 | 35.6 | 25.9 | 31.3 | 46.1 | 30.9 | 7.5 | 36.6 | **57.4** | 31.7 |
| Mol Gen. | 64.4 | 52.9 | 84.4 | **89.4** | 70.2 | 60.9 | 38.6 | 69.3 | 44.5 | 75.4 | 86.6 |
| *Reaction-Centric Tasks* | | | | | | | | | | | |
| Reaction Pred | 39.2 | 92.8 | 91.6 | 95.9 | 55.8 | 48.0 | 90.5 | 82.5 | 93.7 | **99.3** | 98.4 |
| Retrosynthesis | 44.3 | 83.4 | 78.5 | 77.8 | 76.2 | 41.2 | 70.0 | 83.4 | 83.2 | **95.2** | 89.4 |

Table 8: The detailed benchmark results of different models on SciKnowEval (Feng et al., 2024).

comprehension and generation processes, allowing it to arrive at correct answers through "shortcuts" such as option comparison and elimination.

## G.3    ANALYSIS OF GENERALIZABILITY

To further demonstrate the generalizability of our model, we separate the tasks in SciKnowEval and ChemEval based on whether the same target appears in the training dataset of ChemDFM-R. This results in two categories: 1) Same Task Different Instruction (DI), demonstrating the instruction-level generalizability of our model, 2) Different Task (DT), demonstrating the task-level generalizability of our model. The evaluation results are presented in Table 10, from which we draw three key observations.

First, while enhancing the model's chemical capabilities, our training procedure largely preserves strong generalization performance. Compared with our baseline model, Qwen2.5-14B-Instruct, ChemDFM-R not only achieves substantially better performance on DI tasks but also improves on DT tasks, demonstrating that the knowledge acquired through training can be generalized to new tasks.

Second, our model performs significantly better on DI tasks than on DT tasks, indicating that it exhibits stronger instruction-level generalization than task-level generalization.

Finally, our model underperforms o4-mini on molecule-centric DT tasks in ChemEval. Upon further inspection, we find that among the three tasks in this category, only the "SMILES-to-SELFIES and SELFIES-to-SMILES conversion" tasks show inferior performance compared with o4-mini. This outcome mainly arises from the fact that we used only SMILES as the molecular representation during training. Considering that SMILES and SELFIES can be deterministically converted into each other through rule-based transformations, we believe that this weaker performance, or the lack of explicit SELFIES understanding and generation capability, does not materially affect the model's usefulness or effectiveness in practical chemistry applications.

## G.4    MORE EVALUATIONS ON FGBENCH AND CHEMCOTBENCH

In addition to ChemEval and SciKnowEval, we also evaluate ChemDFM-R along with our baseline models on two of the newly proposed benchmarks: FGBench (Liu et al., 2025b), which primarily focuses on evaluating LLMs' capability to determine the property changes after functional-group-level molecular modification, and ChemCOTBench (Li et al., 2025b), which comprehensively evaluates LLMs' capabilities in solving reasoning intensive chemical problems including molecular editing, molecular optimization, and reaction related. The experimental results are illustrated in Table 9, from which we draw three main observations:

First, ChemDFM-R outperforms both traditional chemistry LLMs and Qwen2.5-14B-Instruct. This demonstrates that our model possesses stronger chemical knowledge and more advanced chemical reasoning capabilities, enabling superior performance on reasoning-intensive tasks. This also validates the effectiveness of our ChemDFM-R training pipeline.

Second, compared with the more powerful Qwen3-14B, our model achieves better results only on ChemCOTBench. We believe this is due to the fundamental differences among the benchmarks: unlike ChemCOTBench and ChemEval, which primarily consist of open-ended generative tasks across multiple subfields, FGBench is composed exclusively of multiple-choice questions of property selection and numerical prediction problems of property prediction. For multiple-choice questions, as analyzed in the Appendix G.2, the required comprehension and generation ability is considerably lower than for free-form generation, allowing Qwen3 to circumvent its weaknesses and achieve strong results. For numerical prediction tasks, the performance gap aligns with our analysis in Section 4.1.1, further highlighting the limitations of ChemDFM-R on numerically intensive tasks and pointing toward directions for future improvement. Therefore, we believe that ChemDFM-R's weaker performance on FGBench relative to Qwen3 primarily reflects its remaining room for improvement on tasks related to numerical reasoning and prediction, whereas its superior performance on the broader ChemCOTBench highlights its advantage in overall chemical capability.

Finally, when compared with the 671B DeepSeek-R1 and the advanced closed-source model o4-mini, ChemDFM-R generally shows lower performance on both datasets. Given the substantial differences

Table 9: Benchmark results on FGBench and ChemCOTBench. "MUE" stands for "Molecule Understanding and Editing ", "MO" stands for "Molecule Optimization", and "CR" stands for "Chemical Reaction". We use the recommended metrics of each task, except the MUE tasks. Considering that the tasks in MUE include metrics where either larger or smaller values are preferred, we use the RPS Peng et al. (2025) to obtain a unified measure of average performance. The best performance for each task is indicated using **boldface**.

| Model | FGBench | | ChemCOTBench | | | | |
| | Boolean ACC ↑ | Value RMSE ↓ | MUE RPS ↑ | MO Δ ↑ | MO SR ↑ | CR ACC ↑ | CR FTS ↑ |
|---|---|---|---|---|---|---|---|
| MolInst | 21.2 | 218.7 | 53.2 | 0.020 | 39.2 | 0.3 | 8.3 |
| ChemLLM-20B-DPO | 45.3 | 140.6 | 46.3 | 0.022 | 12.0 | 6.0 | 0.5 |
| ChemDFM-13B-v1.0 | 13.7 | 162.9 | 38.2 | 0 | 0.8 | 1.8 | 0.5 |
| ChemDFM-8B-v1.5 | 22.8 | 216.0 | 22.6 | 0.036 | 10.5 | 0.5 | 0.1 |
| GPT-4o | 59.0 | **63.5** | 67.9 | 0.105 | 51.5 | 15.0 | 40.0 |
| Qwen2.5-14B-Instruct | 63.0 | 68.1 | 57.1 | 0.013 | 43.2 | 4.7 | 16.7 |
| Qwen3-14B (no think) | **76.1** | 85.7 | 59.0 | 0.078 | 34.5 | 8.2 | 35.1 |
| DeepSeek-R1 | 74.0 | 73.6 | 59.4 | 0.318 | 62.5 | 26.7 | 48.4 |
| Qwen3-14B (think) | 59.4 | 75.5 | 39.4 | 0.125 | 32.8 | 10.6 | 25.3 |
| o4-mini | 70.8 | 82.9 | **69.9** | **0.499** | **68.2** | **31.8** | 44.4 |
| ChemDFM-R | 65.0 | 90.1 | 69.1 | 0.184 | 48.7 | 24.4 | **49.2** |

Table 10: Benchmark results on SciKnowEval and ChemEval. "mol." stands for "molecule", "react." stands for "reaction", "DI" stands for "Different Instruction", and "DT" stands for "Different Task". The best performance for each task is indicated using **boldface**. * We use RPS (Peng et al., 2025) to balance the different scales of the scores on different tasks in the ChemEval benchmark.

| Model | SciKnowEval | | | | ChemEval* | | | | |
| | DI mol. | DT text | DT mol. | DT react. | DI mol. | DI react. | DT text | DT mol. | DT react. |
|---|---|---|---|---|---|---|---|---|---|
| MolInst | 39.4 | 69.4 | 38.0 | 41.7 | 10.4 | 10.0 | 47.3 | 42.7 | 8.6 |
| ChemLLM-20B-DPO | 37.0 | 73.6 | 38.9 | 88.1 | 20.9 | 28.4 | 64.5 | 49.7 | 14.8 |
| ChemDFM-13B-v1.0 | 53.6 | 70.2 | 38.3 | 85.0 | 44.9 | 13.0 | 61.5 | 50.4 | 39.2 |
| ChemDFM-8B-v1.5 | 55.9 | 72.2 | 41.4 | 86.9 | 10.1 | 38.1 | 26.7 | 19.1 | 13.3 |
| ether0 | 35.6 | 38.7 | 33.0 | 66.0 | 18.4 | 50.0 | 9.4 | 1.3 | 1.6 |
| GPT-4o | 37.4 | 76.1 | 47.5 | 44.6 | 33.8 | 42.2 | **82.7** | 52.3 | 55.5 |
| Qwen2.5-14B-Instruct | 23.9 | 77.2 | 40.5 | 71.9 | 17.3 | 50.0 | 77.5 | 53.8 | 41.2 |
| Qwen3-14B (no think) | 23.6 | 76.5 | 38.9 | 80.2 | 19.3 | 32.9 | 81.5 | 52.5 | 25.4 |
| DeepSeek-R1 | 40.9 | 80.1 | 26.8 | 83.0 | 14.0 | 55.3 | 81.2 | 50.1 | 46.0 |
| Qwen3-14B (think) | 27.3 | **86.6** | 46.2 | 88.5 | 19.2 | 26.1 | 79.0 | 52.8 | 44.4 |
| o4-mini | 44.7 | 81.3 | **61.9** | **97.3** | 51.7 | 58.5 | 78.4 | **86.4** | 52.4 |
| ChemDFM-R | **68.9** | 76.8 | 45.2 | 94.5 | **95.5** | **66.3** | 80.4 | 62.0 | **60.0** |

in model scale, we consider this partially expected and acceptable. In future work, we plan to extend our methods to larger LLMs to provide stronger chemical generalist reasoning LLMs for the open-source community.

# H    ABLATION STUDY FOR ATOMIZED CHEMICAL KNOWLEDGE ENHANCEMENT

Directly verifying the effect of atomized knowledge enhancement would be extremely costly, since it requires repeatedly performing computationally expensive domain pretraining. Therefore, instead

Table 11: Ablation study results on SciKnowEval and ChemEval. DP represents Domain Pretraining, and Know. represent Knowledge. The best performance for each task is indicated using **boldface**. * We use RPS (Peng et al., 2025) to balance the different scales of the scores on different tasks in the ChemEval benchmark.

| DP Corpus Composition | | SciKnowEval | | | | ChemEval* | | | |
|---|---|---|---|---|---|---|---|---|---|
| Atomized Know. | Text-based Know. | text | mol. | react. | all | text | mol. | react. | all |
| ✗ | ✗ | 66.9 | 30.0 | 34.5 | 49.9 | 44.6 | 25.5 | 25.4 | 34.8 |
| ✗ | ✔ | **67.1** | 30.6 | 36.5 | 50.4 | 43.6 | 25.6 | 28.5 | 35.2 |
| ✔ | ✗ | 65.7 | **31.8** | **37.8** | 50.3 | 51.9 | 25.9 | **33.8** | 40.8 |
| ✔ | ✔ | 66.2 | 31.7 | 37.4 | **50.5** | **53.7** | **26.9** | 31.0 | **41.1** |

of training a model of the same size as ChemDFM-R on the full dataset for comparison, we used Qwen2.5-1.5B as the base model and conducted the full training pipeline on a 10% subset of ChemDFM-R's data. By varying the data composition in the subset of the domain-pretraining corpus, we trained different versions of models for comparison while keeping the computational cost manageable. The results are presented in Table 11.

Compared with the model without any domain pretraining (Row 1), models pretrained on either the text-based-knowledge corpus (Row 2) or the atomized-knowledge corpus (Row 3) show improvements on most tasks. This demonstrates the importance and necessity of domain pretraining for strengthening domain knowledge, and indirectly supports our hypothesis that general domain LLMs generally possess insufficient advanced chemical knowledge. Furthermore, the model pretrained solely on the atomized-knowledge corpus outperforms the model pretrained on the traditional text-based-knowledge corpus on many tasks, despite its corpus being only 20% of the latter. This provides strong evidence that fine-grained atomized knowledge enables more efficient domain knowledge enhancement. Finally, the model pretrained on the combined text-based- and atomized-knowledge corpora (Row 4) achieves the best overall performance, reflecting the complementary strengths of the two corpora and validating the effectiveness of our atomized-knowledge–enhanced domain pretraining approach.

## I  DETAILS OF THE HUMAN ASSESSMENTS OF RATIONALE QUALITY

We hired five graduate-level chemical experts to assess the interactions across the following five dimensions, each of which is scored on a 5-point scale:

- Chemical Correctness: The correctness of chemical knowledge and logic demonstrated throughout the reasoning process.
- Answer Accuracy: Whether the final answer is correct.
- Analytical Coverage: The extent to which different plausible possibilities are explored during reasoning.
- Reasoning Coherence: Whether the reasoning remains focused, coherent, and aligned with the problem.
- Effective Information Density: The density of useful information in the reasoning chain, reflecting the friendliness and efficiency of interaction.

The ten original questions we used are listed as follows.

**Organic Chemistry:**

- (Yao et al., 2025) I have used m-CPBA to convert the carbon-carbon double bond within the [H][C@]12CC[C@@]3(CC[C@]1C)C(=O)C1=C[C@@]4(C)CC[C@@](C(C)C)[C@]4([H])C[C@]1([H])[C@]23C into an epoxide, and obtained chiral epoxy products with different ratios (d.r. = 5:1). Please propose possible reasons.
- (Zhou et al., 2025) Clc1ccccc1 is difficult to react with [O-]C(F)(F)F under normal conditions, but it can be converted into a free radical cation under photocatalytic conditions and

can react with [O-]C(F)(F)F in the presence of [Ag+] to obtain the product in high yield. Please provide the structure of the product.

**Inorganic Chemistry:**

- (Schwarzmann et al., 2025) C[Bi+2]([O]1CCCC1)([O]1CCCC1)([O]1CCCC1)([O]1CCCC1)[O]1CCCC1 is a newly reported strong Lewis acid. Please provide the oxidation state, ligand, and coordination number of the metal ion in C[Bi+2]([O]1CCCC1)([O]1CCCC1)([O]1CCCC1)([O]1CCCC1)[O]1CCCC1, and explain the reason why it has strong Lewis acidity.

- (Mandai et al., 2025) Fc1c(F)c([B-](c2c(F)c(F)c(B3Oc4ccccc4O3)c(F)c2F)(c2c(F)c(F)c(B3Oc4ccccc4O3)c(F)c2F)c2c(F)c(F)c(B3Oc4ccccc4O3)c(F)c2F)c(F)c(F)c1B1Oc2ccccc2O1 is a newly reported stable Lewis acidic anion, which breaks the previous understanding that anions are incompatible with Lewis acids. Please analyze its structure, explain why it can act as a Lewis acid, and indicate its binding sites with Lewis bases.

**Materials Chemistry:**

- (Li et al., 2025a) What is oxygen evolution reaction (OER)? Please propose a reasonable mechanism of heterogeneous OER under acidic conditions.

- (Liu et al., 2025a) C1=Cc2cc3ccc(cc4nc(cc5ccc(cc1n2)[nH]5)C=C4)[nH]3 and c1ccc2nsnc2c1 can form covalent organic frameworks through covalent bonding under certain conditions, which can utilize the excitation energy of singlet and triplet states for photocatalysis. Please determine which is the electron donor and which is the electron acceptor during the electron transfer process through the structural analysis of C1=Cc2cc3ccc(cc4nc(cc5ccc(cc1n2)[nH]5)C=C4)[nH]3 and c1ccc2nsnc2c1.

**Analytical Chemistry:**

- (Wu et al., 2025) CC(C)(C)c1cc2cc(C(C)(C)C)cc3c4cc(C(C)(C)C)cc5cc(C(C)(C)C)cc(c(c1)c23)c54 is a fluorescent material. Please explain the reason why it can emit light from the perspective of molecular structure.

- (Guo et al., 2025b) c1cc(-c2ccc3cc4cc(-c5ccncc5)ccc4cc3c2)ccn1 and N#Cc1cc(C#N)c(C#N)cc1C#N can be co assembled into a eutectic and emit orange light under photoluminescence. After the addition of O=C(O)C(F)(F)F, the eutectic will undergo a transformation, and the luminescence will change from orange light to yellow light. Please explain the reason.

**Polymer Chemistry:**

- (Zhang et al., 2025a) O=S(=O)(Oc1nc(=Cc2ccco2)c(OS(=O)(=O)C(F)(F)F)nc1=Cc1ccco1)C(F)(F)F and O=S(=O)(Oc1nc(=Cc2ccccc2)c(OS(=O)(=O)C(F)(F)F)nc1=Cc1ccccc1)C(F)(F)F are monomers of two polymers that form polymers under solid-state heating conditions. In addition, it was found that O=S(=O)(Oc1nc(=Cc2ccco2)c(OS(=O)(=O)C(F)(F)F)nc1=Cc1ccco1)C(F)(F)F is relatively more prone to dimerization when heated in the solution. Please infer the reason by comparing their structures.

- (Rugh et al., 2025) Under certain conditions, C[Si]1(C)OCCOCCO1 can undergo anionic ring opening polymerization to obtain polymers, which exhibit extremely high ionic conductivity in lithium battery electrolytes, surpassing poly (ethylene oxide). Please analyze the reasons from a structural perspective.

## J    MORE EXAMPLES OF RELIABLE HUMAN-AI COLLABORATION

Figure 17 illustrates a conversation starting from the same turn as that illustrated in the main text. In this conversation, we focus on fully understanding the mechanism of the oxygen evolution

**Turn1:** What is oxygen evolution reaction (OER)? Please propose a reasonable mechanism of heterogeneous OER under acidic conditions.

<think>
To derive the answer, one starts by defining the oxygen evolution reaction (OER) as the electrochemical process that converts water into molecular oxygen, which is a critical step in water splitting. Recognizing that OER can occur under both acidic and basic conditions, the focus is on the mechanism under acidic conditions, where the reaction involves protons (H+) and water molecules. The process is typically facilitated by a catalyst, often a transition metal oxide, which provides active sites for the reaction. The mechanism is then broken down into several steps: first, the adsorption of water molecules onto the catalyst surface, followed by the formation of an intermediate species such as an adsorbed hydroxyl group (OH−) or a water-derived species. Subsequent steps involve the transfer of protons and electrons, leading to the formation of an oxygen-oxygen bond, which is a key step in the reaction. Finally, the desorption of molecular oxygen (O2) from the catalyst completes the cycle. This sequence of adsorption, intermediate formation, bond formation, and desorption is consistent with the known electrochemical steps in OER under acidic conditions, and it is supported by experimental observations and theoretical models.
</think>
<answer>
The oxygen evolution reaction (OER) is an electrochemical process that involves the conversion of water into molecular oxygen (O2) and protons (H+). In heterogeneous OER under acidic conditions, the reaction typically occurs on a solid catalyst surface. The mechanism can be described as follows:

1. **Water adsorption**: Water molecules (H2O) adsorb onto the catalyst surface.
2. **Proton transfer**: Protons (H+) are transferred from the adsorbed water molecules to the catalyst surface, forming hydroxyl (OH-) groups.
3. **Oxygen adsorption**: The hydroxyl groups further react to form adsorbed oxygen species (O*), which are intermediates in the reaction.
4. **Oxygen evolution**: The adsorbed oxygen species combine to form molecular oxygen (O2), which desorbs from the catalyst surface and enters the solution.

The overall reaction can be represented as:

2H2O → O2 + 4H+ + 4e-
</answer>

**Turn2:** Since you mentioned in your thinking that the formation of oxygen-oxygen bond is the key to the reaction, please give an intermediate that reflect the formation of oxygen-oxygen bond and improve the mechanism you provided.

<think>
To incorporate the formation of the oxygen-oxygen bond into the mechanism, one starts by recognizing that this bond formation is a critical step in the oxygen evolution reaction (OER). The process involves the generation of an adsorbed oxygen species (O) on the catalyst surface, which then undergoes a transformation to form an intermediate that contains the oxygen-oxygen bond. This intermediate is typically represented as a peroxide-like species (OOH) or a superoxide-like species (O2−), depending on the specific reaction conditions and the electronic properties of the catalyst. The formation of this bond is facilitated by the transfer of protons and electrons, which are provided by the acidic environment and the catalyst, respectively. The stabilization of the intermediate is crucial for the subsequent steps, including the final desorption of molecular oxygen (O2) from the catalyst surface. By integrating these specific intermediates and their roles in the mechanism, one can clearly illustrate how the oxygen-oxygen bond is formed and how it contributes to the overall OER process under acidic conditions.
</think>
<answer>
In the mechanism of the oxygen evolution reaction (OER) under acidic conditions, the formation of the oxygen-oxygen bond is a critical step. This bond formation occurs through the generation of an intermediate species that contains two oxygen atoms. The specific intermediate formed depends on the reaction conditions and the catalyst used.

One possible intermediate is the peroxide species (OOH), which is formed when two hydroxyl groups (OH-) undergo a coupling reaction. This coupling reaction involves the transfer of an electron and a proton, leading to the formation of the peroxide bond (O-O).

Another possible intermediate is the superoxide species (O2−), which is formed when an oxygen atom (O) undergoes a one-electron reduction. This reduction can occur through the transfer of an electron from the catalyst surface or from another oxygen species.

The formation of these intermediates is essential for the subsequent steps in the OER mechanism, including the final desorption of molecular oxygen (O2) from the catalyst surface. The stability and reactivity of these intermediates are influenced by factors such as the electronic properties of the catalyst and the reaction conditions (e.g., pH, temperature).
</answer>

Figure 17: An example of reliable human-AI collaboration using ChemDFM-R. We draw inspiration from this conversation from Li et al. (2025a). This example illustrates a process of error detection and correction with the help of rationales.

**Turn1:** I have used m-CPBA to convert the carbon-carbon double bond within the [H][C@]12CC[C@@]3(CC[C@]1C)C(=O)C1=C[C@@]4(C)CC[C@@](C(C)C)[C@]4([H])C[C@]1([H])[C@]23C into an epoxide, and obtained chiral epoxy products with different ratios (d.r. = 5:1). Please propose possible reasons.

<think>
The reaction involves m-CPBA (meta-chloroperbenzoic acid) oxidizing a carbon–carbon double bond in a complex cyclic structure to form an epoxide. m-CPBA acts as an electrophilic oxidizing agent, where the peracid's oxygen attacks the double bond, forming a cyclic transition state. The stereochemical outcome is influenced by the complex cyclic framework's rigidity and steric factors. The double bond's geometry and the spatial arrangement of substituents around the double bond determine which face of the alkene is more accessible to the electrophilic oxygen from m-CPBA. The complex cyclic system imposes constraints that favor one diastereomeric pathway over another, leading to a 5:1 diastereomeric ratio. The chiral centers in the complex cyclic structure further influence the transition state energies, with one pathway being thermodynamically favored due to reduced steric strain or optimal orbital overlap during the epoxidation process.
</think>
<answer>
The complex cyclic framework of the molecule and the steric factors around the double bond lead to a preferential attack by the oxidant on one face of the double bond, resulting in a diastereomeric ratio of 5:1.
</answer>

**Turn2:** Please explain in detail how double bond's geometry and spatial arrangement of substituents around the double bond determine which face of the alkene is more accessible.

<think>
The spatial arrangement of substituents around the double bond in the complex cyclic structure creates steric hindrance, which influences the accessibility of the alkene face to the oxidizing agent. The complex cyclic framework imposes rigidity, restricting the approach of m-CPBA to one face of the double bond. Substituents such as methyl group create steric bulk, favoring the less hindered transition state during epoxidation. Additionally, electronic factors, including the electron-donating or electron-withdrawing nature of substituents, further modulate the reactivity of the alkene face. The combination of steric and electronic effects results in a preferential attack on the less congested face, leading to the observed diastereomeric ratio of 5:1 in the epoxide products.
</think>
<answer>
The complex cyclic structure imposes steric constraints that influence the accessibility of the double bond to the oxidizing agent. The arrangement of substituents around the double bond creates a situation where one face is more sterically accessible than the other, leading to a preference for the oxidizing agent to attack from that side, which is reflected in the diastereomeric ratio of the resulting epoxide products.
</answer>

Figure 18: An example of reliable human-AI collaboration using ChemDFM-R. We draw inspiration from this conversation from Yao et al. (2025). This example illustrates a process of information completion.

reaction (OER). Suppose, as a newbie, we are unable to determine the correctness of the answer. With the help of ChemDFM-R's rationale, we could easily discover that the key step of the reaction mentioned in the rationale, which is "the formation of an oxygen-oxygen bond", is absent in the answer. This could serve as a reminder that the answer could be incorrect, and drive us to further request the model to clarify this inconsistency. After this follow-up inquiry, the model provided a better answer.

Figure 18 is a conversation about a reaction proposed in Yao et al. (2025). We first ask ChemDFM-R to explain the reason for the different ratios of the chiral epoxy products. Although the model's answer is relatively broad, ChemDFM-R thoroughly analyzes the influence factors in its rationale, including the bond's geometry and the spatial arrangement of substituents around the double bond, which is not included in the answer. With this information, we can further pursue the follow-up question and obtain an improved answer.

## K    LIMITATION AND FUTURE WORK

In this work, we developed a chemical generalist reasoning model using atomized chemical knowledge enhancement and mixed-source distillation-based chemical rationale learning. Through both benchmark evaluations and human assessments, we demonstrated the strong potential of ChemDFM-R in solving chemical problems and supporting human–AI collaboration. The evaluations also revealed several limitations of our model, which can be summarized in three aspects.

First, as shown in Section 4.1.1, the chemical rationale learning phase substantially enhances performance on molecule- and reaction-centric tasks but weakens general language abilities, especially numerical prediction abilities. This is mainly because our RL stage relies on rule-based methods to calculate rewards, which restricts the types of tasks we can use. Designing reward mechanisms better suited to chemical contexts and allowing a broader range of RL tasks will be an important direction for the improvement of ChemDFM-R.

Second, results in Table 5 indicate that although our model produces concise and informative reasoning chains, which greatly improve its usability and friendliness in human–AI interactions, this conciseness can compromise analytical coverage. Achieving a better balance between maintaining concise reasoning chains and providing sufficiently diverse analyses could further enhance ChemDFM-R's practical value in interactive settings.

Finally, similar to general reasoning models, ChemDFM-R occasionally exhibits inconsistencies between its reasoning process and final answers. In chemistry, this includes both mismatches between reasoning steps and conclusions, and the use of incorrect chemical knowledge within the reasoning. Enhancing reward design to enforce consistency and incorporating stronger chemical supervision will be another key area for future improvement.

