# OpenReview forum: "ChemDFM-R: A Chemical Reasoning LLM Enhanced with Atomized Chemical Knowledge"
_ICLR.cc/2026/Conference — Submitted to ICLR 2026_

### Official Review · Reviewer_rgJV · 2025-10-26

**Soundness:** 3
**Presentation:** 3
**Contribution:** 2
**Rating:** 4
**Confidence:** 3

**Summary:**

This paper presents ChemDFM-R, a large language model for chemical reasoning that integrates atomized chemical knowledge with reasoning ability. The authors first build ChemFG, a 101B-token corpus that explicitly encodes functional–group–level information in molecules and reactions based on their developed toolkit for functional group identification for domain pretraining. Then they propose a mix-sourced distillation method combining chemical knowledge, pseudo-reasoning data, and rationales from general reasoning LLMs (Deepseek-R1, o3-mini), followed by domain-specific reinforcement learning to enhance interpretability and reasoning depth. Experiments on SciKnowEval and ChemEval show that ChemDFM-R significantly outperforms both general and chemical-domain baselines, producing accurate and transparent rationales that support reliable human-AI collaboration in chemistry.

**Strengths:**

- The paper demonstrates substantial technical and engineering effort, presenting a complete and well-executed pipeline that includes data construction, model design, and extensive evaluation.

- The idea of introducing atomized chemical knowledge from the perspective of functional groups is conceptually sound and offers a novel, fine-grained lens for enabling chemical reasoning.

- The proposed toolkit, ChemFG corpus, and the trained ChemDFM-R model are valuable community resources that can support future research in chemistry-focused LLMs and reasoning modeling.

- The model shows solid empirical performance, achieving strong results across multiple chemical benchmarks and demonstrating clear improvements over both domain-specific and general LLM baselines. Human studies are also provided for necessary experiments as justification.

**Weaknesses:**

- From a pure research perspective, while this work represents great engineering efforts, its research value may be limited. In my opinion, the research value of this work lies in the integration of functional groups as atomized chemical knowledge. There are not many insights or conclusions brought beyond that.
- I question the relevance of the inclusion of Figure 6. I understand that this section is used to support the claim of "human-AI collaboration". But I think this is a little hand-wavy by only presenting a case study. More statistical comparison against baselines would be more helpful. Otherwise, I would suggest removing this section to the Appendix and making the claim weaker in the paper.
- I think some experiments and additional analyses would further complement the paper. For example, the authors can consider running experiments on FGBench [1], which specifically targets functional group reasoning. Additionally, the claim made in lines 90-92 needs better support. I do not see from the current experiment results how the decomposed reasoning capability is improved. Experiment results for o3-mini would be beneficial since training data are distilled from it.

[1] Liu, Xuan, et al. "FGBench: A Dataset and Benchmark for Molecular Property Reasoning at Functional Group-Level in Large Language Models." NeurIPS 2025 Datasets & Benchmark Track.

**Questions:**

- More details of the human annotation on quality control should be stated. For example, on what level are the three expert chemists, undergrads, grads, or researchers in the chemistry field? The exact accuracy could be clearly presented, with examples of error cases.
- Any intuition or reasons for choosing 70%, 22%, and 8% for data distribution in mix-sourced distillation?
- In Table 2, why is the performance improvement brought by reinforcement learning (ChemDFM-R) far less on SciKnowEval than ChemEval?

---

> ### Author Response · Authors · 2025-11-25
> **Part [1/6] Author Response**
>
> Thank you for your insightful questions and constructive feedback. We have carefully revised our manuscript and provided detailed responses to each of your comments below. We hope that these replies address your concerns and enhance the quality of our work. We sincerely welcome any further discussions or suggestions.
>
> ## W1. About research value
>
> Thank you for raising this meaningful question. Research value is indeed an aspect we take very seriously when conducting this work. While we appreciate your recognition of the value brought by leveraging functional-group information, we respectfully believe that the contribution of our work goes beyond this single dimension. In our view, the research value of our study lies in the following three aspects.
>
> ### **Methodological Contributions**
>
> Our methodological contributions center on two key innovations.
>
> 1. **Introducing functional groups as atomized knowledge points to enable efficient and fine-grained learning of chemical knowledge.** We appreciate your acknowledgement of this contribution, and therefore do not repeat the discussion here.
> 2. **Proposing mixed-source distillation that integrates multiple data sources to substantially reduce the dependency on large-scale, high-quality reasoning distillation data, thus enabling effective distillation under limited resources.** In our humble opinion, the fact that this approach may seem simple does not mean that it is easy to think of. Although models like Qwen3 also use both reasoning and non-reasoning data, their goal is usually to unify two inference modes. Traditionally, when facing a shortage of distillation data, there are two common strategies. 1) Collecting more real distillation data, which is extremely costly, especially under constrained resources. 2) Generating pseudo-reasoning data, which suffers from a trade-off between diversity and cost: rule-based generation lacks diversity and hurts generalization ability, whereas LLM-based rewriting requires substantial additional resources. **Our work offers a new perspective: simply co-training with non-reasoning data can considerably enhance reasoning performance, reducing dependence on expensive real distillation data.** As mentioned, while the method *may seem simple*, it is *not easy to think of in the first place*.
>
> ### **Engineering Contributions**
>
> Although methodological contributions are often the most recognizable types of research value, we believe that the significance of a study is never limited to methodology alone. In our humble opinion, the engineering efforts, which were crucial for the proposed methods to actually function in practice, are themselves an essential component of research value. **We do not consider engineering contributions to be secondary to methodological ones.**
>
> During the development of this project, we encountered substantial engineering challenges, and addressing them constituted an important part of our contribution. Below, we highlight three of the most prominent ones:
>
> 1. **Construction of a Functional Group Identification Toolkit**
>
>     Functional-group detection has historically received limited attention in the community, and, to the best of our knowledge, no existing tool can identify a sufficiently broad range of functional groups with both high accuracy and high recall. Consequently, we invested substantial effort in building our own toolkit based on RDKit and SMARTS patterns.
>
>     Our final implementation supports 241 functional groups while accounting for numerous overlapping, nested, and ambiguous chemical environments. For example, as shown in Table R1, **even a simple carbonyl group can imply different chemical properties with different naming when bonded with different types of atoms.**
>
>     **Table R1.** Different naming of carbonyl groups (O=C)
>
>     | Name | carbonyl | aldehyde | carboxylic acid | ester | carbonate ester | carbamate ester | S-organyl thiocarbamates | ... |
>     | :---| :---: | :---: | :---: | :---: | :---: | :---: | :---: |:---: |
>     | Attached atom 1 | C | C | C | C | O which attaches to C | N | N | ... |
>     | Attached atom 2 | C | H | OH | O which attaches to C | O which attaches to C | O which attaches to C | S which attactes to C | ... |
>
>     Ultimately, our toolkit achieves 98% accuracy for molecule-level annotation and 89% accuracy for reaction-level annotation, enabling fine-grained, reliable supervision for model training.

---

> ### Author Response · Authors · 2025-11-25
> **Part [2/6] Author Response**
>
> 2. **Construction of a Highly Diverse Instruction-Tuning Dataset**
>
>     Many prior LLM efforts in the chemical domain tend to produce multi-task specialists rather than true generalists. One of the major reasons lies in the lack of task diversity in the chemical domain. To mitigate this limitation, we curated an instruction-tuning dataset spanning 16 categories of tasks (details in Appendix B.1). Specifically, these tasks can be categorized into the following 4 groups.
>
>     * **Popular tasks in Chemical LLM training (7)**: Name Translation, Description Generation, Molecule Design, Property Classification, Property Regression, Reaction Completion, and Exam Questions.
>     * **Tasks mimicking general NLP tasks (3)**: Literature QA, Literature Summarization, and Literature Translation.
>     * **Tasks explored by traditional AI for Chemistry but not widely incorporated into Chemical LLM training (4)**: Step Prediction, Yield Prediction, Temperature Prediction, and Reaction Component Selection.
>     * **Tasks proposed by us (2)**: Property Ordering and Property Selection.
>
>     Except for the first category, no sufficiently large datasets existed for the remaining tasks. Therefore, we spent extensive effort on data collection, cleaning, processing, and filtering. This dataset plays a key role in enabling our model to generalize across a broad spectrum of chemical scenarios.
>
> 3. **Large-Scale Instruction Rewriting for Generalization**
>
>     To further avoid overfitting to specific task patterns and to preserve generalization capabilities, we performed **large-scale instruction rewriting for all tasks in our instruction-tuning dataset**. As a result, each task contains a number of distinct instruction templates amounting to **more than 1/50 of the total data entries** in that task. The detailed rewriting pipeline is demonstrated in Appendix B.2. To keep this already lengthy response focused, we will not go into detail here.
>
> ### **Application Value**
>
> Beyond methodological and engineering contributions, we believe that the application value of the research outcomes is also an essential dimension of research value. In this regard, our work provides the chemistry community with a new model tool for chemical intelligence. The application value of our model lies in two main aspects:
>
> (1) **A more verifiable and interaction-friendly assistant for chemists.**
> Our model offers concise and informative reasoning that enables users to validate answers, identify potential errors, and uncover further analytical directions during free-form dialogue. Compared to non-reasoning chemical models, our model significantly boosts the verifiability of interactions by providing interpretable reasoning. At the same time, our model avoids the opposite problem seen in general reasoning LLMs, whose reasoning chains tend to be excessively long, hard to interact with, and often chemically inaccurate due to insufficient domain knowledge. While not perfect, it represents a more dependable tool for assisting chemical research.
>
> (2) **A foundation for future chemical LLM improvements and agent systems development.**
> Our model also serves as a basis upon which future chemical LLMs and chemical agent systems can be further developed and optimized. We will release the resulting models, including ChemDFM-R and ChemDFM-I, as well as our SFT dataset, mixed-source distillation dataset, RL dataset, and the functional group identification toolkit upon publication to contribute to the open-source community.
>
> ---
>
> **These points summarize our view on the research value of our work. We hope our statements address your concerns, and we sincerely welcome any further discussion.**

---

> ### Author Response · Authors · 2025-11-25
> **Part [3/6] Author Response**
>
> ## W2. Statistical comparison regarding human-AI collaboration
>
> Thank you for your valuable advice. We agree that using only three cases (two of them are in Appendix J) to support the claim of "human-AI collaboration" may appear hand-wavy. To more rigorously evaluate the performance of our model during interactions, we follow your advice and **perform statistical comparisons to the extent permitted by our resources**.
>
> Specifically, we constructed ten graduate-level questions based on recent publications (within the past two years) from several influential chemistry journals. The questions cover different major subfields of chemistry, including organic chemistry, inorganic chemistry, materials chemistry, analytical chemistry, and polymer chemistry. Then, different reasoning LLMs are leveraged to solve these questions through human-AI interactions. Five graduate-level chemistry experts were hired to evaluate these interactions across the following five dimensions (each scored on a 5-point scale) to comprehensively assess the strengths and limitations of our model under scientific interactive situations:
>
> * **Chemical Correctness**: The correctness of chemical knowledge and logic demonstrated throughout the reasoning process.
> * **Answer Accuracy**: Whether the final answer is correct.
> * **Analytical Coverage**: The extent to which different plausible possibilities are explored during reasoning.
> * **Reasoning Coherence**: Whether the reasoning remains focused, coherent, and aligned with the problem.
> * **Effective Information Density**: The density of useful information in the reasoning chain, reflecting the friendliness and efficiency of interaction.
>
> The results are summarized in Table R2, from which we draw three main conclusions.
>
> **Table R2.** Results of human evaluation for human-AI interactions. As the rationales of o4-mini are not accessible, we only use the answer provided by o4-mini.
>
> | Models | Chemical Correctness | Answer Accuracy | Analytical Coverage | Reasoning Coherence | Effective Information Density |
> |:---|:---:|:---:|:---:|:---:|:---:|
> | ether0 | 1.56 | 1.60 | 1.40 | 1.76 | 1.56 |
> | DeepSeek-R1 | 4.10 | 3.88 | **4.34** | 4.42 | 3.62 |
> | Qwen3-14B (think) | 3.56 | 3.92 | 4.24 | 3.98 | 3.00 |
> | o4-mini | 3.94 | 3.96 | 3.84 | 4.16 | 4.12 |
> | ChemDFM-R | **4.52** | **4.28** | 3.94 | **4.62** | **4.70** |
>
> First, **our model outperforms all baselines, including DeepSeek-R1, in both chemical correctness and answer accuracy**, indicating that it possesses a stronger grasp of chemical knowledge. In particular, our model shows a clearer advantage on SMILES-related tasks, demonstrating a more precise understanding of molecular structures.
>
> Second, **our model achieves significantly higher scores in effective information density**. Unlike models such as Qwen3-14B and DeepSeek-R1, which often generate extremely lengthy reasoning chains exceeding 1,000 tokens, our model produces concise yet informative reasoning. This considerably reduces the burden on human users when verifying correctness or identifying errors.
>
> Finally, **our model performs slightly lower than Qwen3-14B and DeepSeek-R1 on analytical coverage**. This is partially because their extremely lengthy reasoning chains allow them to enumerate many possible considerations. It is also true that our model commonly focuses on one or a few major possibilities rather than adequately enumerating all potential factors. This highlights a productive direction for future enhancement of our model’s analytical breadth.
>
> We have added these results and analyses to Section 4.3.1. The ten original questions used for this evaluation are documented in Appendix I.

---

> ### Author Response · Authors · 2025-11-25
> **Part [4/6] Author Response**
>
> ## W3. More experments and analyses
>
> Thank you for your valuable advice. This comment contains three main questions/suggestions, and we respond to each of them as follows:
>
> 1. **More evaluation:**
>
>     Following your suggestion, we conducted additional evaluations on **FGBench[1], which primarily focuses on evaluating LLMs' capability to determine the property changes after functional-group-level molecular modification**, and **ChemCOTBench[2], which comprehensively evaluates LLMs' capabilities in solving reasoning-intensive chemical problems** including molecular editing, molecular optimization, and reaction related tasks. The updated results and analyses have been incorporated into the manuscript as Appendix G.4 and are also transcribed below as Table R1.
>
>     **Table R1.** Results of new experiments on FGBench and ChemCOTBench
>
>     | Model | FGBench || ChemCOTBench |||||
>     |:---|:---:|:---:|:---:|:---:|:---:|:---:|:---:|
>     || Boolean-ACC ↑ | RMSE ↓ | MUE-RPS ↑ | MO-Δ ↑ | MO-SR ↑ | CR-ACC ↑ | CR-FTS ↑ |
>     | MolInst | 21.2 | 218.7 | 53.2 | 0.020 | 39.2 | 0.3 | 8.3 |
>     | ChemLLM-20B-DPO | 45.3 | 140.6 | 46.3 | 0.022 | 12.0 | 6.0 | 0.5 |
>     | ChemDFM-13B-v1.0 | 13.7 | 162.9 | 38.2 | 0 | 0.8 | 1.8 | 0.5 |
>     | ChemDFM-8B-v1.5 | 22.8 | 216.0 | 22.6 | 0.036 | 10.5 | 0.5 | 0.1 |
>     |||||||||
>     | GPT-4o | 59.0 | **63.5** | 67.9 | 0.105 | 51.5 | 15.0 | 40.0 |
>     | Qwen2.5-14B-Instruct | 63.0 | 68.1 | 57.1 | 0.013 | 43.2 | 4.7 | 16.7 |
>     | Qwen3-14B (no think) | **76.1** | 85.7 | 59.0 | 0.078 | 34.5 | 8.2 | 35.1 |
>     |||||||||
>     | DeepSeek-R1 | 74.0 | 73.6 | 59.4 | 0.318 | 62.5 | 26.7 | 48.4 |
>     | Qwen3-14B (think) | 59.4 | 75.5 | 39.4 | 0.125 | 32.8 | 10.6 | 25.3 |
>     | o4-mini | 70.8 | 82.9 | **69.9** | **0.499** | **68.2** | **31.8** | 44.4 |
>     |||||||||
>     | ChemDFM-R | 65.0 | 90.1 | 69.1 | 0.184 | 48.7 | 24.4 | **49.2** |
>
>     From these results, we draw three main observations:
>
>     * First, **ChemDFM-R outperforms both traditional chemistry LLMs and Qwen2.5-14B-Instruct**. This demonstrates that our model possesses stronger chemical knowledge and more advanced chemical reasoning capabilities, enabling superior performance on reasoning-intensive tasks. This also validates the effectiveness of our ChemDFM-R training pipeline.
>
>     * Second, compared with the more powerful Qwen3-14B, our model achieves better results only on ChemCOTBench. We believe this is **due to the fundamental differences among the benchmarks**: unlike ChemCOTBench and ChemEval, which primarily consist of open-ended generative tasks across multiple subfields, **FGBench is composed exclusively of multiple-choice questions of property selection and numerical prediction problems of property prediction**. **For multiple-choice questions**, as analyzed in Appendix G.2, the required comprehension and generation ability is considerably lower than for free-form generation, allowing Qwen3 to circumvent its weaknesses and achieve strong results. **For numerical prediction tasks**, the performance gap aligns with our analysis in Section 4.1.1, further highlighting the limitations of ChemDFM-R on numerically intensive tasks and pointing toward directions for future improvement. **Therefore**, we believe that ChemDFM-R’s weaker performance on FGBench relative to Qwen3 primarily reflects its remaining room for improvement on tasks related to numerical reasoning and prediction, whereas its superior performance on the broader ChemCOTBench highlights its advantage in overall chemical capability.
>
>     * Finally, when compared with the 671B DeepSeek-R1 and the advanced closed-source model o4-mini, ChemDFM-R generally shows lower performance on both datasets. Given the substantial differences in model scale, we consider this partially expected and acceptable. In future work, we plan to extend our methods to larger LLMs to provide stronger chemical generalist reasoning LLMs for the open-source community.
>
> [1]. Xuan Liu, Siru Ouyang, Xianrui Zhong, Jiawei Han, and Huimin Zhao. Fgbench: A dataset and
> benchmark for molecular property reasoning at functional group-level in large language models.
> Advances in Neural Information Processing Systems, 2025.
>
> [2]. Hao Li, He Cao, Bin Feng, Yanjun Shao, Xiangru Tang, Zhiyuan Yan, Li Yuan, Yonghong Tian,
> and Yu Li. Beyond chemical qa: Evaluating llm’s chemical reasoning with modular chemical
> operations. arXiv preprint arXiv:2505.21318, 2025.

---

> ### Author Response · Authors · 2025-11-25
> **Part [5/6] Author Response**
>
> 2. **Results of o3-mini:**
>
>     We did not include o3-mini in the original manuscript because we used the next model in its series, o4-mini. The overall behavior of o3-mini is consistent with that of o4-mini, and we report its performance in Table R2 below.
>
>     **Table R2.** The performance of o3-mini, o4-mini, and ChemDFM-R on SciKnowEval and ChemEval
>
>     | Model | SciKnowEval |||| ChemEval ||||
>     | :--- | :---: | :---: | :---: | :---: | :---: | :---: | :---: | :---: |
>     || text | mol. | react. | all | text | mol. | react. | all |
>     | o3-mini | **83.1** | 53.9 | 94.3 | 73.5 | 61.7 | 33.3 | 53.5 | 62.6 |
>     | o4-mini | 81.3 | **57.0** | **97.3** | **74.0** | 78.4 | 60.7 | 53.6 | 67.3 |
>     | ChemDFM-R | 76.8 | 52.0 | 94.5 | 69.5 | **80.3** | **84.7** | **61.3** | **75.9** |
>
> 3. **Claims in lines 90–92:**
>
>     The decomposition of reasoning capabilities is the motivation and guiding principle behind our mixed-source distillation design. **The core contribution of mixed-source distillation lies in that it reduces the reliance on large-scale real distillation data** by leveraging the expert knowledge implicitly contained in the instruction-tuning data (which does not include explicit reasoning) and the pseudo-rationales (which are constructed from key intermediate steps). Consequently, it better initializes the model’s reasoning ability under limited resources and ultimately improves the performance of the RL-trained LLM. **It offers a new perspective: simply co-training with non-reasoning data can considerably enhance reasoning performance, reducing dependence on expensive real distillation data.** We have revised the corresponding text to focus more on the actual contributions rather than on the design logic.
>
>     **We mainly validate the effectiveness of this idea by comparing the performance of models distilled from different data compositions** (as shown in Table 4). We agree that directly validating changes in individual decomposed reasoning capabilities would be ideal, since this will further prove the correctness of our design logic. However, these three aspects are intertwined and jointly influence the final performance, making them difficult to disentangle for isolated evaluation. The improvement is therefore reflected indirectly through the performances of the resulting models.
>
> ## Q1. More details of the human annotation on quality control
>
> Thank you for your question. We acknowledge that the original manuscript lacked sufficient details regarding human quality control and included some overly approximate numbers, and we apologize for this oversight. The related information is as follows:
>
> * The experts we hired all possess graduate-level expertise in chemistry.
> * The exact accuracy of the functional group annotations is **98%** for molecules and **89%** for reactions. Examples are demonstrated in Appendix A.3.
> * The validation results for the teachers’ rationales are presented in Figure 11: **67%** of the rationales are of high quality, **23%** contain minor flaws that do not affect their overall correctness, and only **10%** exhibit substantive issues. Examples are demonstrated in Appendix C.2.
>
> We have added these details to the corresponding position of the revised manuscript.

---

> ### Author Response · Authors · 2025-11-25
> **Part [6/6] Author Response**
>
> ## Q2. Intuition or reasons for the ratios in mixed-source distillation
>
> Thank you for your valuable question. The proportion of different data sources is indeed an important factor influencing model performance. Due to the high computational cost of our full second-stage training (approximately 3,327 GPU hours), we regret to say that we were unable to perform a rigorous search for the optimal data composition. Instead, our chosen proportions were guided primarily by intuition.
>
> First, given our limited resources available for distillation, we collected as much real distillation data as possible. Second, because training solely on distillation data can harm the model’s overall performance, we propose to incorporate non-reasoning data to preserve basic chemical knowledge and capability. The overall size of the distillation dataset is thus set to be roughly identical to that of the instruction-tuning dataset. Finally, considering that our pseudo-reasoning data have relatively fixed patterns, an excessive amount of such data could reduce the model’s reasoning generalization ability. Therefore, we kept a relatively small portion of pseudo-reasoning data. These considerations led to our final mixed-source distillation dataset.
>
> However, although the specific proportions may not be optimal, this does not undermine the validity or contribution of our mixed-source distillation design. The effectiveness of this approach has been verified through ablation studies, as presented in Section 4.2.
>
> ## Q3. Reason for less improvement on SciKnowEval than ChemEval
>
> Thank you for your valuable question. In our view, **the core reason lies in the difference in task difficulty between the two benchmarks**. The majority of tasks in SciKnowEval are multiple-choice questions (MCQs), whereas the majority in ChemEval require open-ended generation. Compared with generation tasks, MCQs spare models from the burden of generating answers (including SMILES) on their own and also allow models to obtain the correct answer via shortcuts such as eliminating incorrect options.
>
> **The fact that o4-mini performs slightly better than ChemDFM-R on SciKnowEval is consistent with this observation**, as o4-mini is very good at using elimination strategies in MCQs. In contrast, ChemEval requires deeper chemical knowledge and skills, where our model demonstrates a clear performance advantage over o4-mini. This further supports our earlier analysis.
>
> Therefore, the overall difficulty of MCQs and the level of chemical understanding required by MCQs are considerably lower than those demanded by open-ended generation tasks. Given that it is widely recognized that enhanced reasoning capabilities benefit models more on harder problems, it is reasonable that RL training yields smaller improvements on SciKnowEval than on ChemEval.
>
> Originally, the above analysis comparing o4-mini and ChemDFM-R was included in the main manuscript, but due to space limitations, it was moved to Appendix G.2.

---

> ### Author Response · Authors · 2025-12-03
> **Summarization of the discussion with Reviewer rgJV**
>
> Dear ACs/PCs,
>
> Reviewer rgJV **speaks highly of our substantial technical and engineering efforts, the novelty of the integration of functional group knowledge, and our solid and comprehensive experiments**. His/her main concerns focus on the overall research value of our work and the solidity of the case study. The point-to-point summary are as follows.
>
> * **W1: About research value. (Parts [1/6] and [2/6])**
>
>     We elaborate the research value of our work from three complementary dimensions: 1) **methodological contributions**, which include the integration of atomized functional-group knowledge and the proposal of mixed-source distillation; 2) **engineering contributions**, which are no secondary to methodological ones and are very important and substantial during our work; 3) **application value**, which is also significant as we provide a stronger, more verifiable and interaction-friendly assistant for chemists and a foundation for future chemical LLM improvements and agent systems development for the open-source communiy. **A much more detailed analysis with examples are provided in Parts [1/6] and [2/6].**
>
> * **W2: Statistical comparison regarding human-AI collaboration. (Part [3/6])**
>
>     We follow his/her advice and **perform statistical comparisons through human-assessment experiments**. The results show that **ChemDFM-R generates more accurate, concise, and informative rationales during human-AI interactions in practical chemical research scenarios**, thus facilitating more reliable and interaction-friendly human-AI collaborations. Please refer to our response Part [3/6] for the experimental results and more detailed analyses.
>
> * **W3.1: More evaluations on newly proposed benchmarks. (Part [4/6])**
>
>     We **conducted further evaluations on two additional benchmarks, FGBench and ChemCOTBench**. The results and analyses are reported in Appendix G.4 and our response Part [4/6]
>
> * W3.2: The results of o3-mini. (Part [5/6])
>
>     We did not include o3-mini in the original manuscript because we used its successor, o4-mini. We provide the results of o3-mini on SciKnowEval and ChemEval in our response Part [5/6] and the overall behavior of o3-mini is consistent with that of o4-mini.
>
> * **W3.3: The claim made in lines 90-92 (decomposition of reasoning capabilities) needs better support. (Part[5/6])**
>
>     We clarify that **the decomposition of reasoning capabilities is the motivation and guiding principle** behind our mixed-source distillation design. **The core contribution of mixed-source distillation lies in that it reduces the reliance on large-scale real distillation data to achieve better initilization of pre-RL LLMs' reasoning capabilties.** We mainly **validate the effectiveness of this idea by comparing the performance of models distilled from different data compositions (as shown in Table 4)**. We have revised the corresponding text to focus more on the actual contributions rather than on the design intuition.
>
> * Q1: More details of the human annotation on quality control. (Part [5/6])
>
>     We provide the requested details in the Appendix A.3 and C.2.
>
> * Q2: Intuition or reasons for the ratios in mixed-source distillation. (Part [6/6])
>
>     Due to the high computational cost, we were unable to perform a rigorous search for the optimal data composition. Therefore, the choice of the ratios were mainly based on intuitions. Please refer to our response Part [6/6] for detailed explanation of the intuitions.
>
> * Q3: Reason for less improvement on SciKnowEval than ChemEval. (Part [6/6])
>
>     In our opinion, the core reason lies in the difference in task difficulty between the two benchmarks. Please refer to our response Part [6/6] for more detailed analyses.

---

### Official Review · Reviewer_hZbS · 2025-10-28

**Soundness:** 2
**Presentation:** 3
**Contribution:** 3
**Rating:** 6
**Confidence:** 4

**Summary:**

The authors present ChemDFM-R, a chemical-reasoning LLM centered on functional-group knowledge and trained in stages: domain pretraining, domain instruction tuning, mixed-source distillation, and RL; while explicitly preserving general language abilities. It retains performance on text-centric tasks and substantially outperforms its base model on molecule- and reaction-centric evaluations.

**Strengths:**

- S1: Strong concept: A chemical-reasoning LLM that handles a broad set of structure-level tasks (analysis, transformation, and property reasoning) is timely and useful.

- S2: Functional-group–centric reasoning: Framing reasoning around functional groups, rather than atoms or whole-molecule graphs, aligns with chemists’ mental models and can improve interpretability.

- S3 Capability retention: The effort to preserve general LM abilities (and to verify limited degradation on text tasks) is well-motivated; it justifies using an LLM rather than a narrow specialist and supports broader applicability.

**Weaknesses:**

- W1: Lack of uncertainty and significance.
The significance of results cannot be assessed without variability estimates. If these are Pass@1, please run ≥3 independent runs with different seeds and report mean ± s.d., plus ideally paired significance tests. Please do this at least for you model, to see if the others land within your error bars.

 - W2: Code (or full specs) unavailable.
 Please release training and evaluation code (an anonymized repo is fine). If code release isn’t possible, provide a complete appendix with information such as exact prompts/templates and inference settings to ensure reproducibility.

 - W3: Even if near-contemporaneous, cite early chemical-reasoning LLM preprints (e.g., ether0) in Introduction/Related Work for completeness and positioning.

 - W4: No limitations/outlook.
Add a brief Limitations/Outlook section covering scope and failure modes and outline mitigation or future directions.

- W5: Compute and data budget missing.
Please report the compute footprint and the number of tokens used in each training stage.

**Questions:**

- Q1: What do you mean by "an average instruction-entry ratio of 1:50" on L197?

- Q2: Do the SciKnowEval / ChemEval suites include tasks absent from your training data? If yes, quantify: how many (or what fraction) are exact overlaps vs. novel tasks, and how does performance differ between seen and unseen task types? If no, please add an evaluation on tasks not explicitly trained on to support claims of generalization.

- Q3: Please use a common y-axis across subplots in Fig. 5; the current scaling overstates the impact of distillation+RL. Also, due you have any hypothesis for why the second phase yields much smaller gains than the first.

- Q4: What metric is shown in Tables 2,3 and 4?

- Q5: You currently group domain pretraining+instruction tuning and distillation+RL. It would be interesting to get the effect all four steps independently, especially disentangling distillation vs. RL effects.

- Q6: How confindent are you in the "reliability" of your chemical reasoning chatbot? How many of these multi-step discussion did you do and analyze?

---

> ### Author Response · Authors · 2025-11-25
> **Part [1/7] Author Response**
>
> Thank you for your acknowledgement of our contribution and your valuable feedback. We have carefully revised our manuscript and provided detailed responses to each of your comments below. We hope that these replies address your concerns and enhance the quality of our work. We sincerely welcome any further discussions or suggestions.
>
> ## W1. Lack of uncertainty and significance
>
> Thank you for the great advice. We evaluate each task using the metrics recommended by the corresponding datasets, and all results are computed based on single-sample outputs. **We did not include error bars in the initial submission mainly due to computational budget constraints.** The full training of our model, starting from domain pretraining to RL, consumes roughly **30,840** GPU hours (detailed information has been added to the Appendix E.2.). This limitation is also commonly encountered in works involving LLM Training, making significance testing via repeated full training difficult.
>
> However, we agree that variability estimates are important for evaluating the reliability of the results. Although we cannot fully rule out variability introduced during training, we **mitigate randomness at the inference stage by conducting multiple inference runs with different random seeds**. These results including average performance and standard deviation have been updated to Table 2. For your convenience, we have also transcribed Table 2 below as Table R1.
>
> **Table R1.** The average performance of five inference runs and the standard deviation (in parentheses) on SciKnowEval and ChemEval.
>
> | Model | SciKnowEval |||| ChemEval ||||
> | :--- | :---: | :---: | :---: | :---: | :---: | :---: | :---: | :---: |
> || text | mol. | react. | all | text | mol. | react. | all |
> | Qwen2.5-14B-Instruct | 77.0(0.11) | 35.7(0.23) | 72.1(0.24) | 61.3(0.12) | 79.6(1.43) | 24.8(0.63) | 46.8(1.81) | 57.7(0.72) |
> | ChemDFM-I | **78.8**(0.38) | 50.6(0.40) | 91.1(0.70) | **69.7**(0.23) | **81.2**(2.48) | 68.4(3.56) | 54.6(5.19) | 70.7(1.10) |
> | ChemDFM-R | 76.7(0.18) | **51.1**(0.56) | **93.8**(0.37) | 69.1(0.27) | 78.3(2.38) | **83.5**(1.89) | **58.5**(2.61) | **73.8**(1.83) |
>
> The results in Table R1 are consistent with the analyses presented in our original manuscript: the performances on text-centric tasks remain almost intact while those on molecule-centric and reaction-centric tasks increase significantly. Moreover, your suggestion helped us uncover a new and interesting analysis. Results in Table R1 show that **the atomized chemical knowledge enhancement stage consistently improves performance across all task categories**, while **the chemical rationale learning stage further strengthens performance on molecule- and reaction-centric tasks**. We attribute this phenomenon to two factors. *First*, the molecule and reaction-centric tasks typically demand more domain-specific chemical reasoning, such as molecular property prediction or retrosynthesis analysis. In contrast, text-centric tasks rely more on natural language understanding, such as chemical named entity recognition and literature question-answering. As a result, learning chemical reasoning over molecules and reactions provides limited benefit to these text-focused tasks. *Second*, for the sake of answer verifiability, the reinforcement learning tasks do not include purely text-based tasks, which may adversely affect the model’s text reasoning ability. Incorporating text-related tasks into the RL stage through joint training might help preserve performance on text-based tasks. These analyses have been updated to Section 4.1.1.
>
> ## W2. Code unavailable
>
> Thank you for your question. We train our model based on the open-sourced frameworks DeepSpeed-Chat (for all the training stages except RL) and OpenRLHF (for the RL training). To improve transparency, we have added a section in the appendix (Appendix E.3) that provides the detailed training hyperparameters, inference settings, and the system prompt for our model. We will also release the resulting models, as well as our SFT dataset, mixed-source distillation dataset, RL dataset, and our functional group identification toolkit upon publication to contribute to the open-source community.

---

> ### Author Response · Authors · 2025-11-25
> **Part [2/7] Author Response**
>
> ## W3. Lack of citation
>
> We sincerely apologize for the oversight. We have properly cited ether0 and discussed the differences between our approach and theirs in the related work section. While ether0 is a great pioneering work that teaches LLMs to reason about chemistry, it is restricted to specific tasks rather than functioning as a generalist chemical reasoner. As emphasized on its HuggingFace page[1], ether0 is not a general purpose chat model, but a multi-task model specifically for the trained tasks.
>
> Our method introduces two key components that are not included in ether0: (1) atomized functional-group knowledge enhancement to provide LLMs with fine-grained chemical knowledge as reasoning "ingredients", and (2) mixed-source distillation to increase the quality of distillation data with limited resources to facilitate better reasoning.
>
> As a result, we managed to train a generalist chemical reasoning LLM, ChemDFM-R, which could be viewed as a step further from ether0 toward enabling practical reasoning LLMs as AI assistants in chemistry.
>
> [1]. futurehouse/ether0 · Hugging Face, 6 2025. URL https://huggingface.co/futurehouse/ ether0. [Online; accessed 2025-11-17].
>
> ## W4. No limitations/outlook
>
> Thank you for your constructive advice. We have added a limitation section in our revised manuscript. Due to the space limitation, the limitation section is added as Appendix K, and is transcribed below.
>
> > In this work, we developed a chemical generalist reasoning model using atomized chemical knowledge enhancement and mixed-source distillation-based chemical rationale learning. Through both benchmark evaluations and human assessments, we demonstrated the strong potential of ChemDFM-R in solving chemical problems and supporting human–AI collaboration. The evaluations also revealed several limitations of our model, which can be summarized in three aspects.
> >
> > First, as shown in Section 4.1.1, the chemical rationale learning phase substantially enhances performance on molecule- and reaction-centric tasks but weakens general language abilities, especially numerical prediction abilities. This is mainly because our RL stage relies on rule-based methods to calculate rewards, which restricts the types of tasks we can use. Designing reward mechanisms better suited to chemical contexts and allowing a broader range of RL tasks will be an important direction for the improvement of ChemDFM-R.
> >
> > Second, results in Table 5 indicate that although our model produces concise and informative reasoning chains, which greatly improve its usability and friendliness in human–AI interactions, this conciseness can compromise analytical coverage. Achieving a better balance between maintaining concise reasoning chains and providing sufficiently diverse analyses could further enhance ChemDFM-R's practical value in interactive settings.
> >
> > Finally, similar to general reasoning models, ChemDFM-R occasionally exhibits inconsistencies between its reasoning process and final answers. In chemistry, this includes both mismatches between reasoning steps and conclusions, and the use of incorrect chemical knowledge within the reasoning. Enhancing reward design to enforce consistency and incorporating stronger chemical supervision will be another key area for future improvement.
>
> ## W5. Compute and data budget missing
>
> Thank you for your question. We apologize for not providing the computational cost in our original manuscript. We have added a dedicated section in the Appendix (Appendix E.2) to report these details. In total, our full training pipeline requires 30,840 GPU hours on NVIDIA A800 Tensor Core GPUs, including:
>
> * 24,728 GPU hours for domain pretraining,
> * 3,785 GPU hours for instruction tuning,
> * 2,059 GPU hours for mixed-source distillation,
> * 268 GPU hours for reinforcement learning.
>
> The data usage for each stage is as follows:
>
> * The domain pretraining corpus contains 101B tokens;
> * The instruction tuning dataset contains over 2.5M chemistry instructions, paired with 5M general-domain instruction-tuning samples;
> * The mixed-source distillation dataset is of the same scale as the instruction-tuning dataset;
> * The RL training dataset contains 121,811 samples in total.

---

> ### Author Response · Authors · 2025-11-25
> **Part [3/7] Author Response**
>
> ## Q1. The meaning of "an average instruction-entry ratio of 1:50"
>
> Sorry for the confusion. In short, we generate multiple instructions for each task so that the total number of instructions exceeds 1/50 of the number of data entries for that task. This approach aims to diversify the expression of task intentions and prevent the model from overfitting to a few fixed instructions and losing its generalization capability. We will provide more details and include examples of different instructions for the same task in the following paragraph
>
> **Details explanation:**
> > To illustrate, consider a property prediction task with 20,000 entries. Each entry is a molecule-property pair. To construct the SFT dataset, we need to wrap each query with a natural language instruction, such as:
> >
> > "What is the [property name] of the molecule represented by SMILES: [molecule SMILES]?"
> >
> > If we were to use the same instruction for all 20,000 entries, the model could infer the task intention from the repetitive patterns rather than the instruction semantics. This would reduce generalizability and make the model act more like a multi-task specialist instead of a generalist. To mitigate this, we rewrite the seed instructions following the method described in Appendix B.2, ensuring that the number of unique instructions is at least 1/50 of the total entries. For the example task with 20,000 entries, this results in at least 4,000 distinct instructions.
>
> **Examples:**
> For dipole moment prediction task that contains 105,984 entries, we have a total of 3000 different instructions. Ten random examples are listed below.
>
> ```json
> [
>     "Predict the dipole moment (in D) of the molecule represented by the following SMILES notation: {smiles}\ndipole moment:",
>     "Using the provided SMILES {smiles}, calculate the dipole moment (in D) of the molecule.\ndipole moment:",
>     "As an expert chemist, your role is to utilize your extensive knowledge of molecular properties to forecast the characteristics of molecules. Adhere strictly to the prescribed format and refrain from providing any extraneous information. Using the SMILES string of a molecule, determine the dipole moment of the corresponding chemical compound based on its structural composition. Present the dipole moment in D directly, with no further elaboration.  \nSMILES: {smiles}\ndipole moment:",
>     "As an accomplished chemist, you are tasked with anticipating the attributes of molecules through your deep comprehension of molecular characteristics. Please adhere to the designated format meticulously and do not add any further information. With the SMILES string provided for a molecule, assess and predict the dipole moment of that chemical compound based on its architecture. State the dipole moment in D straightaway, excluding any additional commentary.\nSMILES: {smiles}\ndipole moment:",
>     "Predict the dipole moment in D of the chemical compound represented by the SMILES string: {smiles}.\nAs an expert chemist, base your answer solely on the molecule's structure.\nPlease respond with the dipole moment value only.",
>     "Predict the dipole moment of the molecule represented by the SMILES {smiles}, and provide the answer in D.",
>     "Using your knowledge of molecular properties, determine the dipole moment of the molecule with the SMILES string {smiles}.\nYour answer should be in D and provided directly, without any additional information or explanations.\ndipole moment:",
>     "Your expertise in molecular properties is requested to forecast the dipole moment of the molecule represented by its SMILES in D. This prediction holds significance for the subsequent research.\nSMILES: {smiles}\ndipole moment:",
>     "Given the SMILES: {smiles}, please provide the dipole moment of the molecule in D.\ndipole moment:",
>     "You possess expertise in chemistry, and your challenge is to predict molecular properties utilizing your skilled understanding of chemical characteristics. It is imperative that you follow the specified format rigorously and refrain from providing additional information. When given the SMILES string for a molecule, deduce the dipole moment related to the indicated chemical compound based on its structure. Present the dipole moment in D directly, without including any other context.\nSMILES: {smiles}\ndipole moment:"
> ]
> ```

---

> ### Author Response · Authors · 2025-11-25
> **Part [4/7] Author Response**
>
> ## Q2. Supports for claims of generalization
>
> Thank you for your insightful question. First, we would like to clarify that **our training dataset was not specifically designed according to the evaluation benchmarks**. Consequently, there are no tasks in our training data that **exactly** correspond to those in the evaluation benchmarks. However, if we adopt a more relaxed criterion, some tasks may share the same target but are expressed through different instructions. Following your suggestion, we have separately reported the number of tasks and performance for:
>
> 1. **Same Tasks with Different Instructions (DI)** – to demonstrate the instruction-level generalizability of our model.
>
> 2. **Different Tasks (DT)** – to demonstrate the task-level generalizability of our model.
>
> The results are presented in Table R2 below and have been added to Appendix G.3.
>
> **Table R2.** Results on DI tasks and DT tasks in SciKnowEval and ChemEval.
>
> | Model | SciKnowEval |||| ChemEval |||||
> |:------|:-----------:|:---:|:---:|:---:|:---:|:---:|:---:|:---:|:---:|
> || **DI** | DT | DT | DT | **DI** | **DI** | DT | DT | DT |
> || mol. | text | mol. | react. | mol. | react. | text | mol. | react. |
> | MolInst | 39.4 | 69.4 | 38.0 | 41.7 | 10.4 | 10.0 | 47.3 | 42.7 | 8.6 |
> | ChemLLM-20B-DPO | 37.0 | 73.6 | 38.9 | 88.1 | 20.9 | 28.4 | 64.5 | 49.7 | 14.8 |
> | ChemDFM-13B-v1.0 | 53.6 | 70.2 | 38.3 | 85.0 | 44.9 | 13.0 | 61.5 | 50.4 | 39.2 |
> | ChemDFM-8B-v1.5 | 55.9 | 72.2 | 41.4 | 86.9 | 10.1 | 38.1 | 26.7 | 19.1 | 13.3 |
> | ether0 | 35.6 | 38.7 | 33.0 | 66.0 | 18.4 | 50.0 | 9.4 | 1.3 | 1.6 |
> |||||||||||
> | GPT-4o | 37.4 | 76.1 | 47.5 | 44.6 | 33.8 | 42.2 | **82.7** | 52.3 | 55.5 |
> | Qwen2.5-14B-Instruct | 23.9 | 77.2 | 40.5 | 71.9 | 17.3 | 50.0 | 77.5 | 53.8 | 41.2 |
> | Qwen3-14B (no think) | 23.6 | 76.5 | 38.9 | 80.2 | 19.3 | 32.9 | 81.5 | 52.5 | 25.4 |
> |||||||||||
> | DeepSeek-R1 | 40.9 | 80.1 | 26.8 | 83.0 | 14.0 | 55.3 | 81.2 | 50.1 | 46.0 |
> | Qwen3-14B (think) | 27.3 | 86.6 | 46.2 | 88.5 | 19.2 | 26.1 | 79.0 | 52.8 | 44.4 |
> | o4-mini | 44.7 | **81.3** | **61.9** | **97.3** | 51.7 | 58.5 | 78.4 | **86.4** | 52.4 |
> |||||||||||
> | ChemDFM-R | **68.9** | 76.8 | 45.2 | 94.5 | **95.5** | **66.3** | 80.4 | 62.0 | **60.0** |
>
> From these results, we draw three key observations.
>
> First, **while enhancing the model’s chemical capabilities, our training procedure largely preserves strong generalization performance**. Compared with our baseline model, Qwen2.5-14B-Instruct, ChemDFM-R not only achieves substantially better performance on DI tasks but also improves on DT tasks, demonstrating that the knowledge acquired through training can be generalized to new tasks.
>
> Second, **our model performs significantly better on DI tasks than on DT tasks**, indicating that it exhibits stronger instruction-level generalization than task-level generalization.
>
> Finally, our model underperforms o4-mini on molecule-centric DT tasks in ChemEval. Upon further inspection, we find that among the three tasks in this category, only the ``SMILES-to-SELFIES and SELFIES-to-SMILES conversion'' tasks show inferior performance compared with o4-mini. **This outcome mainly arises from the fact that we used only SMILES as the molecular representation during training.** Considering that **SMILES and SELFIES can be deterministically converted** into each other through rule-based transformations, we believe that **this weaker performance, or the lack of explicit SELFIES understanding and generation capability, does not materially affect the model’s usefulness or effectiveness in practical chemistry applications**.

---

> ### Author Response · Authors · 2025-11-25
> **Part [5/7] Author Response**
>
> ## Q3. Rescle y-axis of Figure 4 (original Figure 5) and explanation of the smaller improvements
>
> Sorry for our oversight. We have corrected the y-axis in the new version of the manuscript. Thank you very much for pointing it out.
>
> Regarding the smaller improvement in the second stage, this is to some extent in line with our expectations. Specifically, in our understanding, the two stages of our training pipeline serve different purposes. **The first stage focuses on knowledge enhancement**, during which the model learns new knowledge and skills, **leading to significant performance gains on tasks related to newly acquired knowledge**. In contrast, **the second stage primarily teaches the model how to apply its existing knowledge**. Consequently, the potential for performance improvement is inherently smaller. This is also consistent with the nature of RL training: **improvements are limited if the policy model cannot sample better responses**. Therefore, our **main goal for the second stage is to enable the model to generate rationales, thereby *improving its reliability and interpretability***, rather than substantially boosting task performance.
>
> However, we agree that our current methodology of RL training could be further optimized. In particular, more domain-specific RL algorithm design may help achieve larger performance gains. As our study primarily focuses on atomized knowledge enhancement and resource-efficient distillation, we did not explore specialized designs or improvements for the RL stage in this work. This is one of our key directions for future work, and we welcome further discussions on this topic.
>
> ## Q4. Metrics in Table 2, 3, and 4
>
> Thank you for your question. First, we apologize for the typo in our table captions: all instances of “PRS” should be “RPS”.
>
> Regarding your question, we calculate performance using the metrics recommended for each task in the original papers of the two benchmarks[2,3]. Moreover, for ChemEval, due to the large variation in task difficulty and the presence of metrics with opposite optimization directions (e.g., RMSE, where lower is better, versus ACC or BLEU, where higher is better), we adopt the RPS score proposed by Peng et. al.[4] to unify the standard across tasks. The original metric values are reported in Table 7 in Appendix G.1.
>
> [2]. Kehua Feng, Keyan Ding, Weijie Wang, Xiang Zhuang, Zeyuan Wang, Ming Qin, Yu Zhao, Jianhua Yao, Qiang Zhang, and Huajun Chen. Sciknoweval: Evaluating multi-level scientific knowledge of large language models. arXiv preprint arXiv:2406.09098, 2024.
>
> [3]. Yuqing Huang, Rongyang Zhang, Xuesong He, Xuyang Zhi, Hao Wang, Xin Li, Feiyang Xu, Deguang Liu, Huadong Liang, Yi Li, et al. Chemeval: a comprehensive multi-level chemical evaluation for large language models. arXiv preprint arXiv:2409.13989, 2024.
>
> [4]. Jing Peng, Yucheng Wang, Bohan Li, Yiwei Guo, Hankun Wang, Yangui Fang, Yu Xi, Haoyu Li, Xu Li, Ke Zhang, Shuai Wang, and Kai Yu. A survey on speech large language models for understanding. arXiv preprint arXiv:2410.18908, 2024.

---

> ### Author Response · Authors · 2025-11-25
> **Part [6/7] Author Response**
>
> ## Q5. The effect of distillation v.s. RL training
>
> Thank you for your insightful question. Following your advice, we evaluated the model after the mixed-source distillation stage but before RL training, referred to as ChemDFM-D. The results are reported in Table R3 below.
>
> **Table R3.** The average performance of five inference runs and the standard deviation (in parentheses) on SciKnowEval and ChemEval.
>
> | Model | SciKnowEval |||| ChemEval ||||
> | :--- | :---: | :---: | :---: | :---: | :---: | :---: | :---: | :---: |
> || text | mol. | react. | all | text | mol. | react. | all |
> | Qwen2.5-14B-Instruct | 77.0(0.11) | 35.7(0.23) | 72.1(0.24) | 61.3(0.12) | 79.6(1.43) | 24.8(0.63) | 46.8(1.81) | 57.7(0.72) |
> | ChemDFM-I | **78.8**(0.38) | 50.6(0.40) | 91.1(0.70) | **69.7**(0.23) | **81.2**(2.48) | 68.4(3.56) | 54.6(5.19) | 70.7(1.10) |
> | ChemDFM-D | 74.8(1.50) | 46.8(0.55) | 80.3(0.98) | 65.1(0.95) | 64.0(2.23) | 67.2(1.89) | 44.7(3.58) | 59.2(1.78) |
> | ChemDFM-R | 76.7(0.18) | **51.1**(0.56) | **93.8**(0.37) | 69.1(0.27) | 78.3(2.38) | **83.5**(1.89) | **58.5**(2.61) | **73.8**(1.83) |
>
> The model exhibits a slight performance drop after distillation, which matches our expectations, since **the common purpose of the distillation stage is to initialize the reasoning abilities of models and avoid the early unstable cold start phase at the beginning of reinforcement learning**[5]. Given the limited scale of the real distillation data in our distillation dataset, what likely happens is that the model begins to learn the reasoning-before-answering pattern but does not yet fully excel at it.
>
> Moreover, considering that a small portion of the distillation data still contains errors (as shown in Figure 11), RL training is necessary both to correct these errors and to further enhance the model’s reasoning capabilities. Consequently, the model achieves better overall performance after RL training.
>
> [5]. Daya Guo, Dejian Yang, Haowei Zhang, Junxiao Song, Ruoyu Zhang, Runxin Xu, Qihao Zhu, Shirong Ma, Peiyi Wang, Xiao Bi, et al. Deepseek-r1: Incentivizing reasoning capability in llms via reinforcement learning. arXiv preprint arXiv:2501.12948, 2025.

---

> ### Author Response · Authors · 2025-11-25
> **Part [7/7] Author Response**
>
> ## Q6. About "reliability"
>
> Thank you for your valuable question. First, we would like to clarify the meaning of “reliability” as used in our paper to avoid any misunderstanding. Our intention is not to claim that the model rarely makes mistakes and therefore produces inherently reliable answers. **Instead, by reliability we refer to the model’s ability to provide transparent and verifiable reasoning, which allows users to check the correctness of answers and to identify directions for further discussion.** Because our model outputs meaningful and concise reasoning steps, users can validate the final answer more easily than with traditional chemistry LLMs that produce no reasoning or general reasoning LLMs whose reasoning is excessively long but not informative. This is the sense in which we consider the interactions with our model “more reliable.”
>
> In our original manuscript, we present one detailed case in Section 4.3.2 and two in Appendix J illustrating how more informative and more readable rationales benefit knowledge discovery, error correction, and answer refinement. We acknowledge that the number of cases remains limited and **have expanded the comparative analysis across more questions to further substantiate this claim**.
>
> Specifically, we constructed ten graduate-level questions based on recent publications (within the past two years) from several influential chemistry journals. The questions cover different major subfields of chemistry, including organic chemistry, inorganic chemistry, materials chemistry, analytical chemistry, and polymer chemistry. Then, different reasoning LLMs are leveraged to solve these questions through human-AI interactions. Five graduate-level chemistry experts were hired to evaluate these interactions across the following five dimensions (each scored on a 5-point scale) to comprehensively assess the strengths and limitations of our model under scientific interactive situations:
>
> * **Chemical Correctness**: The correctness of chemical knowledge and logic demonstrated throughout the reasoning process.
> * **Answer Accuracy**: Whether the final answer is correct.
> * **Analytical Coverage**: The extent to which different plausible possibilities are explored during reasoning.
> * **Reasoning Coherence**: Whether the reasoning remains focused, coherent, and aligned with the problem.
> * **Effective Information Density**: The density of useful information in the reasoning chain, reflecting the friendliness and efficiency of interaction.
>
> The results are summarized in Table R4, from which we draw three main conclusions.
>
> **Table R4.** Results of human evaluation for human-AI interactions. As the rationales of o4-mini are not accessible, we only use the answer provided by o4-mini.
>
> | Models | Chemical Correctness | Answer Accuracy | Analytical Coverage | Reasoning Coherence | Effective Information Density |
> |:---|:---:|:---:|:---:|:---:|:---:|
> | ether0 | 1.56 | 1.60 | 1.40 | 1.76 | 1.56 |
> | DeepSeek-R1 | 4.10 | 3.88 | **4.34** | 4.42 | 3.62 |
> | Qwen3-14B (think) | 3.56 | 3.92 | 4.24 | 3.98 | 3.00 |
> | o4-mini | 3.94 | 3.96 | 3.84 | 4.16 | 4.12 |
> | ChemDFM-R | **4.52** | **4.28** | 3.94 | **4.62** | **4.70** |
>
> First, **our model outperforms all baselines, including DeepSeek-R1, in both chemical correctness and answer accuracy**, indicating that it possesses a stronger grasp of chemical knowledge. In particular, our model shows a clearer advantage on SMILES-related tasks, demonstrating a more precise understanding of molecular structures.
>
> Second, **our model achieves significantly higher scores in effective information density**. Unlike models such as Qwen3-14B and DeepSeek-R1, which often generate extremely lengthy reasoning chains exceeding 1,000 tokens, our model produces concise yet informative reasoning. This considerably reduces the burden on human users when verifying correctness or identifying errors.
>
> Finally, **our model performs slightly lower than Qwen3-14B and DeepSeek-R1 on analytical coverage**. This is partially because their extremely lengthy reasoning chains allow them to enumerate many possible considerations. It is also true that our model commonly focuses on one or a few major possibilities rather than adequately enumerating all potential factors. This highlights a productive direction for future enhancement of our model’s analytical breadth.
>
> We have added these results and analyses to Section 4.3.1. The ten original questions used for this evaluation are documented in Appendix I.

---

> ### Author Response · Authors · 2025-12-03
> **Summarization of the discussion with Reviewer hZbS**
>
> Dear ACs/PCs,
>
> Reviewer hZbS **speaks highly of the practical value of ChemDFM-R, our design choice of functional-group-centric training and reasoning, and our efforts to preserve general LM abilities**. His/her major concerns focus on the training details and stronger evaluations. The point-to-point summary are as follows.
>
> * **W1: Lack of uncertainty and significance. (Part [1/7])**
>
>     We **conducted multiple inferences with different random seeds and reported mean performance with standard deviations.**
>
> * W2: Code unavailable. (Part [3/7])
>
>     We promise to open-source both the resulting models, including ChemDFM-R and ChemDFM-I, and the training materials, including our SFT dataset, mixed-source distillation dataset, RL dataset, and the functional group identification toolkit, upon publication.
>
> * W3: Lack of citation (Part [2/7])
>
>     We have added a clear citation of ether0 and provided proper comparisons in the related-work section. We have also compared ChemDFM-R with ether0 across benchmark evaluations and human assessments, where ChemDFM-R shows consistently stronger performance. The results are reported in the corresponding section.
>
> * W4: No limitations/outlook. (Part [2/7])
>
>     We have added a limitation and futrue work section in the revised manuscript as Appendix K.
>
> * W5: Compute and data budget missing. (Part [2/7])
>
>     We report the detailed computation and data budget in Appendix E.2.
>
> * Q1: The meaning of "an average instruction-entry ratio of 1:50". (Part [3/7])
>
>     We provide a detail explanation with examples in our response Part [3/7].
>
> * **Q2: Experimetal supports for claims of generalization. (Part [4/7])**
>
>     We grouped the tasks in SciKnowEval and ChemEval into two categories: 1) **Same Tasks with Different Instructions (DI)**, demonstrating **the instruction-level generalizability** of our model; 2) **Different Tasks (DT)**, demonstrating **the task-level generalizability** of our model. Results and analyses are reported in Appendix G.3 and our response Part[4/7].
>
> * Q3: Rescle y-axis of Figure 4 (original Figure 5) and explanation of the smaller improvements. (Part [5/7])
>
>     We redrew the figure to address the oversight and provide a detailed analyses of the smaller improvements in our response Part [5/7].
>
> * Q4: What metric is shown in Tables 2,3 and 4? (Part [5/7])
>
>     We provide detailed explanation in our response Part[5/7].
>
> * Q5: The effect of distillation v.s. RL training. (Part [6/7])
>
>     We evaluate the pre-RL model, ChemDFM-D, on SciKnowEval and ChemEval and compare the performance between ChemDFM-I, ChemDFM-D, and ChemDFM-R. Please refer to our response Part [6/7] for detailed analyses.
>
> * **Q6: About "reliability". (Part [7/7])**
>
>     We first clarify that “reliability” in our manuscript does **not imply the model rarely makes mistakes and therefore produces inherently reliable answers**. Instead, by reliability we **refer to the model’s ability to provide transparent and verifiable reasoning, which allows users to validate the correctness of answers and to identify directions for further discussion**. Our original manuscript includes one detailed example in Section 4.3.2 and two more in Appendix J illustrating how more concise, informative, and readable rationales support knowledge discovery, error correction, and answer refinement. To further substantiate this point, we conducted a new human-assessment experiment evaluating the quality of the rationales from different reasoning LLMs across multiple metrics, demonstrating the effectiveness and efficiency of ChemDFM-R's rationales during human-AI interaction in practical chemical research scenarios.

---

### Official Review · Reviewer_Pn3p · 2025-10-30

**Soundness:** 3
**Presentation:** 3
**Contribution:** 3
**Rating:** 6
**Confidence:** 4

**Summary:**

This paper presents ChemDFM-R, a LLM for chemistry, built by fine-tuning Qwen2.5-14B through a four-stage pipeline: (1) domain pretraining on a functional-group-centered chemistry corpus (ChemFG), (2) instruction tuning, (3) reasoning distillation with functional group rationales and teacher LLMs, and (4) domain-specific RL with a reward that promotes correct, well-structured reasoning. The model demonstrates strong performance on several chemistry benchmarks and introduces a novel dataset focused on functional group transformations.

**Strengths:**

1. The integration of functional group annotations in both training and evaluation is novel and impactful.

2. The paper performs careful evaluation across benchmarks and multiple tasks, and shows competitive or superior performance.

3. The RL finetuning with structured reward is well-aligned with reasoning goals.

**Weaknesses:**

1. Some implementation details, such as reward shaping specifics, filtering of rationale data, are not specified.

2. The current evaluation focuses mainly on answer correctness. More human evaluation of rationale quality could make it more complete.

3. The datasets used to train the model are not released, which can hinder the reproducibility.

**Questions:**

1. How does the model handle conflicting functional group cues when multiple transformations are possible?

2. Are there examples where the model hallucinates chemically invalid rationales despite producing correct answers?

---

> ### Author Response · Authors · 2025-11-25
> **Part [1/3] Author Response**
>
> Thank you for your insightful questions and constructive feedback. We have carefully revised our manuscript and provided detailed responses to each of your comments below. We hope that these replies address your concerns and enhance the quality of our work. We sincerely welcome any further discussions or suggestions.
>
> ## W1. Implementation details
>
> Thank you for your question. With our apologies, we are not completely certain which specific information you are requesting, and we hope the explanation below addresses your concerns.
>
> 1. **Reward shaping specifics.** We infer that you may be asking how the RL reward is computed. We provide a brief explanation in Appendix D, and here we include additional details. Specifically, we first check whether the model’s response follows the required reasoning format.
>
>     ```text
>     <think>
>     ...
>     </think>
>     <answer>
>     ...
>     </answer>
>     ```
>
>     If not, a reward of 0 is assigned. Otherwise, we continue to evaluate whether the response answers the question correctly. To do this, we extract the answer portion, canonicalize the SMILES strings, and check whether the predicted answer exactly matches the ground-truth answer. If it does not match, the reward is 1; if it matches, the reward is 2.
>
> 2. **Filtering of rationale data.** We assume that you are asking about the distilled rationales from DeepSeek-R1 and o3-mini. As stated in Appendix C.2 (originally in Section3.2), we do not apply any filtering and use all distilled rationales directly. This is because (1) distinguishing correct from incorrect rationales is highly difficult, as errors are often subtle and deeply hidden, and (2) prior work has shown that correctness is not strictly necessary for distillation to be effective when followed by RL training[1,2]. We believe this is because the common purpose of distillation stage is to initialize the reasoning abilities of LLMs and prevent the early unstable cold start phase of RL training[3]. The model can learn to correct mistakes in the distilled rationales during subsequent RL training.
>
> Please let us know if our response addresses your questions, and we sincerely welcome further discussions.
>
> [1]. Shuyao Xu, Cheng Peng, Jiangxuan Long, Weidi Xu, Wei Chu, and Yuan Qi. Harnessing negative signals: Reinforcement distillation from teacher data for llm reasoning. arXiv preprint arXiv:2505.24850, 2025.
>
> [2]. Yige Yuan, Teng Xiao, Shuchang Tao, Xue Wang, Jinyang Gao, Bolin Ding, and Bingbing Xu. Incentivizing reasoning from weak supervision. arXiv preprint arXiv:2505.20072, 2025.
>
> [3]. Daya Guo, Dejian Yang, Haowei Zhang, Junxiao Song, Ruoyu Zhang, Runxin Xu, Qihao Zhu, Shirong Ma, Peiyi Wang, Xiao Bi, et al. Deepseek-r1: Incentivizing reasoning capability in llms via reinforcement learning. arXiv preprint arXiv:2501.12948, 2025.

---

> ### Author Response · Authors · 2025-11-25
> **Part [2/3] Author Response**
>
> ## W2. More human evaluation of rationale quality
>
> Thank you for your valuable advice. We fully agree that a more detailed evaluation of the rationales would make our evaluation more complete. Because such evaluations are inherently subjective and difficult to scale, we originally relied on case study in the original manuscript (Section 4.3.2 and Appendix F, J). Nevertheless, we agree that more statistical analyses would strengthen our claims. Following your advice, we **performed statistical comparisons to the extent permitted by our resources**.
>
> Specifically, we constructed ten graduate-level questions based on recent publications (within the past two years) from several influential chemistry journals. The questions cover different major subfields of chemistry, including organic chemistry, inorganic chemistry, materials chemistry, analytical chemistry, and polymer chemistry. Then, different reasoning LLMs are leveraged to solve these questions through human-AI interactions. Five graduate-level chemistry experts were hired to evaluate these interactions across the following five dimensions (each scored on a 5-point scale) to comprehensively assess the strengths and limitations of our model under scientific interactive situations:
>
> * **Chemical Correctness**: The correctness of chemical knowledge and logic demonstrated throughout the reasoning process.
> * **Answer Accuracy**: Whether the final answer is correct.
> * **Analytical Coverage**: The extent to which different plausible possibilities are explored during reasoning.
> * **Reasoning Coherence**: Whether the reasoning remains focused, coherent, and aligned with the problem.
> * **Effective Information Density**: The density of useful information in the reasoning chain, reflecting the friendliness and efficiency of interaction.
>
> The results are summarized in Table R1, from which we draw three main conclusions.
>
> **Table R1.** Results of human evaluation for human-AI interactions. (As the rationales of o4-mini are not accessible, we only use the answer generated by o4-mini.)
>
> | Models | Chemical Correctness | Answer Accuracy | Analytical Coverage | Reasoning Coherence | Effective Information Density |
> |:---|:---:|:---:|:---:|:---:|:---:|
> | ether0 | 1.56 | 1.60 | 1.40 | 1.76 | 1.56 |
> | DeepSeek-R1 | 4.10 | 3.88 | **4.34** | 4.42 | 3.62 |
> | Qwen3-14B (think) | 3.56 | 3.92 | 4.24 | 3.98 | 3.00 |
> | o4-mini | 3.94 | 3.96 | 3.84 | 4.16 | 4.12 |
> | ChemDFM-R | **4.52** | **4.28** | 3.94 | **4.62** | **4.70** |
>
> First, **our model outperforms all baselines, including DeepSeek-R1, in both chemical correctness and answer accuracy**, indicating that it possesses a stronger grasp of chemical knowledge. In particular, our model shows a clearer advantage on SMILES-related tasks, demonstrating a more precise understanding of molecular structures.
>
> Second, **our model achieves significantly higher scores in effective information density**. Unlike models such as Qwen3-14B and DeepSeek-R1, which often generate extremely lengthy reasoning chains exceeding 1,000 tokens, our model produces concise yet informative reasoning. This considerably reduces the burden on human users when verifying correctness or identifying errors.
>
> Finally, **our model performs slightly lower than Qwen3-14B and DeepSeek-R1 on analytical coverage**. This is partially because their extremely lengthy reasoning chains allow them to enumerate many possible considerations. It is also true that our model commonly focuses on one or a few major possibilities rather than adequately enumerating all potential factors. This highlights a productive direction for future enhancement of our model’s analytical breadth.
>
> We have added these results and analyses to Section 4.3.1. The ten origin question used for this evaluation are documented in Appendix I.
>
> ## W3. Datasets releasing
>
> Thank you for your advice. We are more than willing to contribute to the open-source community by releasing the resulting models and our training data. We are working on the preparation of the releasing. We will release the resulting models, including ChemDFM-R and ChemDFM-I, as well as our SFT dataset, mixed-source distillation dataset, RL dataset, and the functional group identification toolkit upon publication.

---

> ### Author Response · Authors · 2025-11-25
> **Part [3/3] Author Response**
>
> ## Q1. Conflict handling when multiple transformations are possible
>
> Thank you for your question. As Table R1 in our response to W2 demonstrated, although our model achieves a substantial improvement in effective information density, there is still considerable room for enhancing analytical coverage. When multiple transformations or reasoning paths are possible, our model currently tends to focus on the single possibility it considers most likely. It less frequently enumerates and contrasts multiple alternative possibilities during reasoning. This behavior sometimes leads to incorrect conclusions. This limitation is partly a consequence of our distillation design, where we prioritized reducing redundant reasoning and ensuring conciseness. While this greatly improves the effective information density of the reasoning chains and enhances interaction efficiency and frendliness, it inevitably introduces a trade-off that reduces analytical coverage. Improving analytical coverage while maintaining concise reasoning will be one of our key directions for future work.
>
> ## Q2. Hallucinating rationales despite producing correct answers
>
> Thank you for this very insightful question. To answer briefly, our model can indeed exhibit such behaviors. We manually checked 45 sampels from three different tasks. Among them, 5 samples achieve correct answers despite errors in their rationale. One examples is shown below where the incorrect parts is highlighted in italics:
>
> > **Question:** Given the reactants C1CCOC1.[Li+].COC(=O)COc1ccc(Cl)c2nc(C(C)C)c(Cc3ccc(-n4cccn4)cc3)c(C)c12.[OH-], what is the SMILES notation for the resulting product?
> >
> > **ChemDFM-R Rationale:** To derive the product, one first examines the reactants and identifies the ester functional group within the structure, which is known to be reactive toward nucleophilic attack under basic conditions. Recognizing that the reaction involves hydroxide ions, which are strong bases that *deprotonate the ester to form an alkoxide, one then envisions a mechanism in which the alkoxide acts as a nucleophile, attacking the electrophilic carbonyl carbon of the ester.* This nucleophilic attack leads to the formation of a tetrahedral intermediate, which then collapses to expel a leaving group, thereby converting the ester into a carboxylic acid. The presence of lithium ions helps stabilize the deprotonated species during these steps. The overall transformation is consistent with a base-catalyzed hydrolysis reaction that yields the carboxylic acid functional group in the final product.
> >
> > **Answer:** O=C(O)COc1ccc(Cl)c2nc(C(C)C)c(Cc3ccc(-n4cccn4)cc3)c(C)c12
>
> However, **this inconsistency between reasoning and answers is a common challenge that also affects general reasoning LLMs**[4] and **warrants dedicated study as a standalone research question**. **In this work, we focus on a more foundational question: how to train a better chemical generalist reasoning LLMs.** We propose to (1) enrich the model’s reasoning "ingredients" with atomized functional-group knowledge, and (2) to conduct chemical-specific rationale learning with improved efficiency via mixed-source distillation. As a result, (1) compared with traditional non-reasoning chemical LLMs, our model produces more informative and detailed reasoning chains that offer chemists additional angles to validate answers; (2) compared with general reasoning LLMs, our model outputs more accurate and more efficient reasoning chains with higher information density, which are much easier for chemists to interact with.
>
> While the phenomenon you mentioned is not the main target of this work, we fully agree that it is crucial for improving the interaction quality and interpretability of chemical reasoning LLMs. This will be one of our primary directions in future work.
>
> [4]. Yi Chen, Yuying Ge, Rui Wang, Yixiao Ge, Junhao Cheng, Ying Shan, and Xihui Liu. Grpo-care: Consistency-aware reinforcement learning for multimodal reasoning, arXiv preprint arXiv:2506.16141, 2025.

---

> ### Author Response · Authors · 2025-12-03
> **Summarization of the discussion with Reviewer Pn3p**
>
> Dear ACs/PCs,
>
> Reviewer Pn3P **speaks highly of the intergration of functional-group knowledge and our comprehensive evaluations**. His/her major concerns focus on the realization details, rationale-quality evaluations, and dataset release. The point-to-point summary are as follows.
>
> * W1: Implementation details. (Part [1/3])
>
>     We are not fully certain which specific details the reviewer finds insufficient, as the two examples mentioned appear to have been covered in the original manuscript. Nevertheless, we kindly request clarification from the reviewer, and in the meantime, we have provided a more details introduction and explanation in the response.
>
> * **W2: Quality evaluation of rationales. (Part [2/3])**
>
>     We have **added a new human-assessment experiment to quantitatively evaluate the effectiveness and efficiency of ChemDFM-R’s rationales** during human–AI interaction in practical chemical research scenarios.
>
> * W3: Dataset releasing (Part [2/3])
>
>     We promise to open-source both the resulting models, including ChemDFM-R and ChemDFM-I, and the training materials, including our SFT dataset, mixed-source distillation dataset, RL dataset, and the functional group identification toolkit, upon publication.
>
> * Q1: Conflict handling when multiple transformations are possible. (Part [3/3])
>
>     This insightful question reveals an inherent trade-off in rationale generation: conciseness vs. analytical coverage. ChemDFM-R currently prioritizes conciseness, which indeed compromises analytical coverage. Improving analytical completeness while maintaining concise, high-density rationales will be one of our main directions for future research. Detailed discussions are provided in Part [3/3].
>
> * Q2: Will the model hallucinating rationales despite producing correct answers. (Part [3/3])
>
>     Upon manually inspection, ChemDFM-R can occasionally exhibit such behavior. However, we emphasize that such rationale hallucination is a widespread challenge across all reasoning LLMs, which is not the main research question of our work. This will be one of our future work. Please refer to Part [3/3] for detailed analyses.

---

### Official Review · Reviewer_x7Fz · 2025-10-31

**Soundness:** 2
**Presentation:** 2
**Contribution:** 2
**Rating:** 2
**Confidence:** 4

**Summary:**

The authors present a three-step approach to training chemical reasoning models, consisting of: a) pretraining on unstructured chemical data, b) supervised fine-tuning (SFT), and c) reinforcement learning (RL). For (a), the authors collect a pretraining dataset composed of general text corpora from the chemical domain, as well as functional-group annotated molecules and reactions. SFT is performed on three task categories: molecule-centered tasks, reaction-centered tasks, and general-knowledge tasks (e.g., QA questions derived from scientific publications). The authors demonstrate on two benchmarks that their model achieves competitive results, and that each training stage consistently improves performance when comparing models before and after the respective steps.

**Strengths:**

- **(S1 - relevance/novelty) - Training data as a valid contribution.** The authors provide both unstructured data for pretraining and SFT data within the chemical domain. This constitutes a potentially valuable contribution to the community. (However, the quality of the data cannot be assessed, as no supplementary material has been provided.)

- **(S2 - significance) - Results indicate the effect of the deployed training stages.** The results suggest that all three training stages (pretraining, SFT, and RL) contribute to improved performance (however, see (W3) and (W4)), resulting in a final model that is competitive with state-of-the-art generalist LLMs.

**Weaknesses:**

- **(W1 - relevance, novelty, quality) - Core research question has already been answered by prior work.** The main research question — whether models can be trained to reason about chemistry despite chemistry-specific challenges — has been addressed previously [1]. The authors completely ignore this, setting a misguided focus for their work and effectively re-answering an already resolved question. In fact, the authors present an alternative approach to developing a chemical reasoning model, where the main difference lies in the inclusion of pretraining on domain-specific data — a step that [1] does not include. A more interesting research question would have been whether the proposed approach constitutes a more efficient training procedure compared to [1], e.g., in terms of training FLOPs. However, the authors never explore this question and do not compare to [1], making (a) the manuscript low in novelty and (b) the proposed approach of unclear relevance.

- **(W2 - quality) - Claims not backed by insights or references.** Several claims appear overstated and are not substantiated by prior work. For example:
    * "The reasoning-before-answering pattern directly demonstrates how and why the LLM arrives at the answer" (l40f). While reasoning traces may indeed provide insights, this statement feels overly strong. Especially for long traces, it is highly unclear which tokens actually contribute to the final answer.
    * "The advanced domain knowledge is typically insufficient in general-purpose corpora" (l50f). This is the central claim on which the authors’ approach is based. However, it is not supported by a) literature references or b) empirical results in this manuscript. Regarding a), the training data of the base models are largely unspecified. Regarding b), since knowledge about the base models’ training data is missing, their performance can serve as a proxy. As shown in Table 2, the base model appears to include some chemistry knowledge. Notably, Chem-DFM-I outperforms the base model, so it is expected that RL training with Chem-DFM-I would be more efficient. However, it remains unclear whether this is truly the case and whether it justifies the additional pretraining step.

- **(W3 - significance) - Missing error bars and statistical tests.** All results are reported without error bars or statistical tests. Therefore, observed performance differences might have arisen by chance. While the proposed training procedure is computationally expensive and multiple training runs may not be feasible, error bars could at least have been reported across tasks or samples.

- **(W4 - relevance/quality) - Potential problem of data leakage.** Since the presented experiments aim to measure reasoning capability rather than memorization, the authors should ensure that no molecule or reaction appearing in the evaluation benchmarks is included in the pretraining or SFT data.

### References
* [1] Narayanan. *Training a Scientific Reasoning Model for Chemistry.*

### Minor comments
* Functional groups are not well defined. Although the Appendix provides more details, the main manuscript might incorrectly imply that the concept of “functional group” is unambiguous. The authors should clarify this.
* Details of the RL training step are insufficient. The authors mention a multi-task RL scheme, but its realization is not described. For example: were different tasks randomly sampled? Did the authors employ curriculum learning, and if so, did individual tasks require distinct curricula? The exact training data used also remain unspecified.

**Questions:**

- Annotating functional groups is known to be an NP-hard problem. Could the authors please provide more details on their annotation mechanism or model?

- The effectiveness of the functional-group annotated data remains unclear. What performance gain does pretraining only on the unstructured chemical text yield? Conversely, what happens if pretraining is performed solely on the functional-group annotated data?

- What is the completion length of the reasoning traces? I ask because all traces were reportedly checked by human experts, which may indicate that the traces are relatively short. Notably, recent work has shown a clear correlation between the reasoning budget and model performance.

---

> ### Author Response · Authors · 2025-11-25
> **Part [1/7] Author Response**
>
> Thank you for your thoughtful comments and helpful suggestions. We respectfully consider that part of the concerns may stem from our insufficiently clear descriptions, which might have caused misunderstandings. Other comments indeed point out meaningful directions for improvement. In response, we have added more detailed explanations and revised the manuscript accordingly. We hope that our responses can help clarify the misunderstandings and further improve the quality of the paper. We sincerely welcome any further discussions or suggestions.

---

> ### Author Response · Authors · 2025-11-25
> **Part [2/7] Author Response**
>
> ## W1. Comparison with ether0 to clarify our contribution
>
> Thank you for your kind advice. In our Humble Opinion, you may misunderstand the core question of our manuscript. You described our work as focusing on “*whether* a chemical reasoning model can be trained.” However, strictly speaking, this is not a well-defined research question, as general-purpose reasoning models (e.g., DeepSeek-R1) can already reason about chemistry, though their performance remains poor. Consequently, **our study aims not at *whether* such models can be trained, but rather *how* to *better* train a *chemical generalist reasoning* model.** This phrasing contains three key qualifiers, '**chemical-domain-specific**', '**generalist**', and '**reasoning**'. Each of them corresponds to a different perspective of related work. We present a detailed comparison of these aspects in Table R1 below.
>
> **Table R1.** Comparison of our work with different perspectives of related work
>
> | Examples | Chemical-Domain-Specific | Generalist | Reasoning | Drawback |
> | :----- | :-----: | :-----: | :-----: | :----- |
> | DeepSeek-R1, o4-mini | No | Yes | Yes | Limited chemical capability and task performance |
> | ChemLLM-20B, ChemDFM-v1 | Yes | Yes | No | Limited overall performance and low interpretability |
> | ether0 | Yes | No | Yes | Limited task generalizability and unable to engage in unseen tasks and open-ended conversation |
>
> **Based on the clarification above, ether0 is not in conflict or competition with our work.** As its HuggingFace page[1] states, “ether0 is not a general purpose chat model, but a multi-task model specifically for the trained tasks.” It is a strong and pioneering study in chemical reasoning, and we apologize for not citing and comparing it earlier. To more clearly show this distinction and to complete our benchmark comparisons, we **evaluated ether0 on SciknowEval and ChemEval**. The results are shown below as Table R2 and have been updated to Table 3 in the revised manuscript.
>
> **Table R2.** Benchmark results of ether0 and ChemDFM-R
>
> | Models | SciKnowEval | | | | ChemEval | | | |
> | :----- | :-----: | :-----: | :-----: | :-----: | :-----: | :-----: | :-----: | :-----: |
> | | text | mol. | react. | all | text | mol. | react. | all |
> | ether0 | 38.7 | 33.7 | 66.0 | 39.8 | 9.4 | 14.3 | 11.2 | 11.0 |
> | ChemDFM-R | **76.7** | **51.1** | **93.8** | **69.1** | **78.3** | **83.5** | **58.5** | **73.8** |
>
> We have also **tested ether0 on the practical case demonstrated in Figure 5 of our main manuscript**. The full return of ether0 is provided in **Table R3 in a separate comment** to this, where ether0 **misunderstands both the querying reaction, OER, and the intention of the instruction**. As a result, it generates uniformative reasoning and provides undisered answer. When we try to correct it by giving more information on OER at Turn 2, ether0's output is **very much similar to that of Turn 1**, and it was **no longer able to follow the correct reasoning format** (<\|think_end\|> and <\|answer_start\|> are missing). We try multiple different ways of phrasing the query of Turn 2, but observe consistent behavior across all attempts.
>
> Based on these results, ether0 does not generalize well to unseen tasks and does not support free-form human-AI interactions. For this reason, it does not address the main research question of our work: exploring how to better train a chemical generalist reasoning LLM.
>
> To tackle our research question, we introduce two key components: (1) enhancing domain knowledge via fine-grained, atomized functional-group knowledge, and (2) domain-specific rationale learning through mixed-source distillation. These methodological contributions are, to our knowledge, not introduced in ether0. Our experiments have shown that our methods can effectively train a chemical generalist reasoning model (as shown in Tables 2 & 3) and our method is better than the one proposed by ether0 (as shown Tables R2, R3, and Figure 5).
>
> **In summary,** our core research question is not *whether* such models can be trained, but rather *how* to *better* train a *chemical generalist reasoning* model. We apologize for our oversight in not citing ether0, and we will revise the paper to clarify the differences and include comparisons. It is a nice pioneering work, but does not answer our research question. We hope this explanation resolves your concerns and are happy to discuss further if needed.
>
> [1]. futurehouse/ether0 · Hugging Face, 6 2025. URL https://huggingface.co/futurehouse/ ether0. [Online; accessed 2025-11-17].

---

> > ### Author Response · Authors · 2025-11-25
> > ****Table R3.** Ether0's performance on the practical case shown in Figure 5.**
> >
> > > **Turn1:** What is oxygen evolution reaction(OER)? Please propose a reasonable mechanism of heterogeneous OER under acidic conditions.
> > >
> > > **ether0:** *<\|think_start\|>* Okay, let's see... The user is asking about an oxygen evolution reaction (OER), which is a chemical reaction involving oxygen gas. The conditions mentioned are acidic, so the reaction is probably taking place in an acidic medium. The user is asking about a reasonable mechanism of OER under these conditions.
> > >
> > > First, I need to figure out what OER reactions are known to occur under acidic conditions. OER is a type of chemical reaction where oxygen gas (O2) is involved. Common OER reactions include oxidations, possibly of alcohols or alkenes. Given the acidic conditions, the reaction might involve the oxidation of a secondary alcohol to a ketone or the oxidation of an alkene to a carboxylic acid.
> > >
> > > Let's consider the most plausible reactions. One possibility is the oxidation of a secondary alcohol to a ketone. For example, if the starting material is a secondary alcohol, the oxidation would convert the -OH group to a =O group. This would form a ketone.
> > >
> > > Alternatively, if the starting material is an alkene, the oxidation could convert the alkene to a carboxylic acid. For example, if the starting material is a substituted alkene, the oxidation would form a carboxylic acid.
> > >
> > > Given the user's question, the most straightforward answer is the oxidation of a secondary alcohol to a ketone. This is a common reaction in organic chemistry.
> > >
> > > Therefore, the answer is to propose a secondary alcohol oxidation to a ketone. For example, if the starting material is a secondary alcohol, the product would be a ketone. The SMILES notation for the starting material and product would be:
> > >
> > > Starting material: C(C)(C)O
> > > Product: C(C)(C)=O
> > >
> > > Alternatively, if the starting material is an alkene, the product would be a carboxylic acid. For example, if the starting material is CH2=CHCH2COOH, the product would be CH2=CHCH2COOH.
> > >
> > > Given the user's question, the most straightforward answer is the oxidation of a secondary alcohol to a ketone. Therefore, the answer is to propose a secondary alcohol oxidation to a ketone. *<|think_end|>*
> > > *<|answer_start|>* CC(=O)C *<|answer_end|>*
> > >
> > > -------------------------
> > >
> > > **Turn2:** OER is a reaction that oxidizes water to oxygen in an electrocatalytic process, not a reaction that oxidizes secondary alcohols to ketones. Please reconsider and propose a reasonable mechanism of heterogeneous OER under acidic conditions.
> > >
> > > **ether0:** *<|think_start|>* Okay, let's see... The user is asking about an oxygen evolution reaction (OER), which is a chemical reaction involving oxygen gas. The conditions mentioned are acidic, so the reaction is probably taking place in an acidic medium. The user is asking about a reasonable mechanism of OER under these conditions.
> > >
> > > First, I need to figure out what OER reactions are known to occur under acidic conditions. OER is a type of chemical reaction where oxygen gas (O2) is involved. Common OER reactions include oxidations, possibly of alcohols or alkenes. Given the acidic conditions, the reaction might involve the oxidation of a secondary alcohol to a ketone or the oxidation of an alkene to a carboxylic acid.
> > >
> > > Let's consider the most plausible reactions. One possibility is the oxidation of a secondary alcohol to a ketone. For example, if the starting material is a secondary alcohol, the oxidation would convert the -OH group to a =O group. This would form a ketone.
> > >
> > > Alternatively, if the starting material is an alkene, the oxidation could convert the alkene to a carboxylic acid. For example, if the starting material is a substituted alkene, the oxidation would form a carboxylic acid.
> > >
> > > Given the user's question, the most straightforward answer is the oxidation of a secondary alcohol to a ketone. This is a common reaction in organic chemistry.
> > >
> > > Therefore, the answer is to propose a secondary alcohol oxidation to a ketone. For example, if the starting material is a secondary alcohol, the product would be a ketone. The SMILES notation for the starting material and product would be:
> > >
> > > Starting material: C(C)(C)O
> > > Product: C(C)(C)=O
> > >
> > > Alternatively, if the starting material is an alkene, the product would be a carboxylic acid. For example, if the starting material is CH2=CHCH2COOH, the product would be CH2=CHCH2COOH.
> > >
> > > Given the user's question, the most straightforward answer is the oxidation of a secondary alcohol to a ketone. Therefore, the answer is to propose a secondary alcohol oxidation to a ketone. *<|answer_end|>*

---

> > ### Comment · Reviewer_x7Fz · 2025-11-25
> > **First answer to the authors**
> >
> > Dear authors,
> >
> > thank you for your detailed responses. They clearly reflect the effort you put into addressing my concerns. I will carefully review your replies and the updated manuscript in the coming days.
> >
> > I also appreciate your clarity regarding my potential misunderstanding of your research question. I decided to respond directly and clarify my initial interpretation before reading the updated manuscript, so that you have the opportunity to highlight any remaining misunderstandings.
> >
> > I fully agree that the question of whether a reasoning model can be trained on chemical tasks is not well defined. Nonetheless, this was essentially the main conclusion I drew from your results: a model that is RL-trained on chemical tasks outperforms both generalist models, and expert models without RL tuning. This result, however, has already been demonstrated in prior work. For example, Ether0 showed that RL training on chemical tasks improves performance on seen tasks, even though, as you correctly note, Ether0 generalizes poorly to unseen tasks and cannot be used as a chatbot due to its narrow specialization.
> >
> > Your manuscript proposes a multi-stage training pipeline consisting of domain-specific pretraining followed by RL training. However, the current experimental design does not allow you to determine whether the domain-specific pretraining stage is actually necessary. Prior work has shown that chemistry-task RL training directly on a general-corpus-only pretrained model can already be successful. It would therefore have been informative to compare the same base model (a) RL-tuned directly and (b) RL-tuned after your proposed domain-specific pretraining step. Such a comparison would help assess whether domain-specific pretraining leads to more efficient RL tuning, potentially justifying the additional computational cost of the domain-specific pretraining stage.
> >
> > Since this comparison was not conducted, it remains unclear whether the domain-specific pretraining stage is required or offers advantages over a simpler RL-only approach. Consequently, the conclusion that can be drawn from your current experiments is essentially the one stated above, which is why I originally interpreted this as the central research question of your manuscript.

---

> ### Author Response · Authors · 2025-11-25
> **Part [3/7] Author Response**
>
> ## W2. Unsupported claims
>
> Thank you for your valuable feedback. To the best of our understanding, the claims you highlighted are generally regarded as accepted arguments or assumptions in the respective field. We agree that providing additional supporting materials would enhance their reliability. We also acknowledge that some of our wording may have been inappropriate, and we apologize for any confusion caused. We will make the necessary revisions. Specifically:
>
> 1. **The reasoning-before-answering pattern directly demonstrates how and why the LLM arrives at the answer.** First of all, we want to clarify that we are not referring to the latent computations performed by LLMs during generation. Rather, the key idea is that the content in the rationales can assist chemists in validating answers or correcting errors, thus making chemical reasoning LLMs more reliable and verifiable than models without reasoning capabilities. While this represents a widely adopted assumption in reasoning LLM research[2,3], we acknowledge that it remains a hypothesis without formal proof or algorithmic guarantee. Accordingly, we have softened our wording to: “The reasoning-before-answering pattern **can indicate** the LLM thinks this is the answer.”
> 2. **The advanced domain knowledge is typically insufficient in general-purpose corpora.** To our understanding, this is a basic argument for most domain-specific LLMs work[4,5], since the general-purpose corpora usually contain knowledge of a tremendous amount of domains and the advanced knowledge for a specific domain will be insufficient. **As you have mentioned, the performance of LLMs on domain tasks can effectively prove this.** As shown in Table 2, ChemDFM-I, which is the model enhanced with the atomized chemical knowledge, largely outperforms the base general LLM, Qwen2.5-14B-Instruct. Moreover, in Table 3, the chemical enhanced model, ChemDFM-I, even outperforms advanced models such as DeepSeek-R1 and GPT-4o. The relevant data have been compiled in Table R4 to allow straightforward comparison.
>
>     **Table R4.** Benchmark results to show the insufficiency of advanced domain knowledge in general LLMs by performance differences.
>
>     | Models | SciKnowEval | | | | ChemEval | | | |
>     | :----- | :-----: | :-----: | :-----: | :-----: | :-----: | :-----: | :-----: | :-----: |
>     | | text | mol. | react. | all | text | mol. | react. | all |
>     | Qwen2.5-14B-Instruct | 77.0 | 35.7 | 72.1 | 61.3 | 79.6 | 24.8 | 46.8 | 57.7 |
>     | GPT-4o | 76.1 | 44.6 | 44.6 | 61.2 | **82.7** | 35.0 | 52.9 | 63.3 |
>     | DeepSeek-R1 | **80.1** | 30.8 | 83.0 | 62.2 | *81.2* | 19.7 | 47.9 | 57.6 |
>     ||||||||||
>     | ChemDFM-I | *78.8* | *50.6* | *91.1* | **69.7** | *81.2* | *68.4* | *54.6* | *70.7* |
>     | ChemDFM-R | 76.7 | **51.1** | **93.8** | *69.1* | 78.3 | **83.5** | **58.5** | **73.8** |
>
>     Moreover, in the newly added ablation study based on Qwen2.5-1.5B in Appendix H, there is a fairer comparison between the model with domain knowledge enhancement and that without it. The relative results are demonstrated in Table 11 and transcribed below as Table R5.
>
>     **Table R5.** Benchmark results to show the effectiveness of domain pretraining and indirectly prove the claim that general domain LLMs generally possess insufficient advanced chemical knowledge. (DP represents Domain Pretraining)
>
>     | DP Corpus Composition ||SciKnowEval||||ChemEval||||
>     |:---:|:---:|:---:|:---:|:---:|:---:|:---:|:---:|:---:|:---:|
>     | Atomized Knowledge| Text-based Knowledge | text | mol. | react. | all | text | mol. | react. | all |
>     | No  | No  | **66.9** | 30.0 | 34.5 | 49.9 | 44.6 | 25.5 | 25.4 | 34.8 |
>     | Yes | Yes | 66.2 | **31.7** | **37.4** | **50.5** | **53.7** | **26.9** | **31.0** | **41.7** |
>
>    The model with domain knowledge enhancement (Row 2) considerably outperforms that without enhancement. We believe these performance differences in Table R3 and R4 can to some extent support this argument.
>
> [2]. Daya Guo, Dejian Yang, Haowei Zhang, Junxiao Song, Ruoyu Zhang, Runxin Xu, Qihao Zhu, Shirong Ma, Peiyi Wang, Xiao Bi, et al. Deepseek-r1: Incentivizing reasoning capability in llms via reinforcement learning. arXiv preprint arXiv:2501.12948, 2025.
>
> [3]. Jason Wei, Xuezhi Wang, Dale Schuurmans, Maarten Bosma, Fei Xia, Ed Chi, Quoc V Le, Denny Zhou, et al. Chain-of-thought prompting elicits reasoning in large language models. Advances in neural information processing systems, 35:24824–24837, 2022.
>
> [4]. Shanghua Gao, Ada Fang, Yepeng Huang, Valentina Giunchiglia, Ayush Noori, Jonathan Richard Schwarz, Yasha Ektefaie, Jovana Kondic, and Marinka Zitnik. Empowering biomedical discovery with ai agents. Cell, 187(22):6125–6151, 2024.
>
> [5]. Botao Yu, Frazier N Baker, Ziqi Chen, Xia Ning, and Huan Sun. Llasmol: Advancing large language models for chemistry with a large-scale, comprehensive, high-quality instruction tuning dataset. First Conference on Language Modeling, 2024.

---

> ### Author Response · Authors · 2025-11-25
> **Part [4/7] Author Response**
>
> ## W3. Error bars and statistical tests
>
> Thank you for your great advice. We agree that error bars are important for the reliability of the results. We do not include error bars in our initial manuscript mainly due to the budget limit. (The full training of our model, starting from domain pretraining to RL, consumes roughly 30,840 GPU hours. Detailed information has been added to Appendix E.2.) To increase the reliability of our results, we follow your advice to run multiple inferences with different random seeds and report the average performances and standard deviations in Table 2. For your convenience, we have also transcribed Table 2 below as Table R6.
>
> **Table R6.** The average performance of five inference runs and the standard deviation (in parentheses) on SciKnowEval and ChemEval.
>
> | Model | SciKnowEval |||| ChemEval ||||
> | :--- | :---: | :---: | :---: | :---: | :---: | :---: | :---: | :---: |
> || text | mol. | react. | all | text | mol. | react. | all |
> | Qwen2.5-14B-Instruct | 77.0(0.11) | 35.7(0.23) | 72.1(0.24) | 61.3(0.12) | 79.6(1.43) | 24.8(0.63) | 46.8(1.81) | 57.7(0.72) |
> | ChemDFM-I | **78.8**(0.38) | 50.6(0.40) | 91.1(0.70) | **69.7**(0.23) | **81.2**(2.48) | 68.4(3.56) | 54.6(5.19) | 70.7(1.10) |
> | ChemDFM-R | 76.7(0.18) | **51.1**(0.56) | **93.8**(0.37) | 69.1(0.27) | 78.3(2.38) | **83.5**(1.89) | **58.5**(2.61) | **73.8**(1.83) |
>
> The results in Table R6 are consistent with the analyses presented in our original manuscript: the performances on text-centric tasks remain almost intact while those on molecule-centric and reaction-centric tasks increase significantly. Moreover, your suggestion helped us uncover a new and interesting analysis. Results in Table R6 show that **the atomized chemical knowledge enhancement stage consistently improves performance across all task categories**, while **the chemical rationale learning stage further strengthens performance on molecule- and reaction-centric tasks**. We attribute this phenomenon to two factors. *First*, the molecule and reaction-centric tasks typically demand more domain-specific chemical reasoning, such as molecular property prediction or retrosynthesis analysis. In contrast, text-centric tasks rely more on natural language understanding, such as chemical named entity recognition and literature question-answering. As a result, learning chemical reasoning over molecules and reactions provides limited benefit to these text-focused tasks. *Second*, for the sake of answer verifiability, the reinforcement learning tasks do not include purely text-based tasks, which may adversely affect the model’s text reasoning ability. Incorporating text-related tasks into the RL stage through joint training might help preserve performance on text-based tasks. These analyses have been updated to Section 4.1.1.
>
> ## W4. Potential problem of data leakage
>
> Thank you for your important question. Data leakage is a crucial problem for the evaluation of LLMs. We have **taken measures to reduce the risk of data leakage as possible as we can.** Specifically, we extract the SMILES of the molecules and reactions from the benchmarks and make sure the same molecules and reactions do not appear in the relative tasks of the training datasets. We apologize that we do not include this important statement in our manuscript, and have added it to our manuscript as Appendix E.1.
>
> ## M1. Missing clear definition of functional groups
>
> We apologize for the confusion. The term “functional group” mentioned in our manuscript refers to the widely used concept in chemistry[6]. We apologize for not making this explicit in the text and have added a citation to the Wikipedia page on functional groups in the main manuscript to avoid any misunderstanding.
>
> [6]. Functional group - wikipedia, the free encyclopedia. URL http://www.taggedwiki. zubiaga.org/new_content/d4b57445be60b968ab0ea76fbc0b95a8. [Online; accessed 2025-11-17].

---

> ### Author Response · Authors · 2025-11-25
> **Part [5/7] Author Response**
>
> ## M2. The details of the RL training
>
> We apologize for the confusion. Since RL is not the main focus of our work, we implement RL training using a standard approach widely used in the general domain. Concretely, all data from different tasks are mixed to form the RL dataset, with random sampling applied during training and without using curriculum learning. The improving of RL training method is one of the major future work of us and will be exploited in our following work.
>
> As for the RL training data, we apologize for not explaining this clearly in the initial manuscript. The RL tasks are a subset of the tasks in the instruction-tuning dataset whose answers can be easily verified. The data for these tasks are sampled from the same sources as the corresponding SFT data (as detailed in Appendix B). We also take care to avoid any overlap between the SFT and RL data for the same tasks whenever possible. We will open-source both our SFT and RL datasets upon publication. To provide clearer context, we include examples of RL data below and add more detailed descriptions in Appendix D and E.3.
>
> ```json
> {
>     "Instruction": "Our research team is working to complete a reaction and requires your expertise. The reaction format is as follows: \"reactant1.reactant2.reactant3...>reagent1.reagent2.reagent3...>product1.product2.product3...\". However, due to contamination, certain elements are missing and represented by a blank (\"___\"). \nIncomplete reaction: CCSc1ccc(O)cc1.C(C)(Br)c1ccnc2ncnn12>___>CCSc1ccc(OC(C)c2ccnc3ncnn23)cc1\nCompletion:",
>     "Label": "COCCOC.[Na+]"
> },
> {
>     "Instruction": "Rank the molecules by their hydration free energy from highest to lowest based on their SMILES. SMILES List: CC(=O)C1CCCCC1,CCNCC,c1ccc(cc1)C=O,COC(=O)CCl\nOrder of Hydration Free Energy:",
>     "Label": "CCNCC,c1ccc(cc1)C=O,COC(=O)CCl,CC(=O)C1CCCCC1"
> },
> {
>     "Instruction": "Could you specify the molecular formula for the molecule listed as SMILES?\nSMILES: CCOC1CC(CC(O)c2ccc(OC(C)C)cc2)C1\nMolecular Formula:",
>     "Label": "C17H26O3"
> }
> ```
>
> ## Q1. Annotation functional groups is an NP-Hard problem
>
> We appreciate this insightful question. Our functional group identification toolkit is directly implemented based on the RDKit package and SMARTS expressions. Although subgraph matching is an NP-Hard problem, the scale of graphs corresponding to most chemical molecules is pretty limited. To verify this statement, we conducted a statistical analysis of the molecules in our corpus and summarized the results as follows:
>
> * The graphs corresponding to the molecules in our domain pretrian dataset typically contain around **27.09** nodes (atoms) and **29.05** edges (bonds) on average.
> * **99.5%** of the molecules have less than 200 atoms and less than 200 bonds, and **95%** of the molecules have less than 100 atoms and less than 100 bonds.
> * The maximum number of atoms in a single molecule is **337**, and the maximum number of bonds in a single molecule is **441**.
>
> Therefore, it is not necessary to use a polynomial-time algorithm for this task, as our current method is able to complete the annotation within a reasonable time.

---

> ### Author Response · Authors · 2025-11-25
> **Part [6/7] Author Response**
>
> ## Q2. The effectiveness of the functional-group annotated data remains unclear
>
> Thank you for your valuable and constructive advice. The absence of this ablation study in the initial manuscript was mainly due to the restricted computational resources. **Directly verifying the effect of atomized knowledge enhancement would be extremely costly, since it requires repeatedly performing computationally expensive domain pretraining.** However, we totally agree with you that this ablation study would clarify the effectiveness of our atomized knowledge enhancement design, and thus make our work more solid.
>
> Therefore, following your advice, we manage to **achieve this ablation study with a compromise setting**. Specifically, instead of training a model of the same size as ChemDFM-R on the full dataset for comparison, we used Qwen2.5-1.5B as the base model and conducted the full training pipeline on a 10\% subset of ChemDFM-R's data. By varying the data composition in the subset of the domain-pretraining corpus, we trained different versions of models for comparison while keeping the computational cost manageable. The results are presented in Table 11 of the main manuscript and also transcribed below as Table R7.
>
> **Table R7.** Benchmark results to show the effectiveness of domain pretraining and indirectly prove the claim that general domain LLMs generally possess insufficient advanced chemical knowledge. (DP represent Domain Pretraining)
>
> | DP Corpus Composition ||SciKnowEval||||ChemEval||||
> |:---:|:---:|:---:|:---:|:---:|:---:|:---:|:---:|:---:|:---:|
> | Atomized Knowledge| Text-based Knowledge | text | mol. | react. | all | text | mol. | react. | all |
> | No  | No  | 66.9 | 30.0 | 34.5 | 49.9 | 44.6 | 25.5 | 25.4 | 34.8 |
> | No  | Yes | **67.1** | 30.6 | 36.5 | 50.4 | 43.6 | 25.6 | 28.5 | 35.2 |
> | Yes | No  | 65.7 | **31.8** | **37.8** | 50.3 | 51.9 | 25.9 | **33.8** | 40.8 |
> | Yes | Yes | 66.2 | 31.7 | 37.4 | **50.5** | **53.7** | **26.9** | 31.0 | **41.7** |
>
> Compared with the model without any domain pretraining (Row 1), models pretrained on either the text-based-knowledge corpus (Row 2) or the atomized-knowledge corpus (Row 3) show improvements on most tasks. This demonstrates the importance and necessity of domain pretraining for strengthening domain knowledge, and indirectly supports our hypothesis that general domain LLMs generally possess insufficient advanced chemical knowledge. Furthermore, the model pretrained solely on the atomized-knowledge corpus outperforms the model pretrained on the traditional text-based-knowledge corpus on many tasks, despite its corpus being only 20\% of the latter. This provides strong evidence that fine-grained atomized knowledge enables more efficient domain knowledge enhancement. Finally, the model pretrained on the combined text-based- and atomized-knowledge corpora (Row 4) achieves the best overall performance, reflecting the complementary strengths of the two corpora and validating the effectiveness of our atomized-knowledge–enhanced domain pretraining approach.
>
> Due to the space limit, we have added this new ablation study to our revised manuscript as Appendix H.

---

> ### Author Response · Authors · 2025-11-25
> **Part [7/7] Author Response**
>
> ## Q3. Questions reguarding to the length of ChemDFM-R's rationales
>
> Thank you for your valuable question.
>
> First, we would like to point out that **the relationship between rationale length and model performance remains controversial**. While some works in the math or code domains focus on inference-time scaling and show performance gains with longer rationales, these comparisons are typically conducted **within the same model**[7]. There is no guarantee that a model producing longer rationales will always outperform another model whose rationales are shorter. Moreover, excessively long rationales are often considered inefficient and difficult for humans to utilize, motivating research aimed at reducing rationale length while maintaining performance[8,9].
>
> Regarding your question, it is indeed true that the rationales produced by our model are much shorter than those of DeepSeek-R1, which typically contain thousands of tokens including excessive erroneous attempts and meaningless numerical computations. We argue that this actually demonstrates the efficiency of our model: **ChemDFM-R avoids redundant reasoning steps while achieving comparable or even superior performance relative to DeepSeek-R1**.
>
> Movrover, to better assess the quality of our rationales compared with general reasoning LLMs, we conducted a human assessment experiments based on 10 different questions constructed from newly published papers across 5 subfield of chemistry. The detailed results and analyses are demonstrated in Section 4.3.1 and Appendix I. Through this experiment, we found out that compared with general reasoning LLMs such as Qwen3 and DeepSeek-R1, **our model can achieve better responses in terms of chemical correctness and answer accuracy while maintain our rationales reaonably short and our effective information density considerably high.** Specifically, we transcribe the relative results along with an additional statistics about the length of the response in Table R8 below.
>
> **Table R8** Results of human evaluation on rationale quality during human-AI interactions.
>
> | Models | Chemical Correctness | Answer Accuracy | Effective Information Density | Average Words in Response |
> |:---|:---:|:---:|:---:|:---:|
> | DeepSeek-R1 | 4.10 | 3.88 | 3.62 | 1359.0 |
> | Qwen3-14B (think) | 3.56 | 3.92 | 3.00 | 1747.6 |
> | ChemDFM-R | **4.52** | **4.28** | **4.70** | 211.9 |
>
> [7]. Dacheng Li, Shiyi Cao, Chengkun Cao, Xiuyu Li, Shangyin Tan, Kurt Keutzer, Jiarong Xing, Joseph E. Gonzalez, and Ion Stoica. S*: Test time scaling for code generation. In Findings of the Association for Computational Linguistics: EMNLP 2025, pp. 15964–15978, 2025.
>
> [8]. Heming Xia, Chak Tou Leong, Wenjie Wang, Yongqi Li, and Wenjie Li. TokenSkip: Controllable chain-of-thought compression in LLMs. In Proceedings of the 2025 Conference on Empirical Methods in Natural Language Processing, pp. 3351–3363, 2025.
>
> [9]. Jingyang Yi, Jiazheng Wang, and Sida Li. Shorterbetter: Guiding reasoning models to find optimal inference length for efficient reasoning. Advances in neural information processing systems, 2025.

---

> ### Author Response · Authors · 2025-11-26
> **Author response to the first reply**
>
> Dear Reviewer x7Fz,
>
> Thank you very much for your timely and sincere reply. It is extremely helpful for us to better address your concerns and further improve the quality of our manuscript.
>
> **To avoid misunderstandings and ensure a unified interpretation, please allow us to first summarize the common ground between us and your main concerns:**
>
> **You have acknowledged that:**
>
> 1. Our model outperforms previous chemistry LLMs (including ether0) as well as general reasoning LLMs, exhibiting stronger chemical reasoning capabilities.
>
> 2. Our model functions as a generalist chemical reasoner, demonstrating superior task generalization and multi-turn interaction ability, capabilities that ether0 does not possess.
>
> **Your main remaining concern is:**
>
> You would like us to demonstrate the effectiveness of our key methodological designs.
>
> **Please let us know if our summary does not correctly capture the current state of our discussion.**
>
> ---
>
> We sincerely appreciate your recognition of the performance of our model. Your recent reply greatly helped us understand your initial comments. **Instead of ``misunderstanding'' our core research question, your primary concern is that our original experiments only allowed you to arrive at that interpretation.** (Please let us know if this understanding is not accurate, as the difference between these two interpretations is substantial from the authors’ perspective: **the former questions the core value of our work, while the latter requests additional evidence for strengthening our claims.**)
>
> ---
>
> Regarding your core concern, we fully agree that more direct validations of our methodologies would further solidify our work. Our work includes two primary methodologies (We humbly emphasize that RL is only one of the stages in the pipeline rather than the focus of our contribution.):
>
> 1. Using **atomized chemical knowledge** can more efficiently enhance LLM's chemical understanding.
> 2. **Mixed-source distillation** can significantly improve the distillation quality under limited resources, thus improving final reasoning performance.
>
> To validate these:
>
> 1. **Effectiveness of mixed-source distillation**
>
>     This was validated by **the ablation studies in Section 4.2 in our original manuscript**. Since you did not raise concerns regarding this part, we will not elaborate again here.
>
> 2. **Effectiveness of atomized knowledge enhancement**
>
>     In the first round of responses, we added **a new ablation study to validate this**. Given the high computational cost, we conducted parallel experiments on a smaller model and dataset. The results are provided **in Part[6/7] as the response to Q2 and added to Appendix H**. We notice that our original expression may be misleading, potentially suggesting that the evaluated models had received only domain pretraining. We have revised it to avoid misunderstanding. While pratrained on corpora with different compositions, the models are evaluated after completing the full remaining training pipeline, including RL. To avoid redundancy, we do not elaborate here. Please refer to Part[6/7] for detailed analyses.
>
> Additionally, **you explicitly questioned the effectiveness of the whole domain pretraining** in your latest response. **We believe that the results in Table R7 (Part[6/7]) can also address this concern**. In Table R7, Row 1 is the model which is RL-tuned without domain pretraining, while the other Rows are RL-tuned models after domain pretraining from the same base model. Specifically:
>
> * Domain pretraining on either type of knowledge (Rows 2 and 3) improves performance on molecule- and reaction-centric tasks, demonstrating that general models still have notable gaps in their chemical knowledge, while domain pretraining can mitigate these gaps and consequently result in more capable RL-tuned models.
>
> * The corpus combining text-based knowledge and atomized knowledge facilitates the best final performance, validating the design choice of our method.
>
> ---
>
> In addition, **we also wish to emphasize that the resulting model itself already represents a valuable contribution. Previous models have significant limitations**: ChemLLM/ChemDFM cannot provide reasoning chains, making their responses less reliable; ether0 does not support multi-turen, open-ended human-AI interaction under practical situation; DeepSeek-R1/Qwen3 generate excessively long reasoning chains with numerous chemical errors, leading to poor accuracy and low interaction friendliness (supported by the newly added human assessment experiments in Section 4.3.1). Our model mitigates these issues, offering **a more verifiable, more concise, and more practical assistant to chemistry researchers** and **a stronger foundation for future model improvement and chemical agent systems development to the open-source community**.
>
> We hope the above clarification and explanation address your concerns. We sincerely welcome further discussions.

---

> ### Author Response · Authors · 2025-12-03
> **Summarization of the discussion with Reviewer x7Fz**
>
> Dear ACs/PCs,
>
> Although reviewer x7Fz initially gave our paper a score of 2, we are glad to report that **our subsequent communication has led to meaningful progress**. We have **addressed one of his/her two major concerns**. For the remaining one, the reviewer mentioned (prior to the leakage incident) that he/she **had not had time to examine our response but would carefully review it afterward**.
>
> Specifically, both major concerns are related to W1 of his/her original review. **The first major concern of reviewer x7Fz which we believe has been addressed is the original concern he/she stated in W1**, where he/she misidentifies the core research question of our paper and so that reach the conclusion that the previous work ether0 has already solve the research question. **In our response (Part [2/7])**, we clarified our actual research question and conducted a detailed comparison between ChemDFM-R and related works including ether0. We further compared ChemDFM-R and ether0 across benchmark evaluations, case analyses, and human assessments, where ChemDFM-R shows consistently stronger performance. These experimental results have also been added to the corresponding section of our manuscripts.
>
> **In the recent reply from reviewer x7FZ, he/she has recognized these differences and the superior performance of ChemDFM-R compared to ether0.** However, he/she **further clarify a second major concern**. As he/she stated, the strong wording in W1 was **not meant to question the core value of our work**, but rather **reflected the belief that the original experiments were insufficient to substantiate the effectiveness of our methods**, an issue overlapping with Q2 in the original review. In our most recent response, we **systematically presented two ablation studies demonstrating the effectiveness of our two methodological contributions**: the original ablation study in Section 4.2 and the newly added ablation study in Appendix H (also in Part [6/7]). In addition, we added a new analysis directly addressing his/her specific concern regarding the effectiveness of the whole domain-pretraining stage. We believe these new results sufficiently support our contributions and address the reviewer’s concerns.
>
> The point-by-point summary of the remaining comments from reviewer x7Fz's and our responses are as follows:
>
> * **W2: Unsupported claims (Part [3/7]).**
>
>     Although these claims are widely accepted arguments or assumptions in the respective field, we address this concern by **softening our wording and supporting them with 1) previous publications and 2) experimental results.**
>
> * **W3: Missing error bars and statistical tests. (Part [4/7])**
>
>     Following his/her advice, we **run multiple inferences with different random seeds and report the average performances and standard deviations.**
>
> * **W4: Data leakage Problem. (Part [4/7])**
>
>     We clarify that we **have taken measures to reduce the risk of data leakage as possible as we can**. The measures have been briefly introduced in Part[4/7] and detailed in Appendix E.1.
>
> * M1: Missing defination of functional groups. (Part [4/7])
>
>     A footnote citation of the wikipedia page of functional groups have been added to its first appearance.
>
> * M2: Details of RL. (Part [5/7])
>
>     Additional details have been added to Appendix D and Appendix E.3.
>
> * Q1: Annotating functional group is a NP-Hard problem. (Part [5/7])
>
>     We clarify that although it is a subgraph matching problem, the scale of the molecules' graph is very limited. We provide relevant statistics to support this.
>
> * **Q2: The effectiveness of the functional-group annotated data remains unclear. (Part [6/7])**
>
>     We **added a new ablation study (Appendix H) to directly validate this point.**
>
> * Q3: Reviewer x7Fz believes longer rationales will results in stronger performance. Based on this statements, he/she concerns that **the limited length of ChemDFM-R's rationales is a drawback.** (Part [7/7])
>
>     We clarify that the **"longer-is-better" trend generally holds within the same model, and does not necessarily apply across different models**. Moreover, we emphasize that **ChemDFM-R’s superior performance with considerably shorter rationales actually demonstrates the effectiveness and efficiency of its reasoning**. The newly added human-assessment experiment further shows that shorter rationales can be more informative and more interaction-friendly.

---

### Author Response · Authors · 2025-12-03
**Outline of our paper**

## Main Text

* **Section 1** is the introduction of our paper.
* **Section 2** introduces **ChemFG**, including the development of our functional-group identification tool and the validation of its extraction quality. Additional details on large-scale data collection and corpus statistics are provided in Appendix A.
* **Section 3** introduces **the training pipeline of ChemDFM-R, including the details of the proposed mixed-source distillation method**.
* **Section 4** reports our comprehensive experiments:
  * **Subsection 4.1** evaluates ChemDFM-R on representative **benchmarks**, comparing it with both our baseline models (Section 4.1.1) and advanced general LLMs (Section 4.1.2). Extended experimental results are available in Appendix G.
  * **Subsection 4.2** presents the **ablation studies** validating our key design choices. Due to space constraints, the ablation of ChemFG and atomized knowledge is provided in Appendix H, while the ablation of mixed-source distillation remains in this subsection.
  * **Subsection 4.3** demonstrates the capabilities of ChemDFM to **facilitate more verifiable and interpretable human-AI interactions**, through human assessment (4.3.1) and case demonstration (4.3.2 and Appendix J)
* **Sections 5 and 6** are the related work and the conclusion of our paper.

## Appendix

* **Appendix A**: **Details of ChemFG**, including the large-scale data collection process, data statistics, and details and examples of quality control.
* **Appendix B**: **Details of the instruction tuning dataset of ChemDFM-R**, including task compositions and the realization of large-scale instruction rewriting.
* **Appendix C**: **Details of the real distillation data collection**, including example comparison of distillation results and details of quality control.
* **Appendix D**: **Details of reinforcement learning for ChemDFM-R**.
* **Appendix E**: **Training details of ChemDFM-R**, including data-leakage mitigation, computation and data budgets, and training/inference hyperparameters.
* **Appendix F**: **Demonstrations of ChemDFM-R's rationales**.
* **Appendix G**: **Details of benchmark evaluations**, including task-wise performances on ChemEval and Sciknoweval, analyses of different levels of generalizability of ChemDFM-R, and more evaluations on newly published benchmarks.
* **Appendix H**: **Ablation study validating the effectiveness of ChemFG and atomized knowledge based on functional groups**.
* **Appendix I**: **Details of human assessments in Section 4.3.1**, including the definitions of metrics and the exact questions used.
* **Appendix J**: **More case studies illustrating ChemDFM-R's potential** in practical human-AI interaction scenarios, complementing Section 4.3.2.
* **Appendix K**: **Limitation and outlook**.

---

### Author Response · Authors · 2025-12-03
**Summarization of our paper and the improvements during rebuttal**

Dear ACs/PCs,

We sincerely appreciate your valuable time and effort. We understand the tremendous extra workload caused by the current situation, and we hope to ease your burden as much as possible. To this end, we will first **summarize the main contributions of our paper** and **the new evaluations added during the rebuttal to address reviewers' major concerns**. **A brief outline of our paper** is also provided to help you quickly locate the relevant content while reading our rebuttals.

In addition, **a dedicated Official Comment is provided at the end of each reviewer’s discussion, summarizing their concerns and explaining how we address them**.

## Our main contributions

In this work, we aim to answer the question of **how** to **better** train a **chemical generalist reasoning model** to facilitate **more verifiable and interpretable human-AI interactions** in practical chemical research scenarios. There are two major challenges hindering prior LLMs from excelling in chemical reasoning:

1. **Superficial domain knowledge**. Existing chemical corpora primarily consist of literature and high-level phenomena. Such data mainly focus on the direct correlation between molecules and their properties or reactivities, which are highly complex in reality. A proper breakdown into internal mechanisms and atomized knowledge is missing in these corpora.
2. **Lack of chemical reasoning logic**. Chemical reasoning differs fundamentally from that in mathematics and programming. These differences also make it difficult to acquire large-scale, high-quality distillation rationales from current advanced reasoning LLMs.

To overcome these challenges, our contributions are threefold:

1. To overcome the first challenge, we **introduce functional-group knowledge as finer-grained, atomized chemical knowledge to more efficiently enhance the domain knowledge of general LLMs**. We develop a comprehensive identification toolkit and construct a 101B-token corpus, ChemFG.
2. To overcome the second challenge, we **propose a mixed-source distillation method that effectively initializes the reasoning ability of pre-RL LLMs under limited resources**. Instead of relying solely on costly real distillation data, we decompose reasoning ability into three components and enhance each with tailored data sources.
3. Extensive experiments demonstrate that the resulting model, ChemDFM-R, **achieves advanced performance on benchmark evaluations** (Section 4.1) and **generates high-quality, informative rationales in practical chemical scenarios** (Section 4.3). **Ablation studies (Section 4.2 and Appendix H) further confirm the effectiveness of our methodological designs.** Overall, ChemDFM-R offers a more verifiable, concise, and practical assistant for chemical researchers and a strong foundation for future model improvements and chemical agent systems design for the open-source community.

## The new evaluations addressing reviewers' major concerns

1. Comparison with ether0 (Reviewers x7Fz and hZbS).

    Our work differs from ether0 in three perspectives. **In terms of research questions**, we aim at better training a chemical general reasoning LLM, whereas ether0 is designed as a multi-task specialist. **In terms of methods**, our proposed atomized knowledge enhancement and mixed-source distillation methods are not incorporated by ether0. **In terms of performances**, we have added evaluations of ether0 in both benchmark evaluations and human assessments, where ChemDFM-R shows consistently stronger performance.

2. Ablation study for functional group knowledge (Reviewer x7Fz).

    We have added **a new ablation study to demonstrate the effectiveness of domain-pretraining and the atomized functional-group knowledge** in Appendix H, while the effectiveness of mixed-source distillation is validated by the original ablation study in Section 4.2.

3. The risk of data leakage and generalizability validation (Reviewers x7Fz, hZbS).

    We have **taken measures to reduce the risk of data leakage as much as possible**, and the detailed methods are added as Appendix E.1.

    Additionally, we further grouped the tasks into two categories to illustrate **the instruction-level and the task-level generalizabilities** of ChemDFM-R in Appendix G.3.

4. More evaluation on new benchmarks (Reviewer rgJV).

    We have conducted benchmark evaluations on two more benchmarks, FGBench and ChemCOTBench.

5. Statistic comparison of rationale quality (Reviewers Pn3p, hZbS, and rgJV).

    The effectiveness and efficiency of ChemDFM-R's rationales was originally demonstrated only by case study. We have added **a human assessment to quantitatively assess the quality of rationales from different models across multiple metrics**. The results are demonstrated in Section 4.3.1, showing that **ChemDFM-R facilitates more concise, informative, and interaction-friendly human-AI interactions**.

---

### Meta-Review · Area_Chair_61Gr · 2025-12-23

**Summary:**

On the strenghts side, reviewers generally agreed that addressing limitations of general-purpose LLMs in chemistry, where mechanistic reasoning and interpretability are crucial, is an important and timely problem.
The key methodological contribution (highlighted by all reviewers including the one more leaning on negative) is the introduction of functional-group centric atomized chemical knowledge (ChemFG), which aligns well with chemists’ reasoning patterns and provides fine-grained supervision beyond text-only corpora. The proposed mixed-source distillation strategy is viewed as a well-engineered approach to improve reasoning quality when high-quality expert rationales are scarce.
The empirical performance of ChemDFM-R is also a strenght, with good performance across multiple benchmarks (SciKnowEval, ChemEval, ChemCOTBench), often outperforming both domain-specific and general reasoning LLM baselines.

The most significant weakness is novelty and positioning: most reviewers argued that prior work (e.g., ether0) already addressed a very similar problem (how to RL-tune LLMs so that they can reason about chemistry), and questioned whether domain pretraining is strictly necessary beyond RL alone. This issue is worsened by the fact that the original paper failed to reference such very relevant related works (those the Authors recognized this limitation in the rebuttal).
Also, several reviewers initially found overstated claims, particularly regarding interpretability through reasoning traces. Concerns were also raised about reproducibility and transparency, including missing error bars, incomplete RL details, and unavailable datasets and code (again many of these were addressed in the rebuttal).

**Reviewer Concerns:**

The rebuttal substantially improved experimental completeness by adding new benchmarks (FGBench, ChemCOTBench), explicit comparisons with ether0, and ablation studies isolating the impact of atomized functional-group knowledge and mixed-source distillation

Concerns about unsupported claims were handled appropriately: the authors softened overstrong language around interpretability and reliability, added empirical evidence, and clarified assumptions.

Requests for error bars and robustness were partially resolved by reporting mean ± standard deviation over multiple inference runs, though results are still largely based on single full training runs.

Potential data leakage issues are  now explicitly addressed with molecule- and reaction-level de-duplication checks, documented in the revised manuscript.

Reproducibility concerns were only partially mitigated by an explicit commitment to open-sourcing models, datasets, and tooling upon publication, alongside detailed compute and data budget reporting. This is not entirely satisfactory given the heavily empirical nature of the
work and the initial lack of experimental details in the manuscript.

The most persistent issue is novelty and research positioning: despite improved comparisons, at least one reviewer remains unconvinced that the core contribution goes substantially beyond prior RL-based chemical reasoning LLMs, questioning whether domain pretraining is strictly necessary versus RL alone.

Also, the relevance of the most interesting methodological contribution (functional group knowledge) is somewhat reduced by the poor results of the model on FGBench, which specifically targets functional group reasoning.

An additional unsolved limitation concerns the true impact of interpretability of reasoning traces: the authors acknowledge cases where rationales contain chemical errors despite correct answers, and the impact of such failures on claimed interpretability is not convincingly assessed.

**Reviewer Scores:**

Only Reviewer x7Fz posted a reaction on the Authors rebuttal. It seems evident that the reviewer is not convinced by the novelty claims, but at least partially satisfied by some of the answers provided. I would project this to a score raise to 4.

Reviewers Pn3p and hZbS were borderline positive but their criticisms on inconsistency between reasoning and answers as well as on reproducibility were not positively addressed in the rebuttal. So I would expect them not to raise their score nor to champion for the paper during the discussion.

Reviewer rgJV was also unconvinced by the methodological novelty. In addition, this reviewer asked for empirical analysis based on FGBench, which is were the model did not behave accordingly to the the claimed superiority due to the introduction of functional group knowledge. I would expect this reviewer not to raise the score.

Overall, my expectation for the final average score for the paper is a 5.00 with no reviewer championing for acceptance, which would position the contribution below the acceptance bar.

---

### Decision · Program_Chairs · 2026-01-26

Reject